# Reproducing Kernel Banach Space Models for Neural Networks with Application to Rademacher Complexity Analysis

**Alistair Shilton, Sunil Gupta, Santu Rana, Svetha Venkatesh**

Applied Artificial Intelligence Initiative,
Deakin University, Geelong, Australia
`alistair.shilton@deakin.edu.au, sunil.gupta@deakin.edu.au,`
`santu.rana@deakin.edu.au, svetha.venkatesh@deakin.edu.au`

## Abstract

This paper explores the use of Hermite transform based reproducing kernel Banach space methods to construct *exact* or *un-approximated* models of feedforward neural networks of arbitrary width, depth and topology, including ResNet and Transformers networks, assuming only a feedforward topology, finite energy activations and finite (spectral-) norm weights and biases. Using this model, two straightforward but surprisingly tight bounds on Rademacher complexity are derived, precisely (1) a general bound that is width-independent and scales exponentially with depth; and (2) a width- and depth-independent bound for networks with appropriately constrained (below threshold) weights and biases.

## 1 Introduction

A significant challenge in neural networks is understanding how large models, despite their high capacity to overfit training data, can still generalize effectively (Neyshabur et al., 2014). Learning theory tells us that inductive bias plays an important role in explaining this phenomena, where inductive bias is the restriction of the space of potential learned functions (neural networks) to a small subset $\mathcal{F}$ of the total space of space, either explicitly through regularization or implicitly through the training algorithm used. Rademacher complexity (Bartlett & Mendelson, 2002; Steinwart & Christman, 2008) is one measure of the complexity or expressive power of $\mathcal{F}$ that has been used to understand inductive bias through the lens of uniform convergence - that is, the rate at which the empirical risk (on the training dataset of $N$ samples) converges to actual risk (on the data distribution) (for a discussion of alternative approaches see (Valle-Pérez & Louis, 2020)). A representative approach to Rademacher complexity analysis in neural networks is "peeling" (Neyshabur et al., 2015; Golowich et al., 2018; Truong, 2022). In this approach, the total compexity is bounded by "peeling off" the output layer $D$ to extract factors due to that layer and thus express the total Racehmacher complexity as a product of terms due to the output layer $D$ and the Rademacher complexity of the preceeding $(D-1)$-layer network. The process is then repeated, peeling off successive layers until the process terminates at the input layer. This typically results in a bound that exhibits width independence (assuming popular schemes such as LeCun, He or Glorot weight scaling) and exponential depth dependence, and contains some (typically Lipschitz-type) scaling term due to the neural activations as well a (typically depth-exponential) "nuisance factor". For example in (Neyshabur et al., 2015) it is shown that, for a simple unbiased layerwise network with spectral-norm bound weight matrices $\mathbf{W}^{[j-1:j]}$ and Lipschitz activations, the Rademacher complexity is bounded as:

$$\mathcal{R}_N\left(\mathcal{F} : \left\|\mathbf{W}^{[j-1:j]}\right\| \leq \omega^{[j-1:j]}\right) \sim \mathcal{O}\left(\frac{2^D \prod_{j=1}^{D} \omega^{[j-1:j]}}{\sqrt{N}}\right)$$

39th Conference on Neural Information Processing Systems (NeurIPS 2025).

This bound can be refined in various ways (eg. (Golowich et al., 2018)), but the basic form remains, as do the nuisance factors (the term $2^D$ in the above bound, for example) in one form or another.

An alternative approach is to construct a bilinear (dual) representation of the model that splits the input $\boldsymbol{x} \in \mathbb{X}$ and parameters $\Theta \in \mathbb{W}$ into separate terms in a dual representation:

$$\mathbf{f}\left(\boldsymbol{x};\Theta\right) = \langle \boldsymbol{\Psi}\left(\Theta\right), \boldsymbol{\phi}\left(\boldsymbol{x}\right)]$$

where $\boldsymbol{\Psi} : \mathbb{W} \to \mathcal{W}$, $\boldsymbol{\phi} : \mathbb{X} \to \mathcal{X}$ are feature maps and $\langle \cdot, \cdot] : \mathcal{W} \times \mathcal{X} \to \mathcal{R}$ is a continuous bilinear product. Examples of this type of model are the neural network Gaussian process (Rasmussen & Williams, 2006) (NNGP) models (Neal, 1996), which treat all layers prior to the output as fixed and model the influence of the weights in the output layer; neural tangent kernel (NTK) models (Jacot et al., 2018; Daniely, 2017; Daniely et al., 2016), which model the (first-order) variation of the weights about their initial values in a reproducing kernel Hilbert space (RKHS) (Aronszajn, 1950) (see for example (Du et al., 2019b; Allen-Zhu et al., 2019; Du et al., 2019a; Zou et al., 2020; Zou & Gu, 2019; Arora et al., 2019b,a; Cao & Gu, 2019)); and reproducing kernel Banach space (RKBS) (Lin et al., 2022; Zhang et al., 2009; Zhang & Zhang, 2012; Song et al., 2013; Sriperumbudur et al., 2011; Xu & Ye, 2014) approaches such as (Shilton et al., 2023), which recursively construct feature maps $\boldsymbol{\Psi} : \mathbb{W} \to \mathcal{W}$, $\boldsymbol{\phi} : \mathbb{X} \to \mathcal{X}$ to exactly model the neural network (beyond first-order).[1] In all cases the utility of the model in the context of Rademacher complexity analysis is that it makes the construction of bounds straightforward through the use of either the Cauchy-Schwarz inequality (if $\langle \cdot, \cdot]$ is an inner product) or the continuity of the bilinear product; and moreover, as peeling is not applied directly to the Rademacher complexity, nuisance factors arising from this procedure may be avoided. However the assumptions made by these models (wide-networks, lazy training, restrictions on neural activations and network topology etc (Arora et al., 2019b; Lee et al., 2019; Bai & Lee, 2019)) can complicate analysis and limit their applicability.

Our goal in this paper is to address two questions, (1) can we formulate an *exact* (non-approximate) model for a wide class of neural networks, including ResNet and Transformers, avoiding entirely the question of gaps between the performance of the neural network and its model; and (2) can such a model be used to derive straightforward, non-vacuous, widely applicable, training-independent bounds on Rademacher complexity *without* nuisance factors. We answer these questions with the following contributions:

1. **Exact RKBS model (Theorem 1):** For feedforward neural network with arbitrary topology, finite weight and biases and finite-energy neural activations, we construct an exact model that recasts neural networks as elements in a reproducing kernel Banach space (RKBS) defined by the bilinear product:

$$\mathbf{f}\left(\boldsymbol{x};\Theta\right) = \langle \boldsymbol{\Psi}\left(\Theta\right), \boldsymbol{\phi}\left(\boldsymbol{x}\right)]_{\mathbf{g}}$$

where $\boldsymbol{\Psi} : \mathbb{W} \to \mathcal{W}$ is a weight/bias feature map, $\boldsymbol{\phi} : \mathbb{X} \to \mathcal{X}$ is a data feature map, and $\langle \cdot, \cdot]_{\mathbf{g}} : \mathcal{W} \times \mathcal{X} \to \mathbb{R}$ is a continuous bilinear form characterized by an indefinite metric $\mathbf{g}$.

2. **Rademacher Complexity Bound (Theorem 4):** We observe that, for our RKBS model:

$$\|\mathbf{f}\left(\boldsymbol{x};\Theta\right)\|_2 \leq C_\Theta \|\boldsymbol{\phi}\left(\boldsymbol{x}\right)\|_2$$

where $C_\Theta \leq 1$ and, using this, derive a straightforward non-asymptotic bound for the Rademacher complexity of a very general class of neural networks (including ResNet and Transformers) that is width-independent, depth-exponential and contains no nuisance-factors. For example for a scalar-valued, layerwise, fully-connected, unbiased ReLU network of depth $D$, our bound is exactly:

$$\mathcal{R}_N\left(\mathcal{F} : \|\mathbf{W}^{[j-1:j]}\|_2 \leq \omega^{[j-1:j]}\right) \leq \frac{\prod_{j=1}^D \omega^{[j-1:j]}}{\sqrt{N}}$$

where $\|\cdot\|_2$ is the spectral norm. More generally, we derive conditions under which the Rademacher complexity bound is both *width-* and *depth-independent*, and subsequently $\mathcal{R}_N(\mathcal{F}) \leq \frac{1}{\sqrt{N}}$, and discuss implications for ReLU, ResNet and Transformer networks.

---

[1]Beyond bilinear RKBS (Lin et al., 2022), more general RKBS models have been used in eg. (Bartolucci et al., 2023; Sanders, 2020; Shilton et al., 2023; Parhi & Nowak, 2021; Unser, 2021, 2019; Spek et al., 2022).

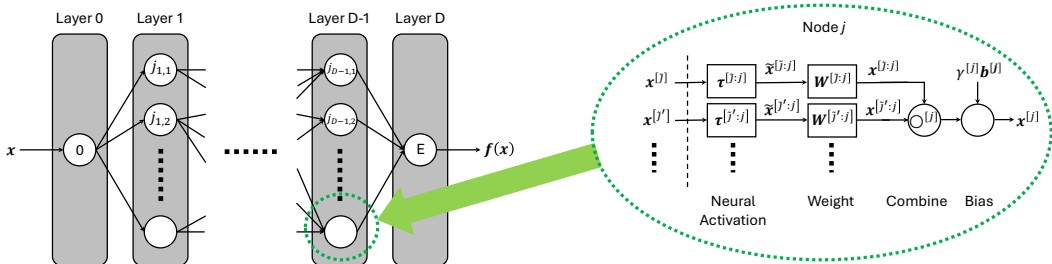

Figure 1: Layerwise feedforward neural network structure. Each layer $\jmath \in \mathbb{Z}_D$ contains nodes $\mathbb{L}^{[\jmath]}$, where the output of the node is computed as shown in the inset. Note that a computational skeleton (Daniely et al., 2016) with one input and one output can be modified to this form by inserting skip nodes (nodes with $\mathbb{A}^{[j]} = \{\jmath\}$, $\mathbf{W}^{[\tilde{\jmath}:j]} = \mathbf{I}$, $\mathbf{b}^{[j]} = \mathbf{0}$, $\boldsymbol{\tau}^{[\tilde{\jmath}:j]} = \mathrm{id}$) into the graph as required.

## 1.1 Mathematical Notations

**Vectors and matrices:** Column vectors are $\mathbf{a}, \mathbf{b}$ with elements $a_i, b_j$. Matrices are $\mathbf{A}, \mathbf{B}$ with elements $A_{i,j}$, rows $\mathbf{A}_{i:}$ and columns $\mathbf{A}_{:j}$. $|\mathbf{a}|$ and $\mathrm{sgn}(\mathbf{a})$ are the elementwise norm and sign. $\|\mathbf{A}\|_2 = \sigma_{\max}(\mathbf{A})$ is the spectral norm and $\|\mathbf{A}\|_F$ is the Frobenius norm. $\mathbf{A} \odot \mathbf{B}$, $\mathbf{A} \otimes \mathbf{B}$, $\mathbf{A} \otimes^{\ddagger} \mathbf{B}$ are Hadamard, Kronecker and columnwise Kronecker (Khatri-Rao) product. $\mathbf{A} \oplus \mathbf{B} = [\mathbf{A}^{\mathrm{T}}\ \mathbf{B}^{\mathrm{T}}]^{\mathrm{T}}$ is columnar matrix concatenation. $\mathbf{A}^{\bigcirc k} = \mathbf{A} \bigcirc^k \overset{\text{times}}{\cdots} \bigcirc \mathbf{A}$ is the exponentiation for operator $\bigcirc$.

**Products and norms:** $\langle \cdot, \cdot \rangle$, $\langle\!\langle \cdot, \cdot, \cdot, \ldots \rangle\!\rangle$ and $\langle \cdot, \cdot ]$ are inner, multilinear and bilinear products, where $\langle \mathbf{a}, \mathbf{b} \rangle = \sum_i a_i b_i$, $\langle\!\langle \mathbf{a}, \mathbf{b}, \mathbf{c}, \ldots \rangle\!\rangle = \sum_i a_i b_i c_i \ldots$, $\langle \mathbf{a}, \mathbf{b} ]_{\mathbf{g}} = \sum_i g_i a_i b_i$ and $\langle \mathbf{A}, \mathbf{b} ]_{\mathbf{g}} = \sum_i g_i \mathbf{A}_{i:} b_i$ throughout. We also find it convenient to define an operator form $\langle\!\langle \cdot \rangle\!\rangle_i \mathbf{a}_i = \langle\!\langle \mathbf{a}_1, \mathbf{a}_2, \ldots \rangle\!\rangle$.

**Sets and functions:** $\mathbb{N} = \{0, 1, \ldots\}$, $\mathbb{Z}_+ = \{1, 2, \ldots\}$, $\mathbb{Z}_N = \{1, \ldots, N\}$. $\partial \mathbb{A}$ is the boundary of $\mathbb{A}$. $\mathrm{id}(\mathbf{a}) = \mathbf{a}$. $[a]_+ = \max\{a, 0\}$, $[\mathbf{a}]_+ = [[a_i]_+]_i$. $\mathbf{a}^{\langle \odot b \rangle} = \mathrm{sgn}(\mathbf{a}) \odot |\mathbf{a}|^{\odot b}$ (Der & Lee, 2007). $L^2(\mathbb{R}^{\tilde{H}}, e^{-\|\boldsymbol{\zeta}\|_2^2}) = \{\boldsymbol{\tau} : \mathbb{R}^{\tilde{H}} \to \mathbb{R}^H \,|\, \int_{\boldsymbol{\zeta} \in \mathbb{R}^{\tilde{H}}} \|\boldsymbol{\tau}(\boldsymbol{\zeta})\|_2^2 e^{-\|\boldsymbol{\zeta}\|_2^2} d\boldsymbol{\zeta} < \infty\}$ are the finite-energy functions.

**Multi-indices:** Multi-indices are $\mathbf{k}, \mathbf{l} \in \mathbb{N}^n$ with elements $k_i, l_j$. $|\mathbf{k}| = \sum_i k_i$, $\mathbf{k}! = \prod_i k_i!$, $\mathbf{a}^{\mathbf{k}} = \prod_i a_i^{k_i}$, $\binom{\mathbf{k}}{\mathbf{l}} = \prod_i \binom{k_i}{l_i}$. $\frac{\partial^{\mathbf{k}}}{\partial \boldsymbol{x}^{\mathbf{k}}} = \prod_i \frac{\partial^{k_i}}{\partial x_i^{k_i}}$. We use the shorthands $\mathbf{k} \succ_n l$ for $\mathbf{k} \in \mathbb{N}^n$ and $|\mathbf{k}| > l$, $\mathbf{k} \succeq_n l$ for $\mathbf{k} \in \mathbb{N}^n$ and $|\mathbf{k}| \geq l$, $\mathbf{k} \prec_n l$ for $\mathbf{k} \in \mathbb{N}^n$ and $|\mathbf{k}| < l$, $\mathbf{k} \preceq_n l$ for $\mathbf{k} \in \mathbb{N}^n$ and $|\mathbf{k}| \leq l$.

**Hermite Polynomials:** $He_k(x)$ are the (probabilist's) Hermite polynomials. $He_{\mathbf{k}}(\boldsymbol{x}) = \prod_i He_{k_i}(x_i)$ are the multivariate Hermite polynomials. $\mathrm{He}_k = He_k(0)$, $\mathrm{He}_{\mathbf{k}} = He_{\mathbf{k}}(\mathbf{0})$ are the Hermite numbers (Abramowitz et al., 1972; Morse & Feshbach, 1953; Olver et al., 2010; Rahman, 2017).

**Indexing Conventions:** Layers are $\jmath \in \mathbb{Z}_D$ (there are $D$ layers). Nodes are $j \in \mathbb{Z}_E$ (there are $E$ nodes). Layer $\jmath$ contains nodes $\mathbb{L}^{[\jmath]} \subseteq \mathbb{Z}_E$: $\cup_{\jmath \in \mathbb{Z}_D} \mathbb{L}^{[\jmath]} = \mathbb{Z}_E$, $\mathbb{L}^{[\jmath]} \cap \mathbb{L}^{[\jmath']} = \emptyset \,\forall \jmath \neq \jmath'$. Node $j \in \mathbb{L}^{[\jmath]}$ in layer $\jmath$ has parents $\tilde{\jmath} \in \mathbb{A}^{[j]} \subseteq \mathbb{L}^{[\jmath-1]}$. $\mathbb{L}^{[0]} = \{0\}$, $\mathbb{L}^{[D]} = \{E\}$ are the input/output layers.

## 2 Setting and Assumptions

We consider layerwise feedforward neural networks as shown in Figure 1. This contains $E$ nodes $j \in \mathbb{Z}_E$ arranged in $D$ layers $\jmath \in \mathbb{Z}_D$ and a virtual input node $j = 0$ (in virtual layer $\jmath = 0$), where layer $\jmath \in \mathbb{Z}_D$ contains nodes $\mathbb{L}^{[\jmath]} \subseteq \mathbb{Z}_E$ and layer $D$ contains a single output node $E$. A node $j \in \mathbb{L}^{[\jmath]}$ has parents $\mathbb{A}^{[j]} \subseteq \mathbb{L}^{[\jmath-1]}$, with its function being specified by an operator $\bigcirc^{[j]} \in \{\oplus, \sum, \otimes, \langle\!\langle \cdot \rangle\!\rangle\}$. Given input $\boldsymbol{x}$, data flows from node $j = 0$ to node $j = E$ as per Figure 1:

$$
\begin{rcases}
\boldsymbol{x}^{[0]} = \boldsymbol{x} \\
\downarrow \\
\left.\begin{array}{l}
\boldsymbol{x}^{[j]} = \bigcirc^{[j]}_{\tilde{\jmath} \in \mathbb{A}^{[j]}} \mathbf{W}^{[\tilde{\jmath}:j]\mathrm{T}} \boldsymbol{x}^{[\tilde{\jmath}:j]} + \gamma^{[j]} \mathbf{b}^{[j]} \in \mathbb{R}^{H^{[j]}} \\
\boldsymbol{x}^{[\tilde{\jmath}:j]} = \boldsymbol{\tau}^{[\tilde{\jmath}:j]}(\boldsymbol{x}^{[\tilde{\jmath}]}) \in \mathbb{R}^{H^{[\tilde{\jmath}:j]}} \quad \forall \tilde{\jmath} \in \mathbb{A}^{[j]}
\end{array}\right] \forall j \in \mathbb{Z}_E \quad \text{(a)} \\
\downarrow \\
\mathbf{f}(\boldsymbol{x}; \Theta) = \boldsymbol{x}^{[E]}
\end{rcases}
\tag{1}
$$

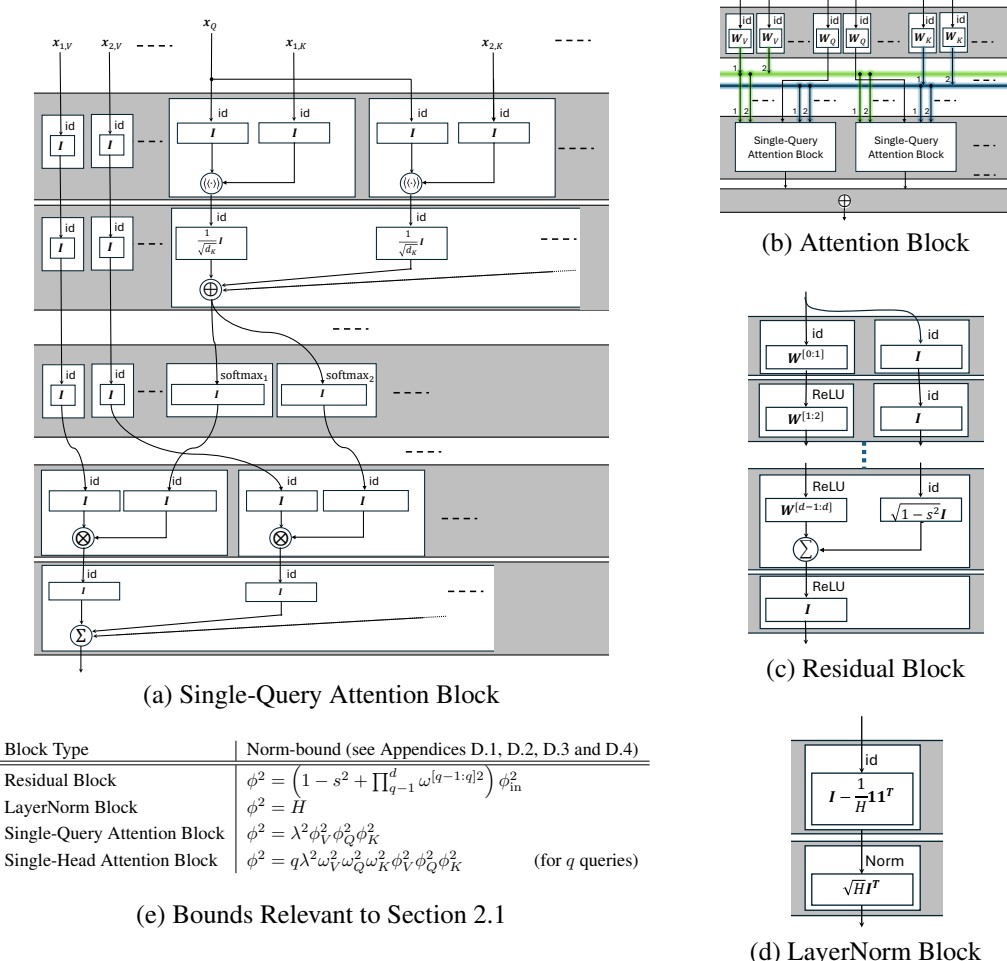

(a) Single-Query Attention Block

(b) Attention Block

(c) Residual Block

(d) LayerNorm Block

| Block Type | Norm-bound (see Appendices D.1, D.2, D.3 and D.4) |
|---|---|
| Residual Block | $\phi^2 = \left(1 - s^2 + \prod_{q-1}^d \omega^{[q-1:q]2}\right)\phi_{\text{in}}^2$ |
| LayerNorm Block | $\phi^2 = H$ |
| Single-Query Attention Block | $\phi^2 = \lambda^2 \phi_V^2 \phi_Q^2 \phi_K^2$ |
| Single-Head Attention Block | $\phi^2 = q\lambda^2 \omega_V^2 \omega_Q^2 \omega_K^2 \phi_V^2 \phi_Q^2 \phi_K^2$    (for $q$ queries) |

(e) Bounds Relevant to Section 2.1

Figure 2: Residual, attention and LayerNorm blocks. In the residual block $s \in (0, 1)$. The single-query attention block (a) is for a single query $\boldsymbol{x}_{1,Q}$ with keys $\boldsymbol{x}_{1,K}, \boldsymbol{x}_{2,K}, \dots$ and values $\boldsymbol{x}_{1,V}, \boldsymbol{x}_{2,V}$, .... A single-head attention block (b) is formed from multiple single-query blocks (the usual (matrix) output has been vectorized here). See Table 3 for definitions of neural activations used here.

where $\boldsymbol{\tau}^{[\widetilde{\jmath}:j]} : \mathbb{R}^{H^{[\widetilde{\jmath}]}} \to \mathbb{R}^{H^{[\widetilde{\jmath}:j]}}$ are neural activation functions; $\mathbf{W}^{[\widetilde{\jmath}:j]}$ are weight matrices; $\mathbf{b}^{[j]}$ biases; $\gamma^{[j]} \in \{0, 1\}$ (unbiased and biased); and $\Theta = \{\mathbf{W}^{[\widetilde{\jmath}:j]}, \mathbf{b}^{[j]} : j \in \mathbb{Z}_E, \widetilde{\jmath} \in \mathbb{A}^{[j]}\}$. We assume that:

1. **Bounded inputs:** $\boldsymbol{x} \in \mathbb{X} = \{\boldsymbol{x} \in \mathbb{R}^n : \|\boldsymbol{x}\|_2 \le 1\}$.

2. **Finite weights/biases:** $\Theta \in \mathbb{W} = \{\mathbf{W}^{[\widetilde{\jmath}:j]}, \mathbf{b}^{[j]} : \|\mathbf{W}^{[\widetilde{\jmath}:j]}\|_2, \|\mathbf{b}^{[j]}\|_2 < \infty : \widetilde{\jmath} \in \mathbb{A}^{[j]}, j \in \mathbb{Z}_E\}$.

3. **Finite activations:** $\boldsymbol{\tau}^{[\widetilde{\jmath}:j]} \in L^2(\mathbb{R}^{H^{[\widetilde{\jmath}]}}, e^{-\|\boldsymbol{\varsigma}\|_2^2}) \,\forall \widetilde{\jmath} \in \mathbb{A}^{[j]}, j \in \mathbb{Z}_E$.

$\Big($4. **Lipschitz/Bounded activations:** $\boldsymbol{\tau}^{[\widetilde{\jmath}:j]}$ is either Lipschitz or bounded $\forall \widetilde{\jmath} \in \mathbb{A}^{[j]}, j \in \mathbb{Z}_E$, and $\boldsymbol{x} \in \partial\mathbb{X} = \{\boldsymbol{x} \in \mathbb{R}^n : \|\boldsymbol{x}\|_2 = 1\}$ if any $\boldsymbol{\tau}^{[\widetilde{\jmath}:j]}$ are non-Lipschitz.$\Big)$

Note that assumption 4 is not required when constructing our bilinear feature space model of neural networks, but is required to cast this model in RKBS and subsequently derive our Rademacher complexity bound. The set of neural networks satisfying our assumptions is $\mathcal{F}$, and its dual is $\mathcal{F}^\star$:

$$\begin{aligned} \mathcal{F} &= \{\mathbf{f}(\cdot; \Theta) : \mathbb{X} \to \mathbb{R}^m \,|\, \mathbf{f} \text{ as per (1) satisfying assumptions 1-4}\} \\ \mathcal{F}^\star &= \{\mathbf{f}(\boldsymbol{x}; \cdot) : \mathbb{W} \to \mathbb{R}^m \,|\, \mathbf{f} \text{ as per (1) satisfying assumptions 1-4}\} \end{aligned} \tag{2}$$

This model is rather general to encompass a wider variety of network architectures. Residual (He et al., 2016) blocks can be built using additive nodes $\bigcirc^{[j]} = \sum$ as shown in Figure 2c. Single

| Neural Activation | | $\tau \in L^2(\mathbb{R}^H, e^{-\|\zeta\|_2^2})$ | Lipschitz ($L_r$) | Bounded ($B$) | Valid here |
|---|---|---|---|---|---|
| Linear | $\tau^{[\tilde{j};j]}(\zeta) = \zeta$ | ✓ | ✓ (1) | × | ✓ |
| ReLU | $\tau^{[\tilde{j};j]}(\zeta) = [\zeta]_+$ | ✓ | ✓ (1) | × | ✓ |
| Poly-ReLU | $\tau^{[\tilde{j};j]}(\zeta) = [\zeta]_+^{\odot p}$ | ✓ | ✓ ($pa^{p-1}$) | × | ✓ |
| Tanh | $\tau^{[\tilde{j};j]}(\zeta) = [\tanh(\zeta_i)]_i$ | ✓ | ✓ (1) | - | ✓ |
| Sigmoid | $\tau^{[\tilde{j};j]}(\zeta) = \left[\frac{1}{1+e^{-\zeta_i}}\right]$ | ✓ | ✓ ($\frac{1}{2}$) | - | ✓ |
| Softmax | $\tau^{[\tilde{j};j]}(\zeta) = \left[\frac{e^{\lambda\zeta_{i'}}}{\sum_{i''} e^{\lambda\zeta_{i''}}}\right]_{i'}$ | ✓ | ✓ ($\lambda$) | - | ✓ |
| Softmax$_i$ | $\tau^{[\tilde{j};j]}(\zeta) = \left[\delta_{i,i'}\frac{e^{\lambda\zeta_{i'}}}{\sum_{i''} e^{\lambda\zeta_{i''}}}\right]_{i'}$ | ✓ | ✓ ($\lambda$) | - | ✓ |
| Norm | $\tau^{[\tilde{j};j]}(\zeta) = \frac{\zeta}{\|\zeta\|_2}$ | ✓ | × | ✓ (1) | ✓ |

Figure 3: Characteristics of common neural activation functions. We include poly-ReLU (Cho & Saul, 2009) here as an example where the Lipschitz constant $L_a$ of $\tau|_a$ depends on the radius $a$. See (Gao & Pavel, 2017) for more detail regarding the softmax Lipschitz constant.

query attention blocks (Vaswani et al., 2017) can be constructed as shown in Figure 2a using not just additive but also inner-product $\bigcirc^{[j]} = \langle\!\langle \cdot \rangle\!\rangle$, multiplicative $\bigcirc^{[j]} = \otimes$ and columnar concatenation $\bigcirc^{[j]} = \oplus$ nodes. Full attention block can be constructed as shown in Figure 2b (and similarly multi-head attention). Finally, a LayerNorm (layer normalization (Ba et al., 2016)) block is shown in Figure 2d. We note that blocks of this sort may be combined to form more general networks. Later, we find it convenient to include non-trivial nodes or blocks in the network, so for example we may speak of an "attention node" $j$ that encompasses (abstracts away) a complete attention block (Figure 2b).

## 2.1 Characterization of Neural Activations

As noted previously, we assume all activation functions in the network are Lipschitz/bounded and finite energy. The finite-energy assumption allows us to apply the Hermite transform to the neural activation functions and subsequently construct our bilinear model of the network, while the Lipschitz/bounded property suffices to ensure that the bilinear model is continuous. Starting with the finite-energy assumption, the multivariate (probabilist's) Hermite polynomials (Abramowitz et al., 1972; Morse & Feshbach, 1953; Olver et al., 2010; Rahman, 2017) are, for multi-index $\mathbf{k} \in \mathbb{N}^n$:

$$He_{\mathbf{k}}(\zeta) = (-1)^{|\mathbf{k}|} e^{\frac{1}{2}\|\zeta\|_2^2} \frac{\partial^{\mathbf{k}}}{\partial\zeta^{\mathbf{k}}} e^{-\frac{1}{2}\|\zeta\|_2^2} = \sum_{0 \preceq_n \mathbf{l} \leq \mathbf{k}} \binom{\mathbf{k}}{\mathbf{l}} He_{\mathbf{k}-\mathbf{l}} \zeta^{\mathbf{l}}$$

where $He_{\mathbf{k}} = He_{\mathbf{k}}(\mathbf{0})$ are Hermite numbers. These form an orthogonal basis of $L^2(\mathbb{R}^n, e^{-\|\zeta\|_2^2})$ (Appendix A). By assumption $\tau^{[\tilde{j};j]} \in L^2(\mathbb{R}^{H^{[\tilde{j}]}}, e^{-\|\zeta\|_2^2})$, and thus the Hermite transform exists:[2]

$$\tau^{[\tilde{j};j]}(\zeta) = \tau^{[\tilde{j};j]}(\mathbf{0}) + \sum_{\mathbf{k}\succ_{H^{[\tilde{j}]}}0} \mathbf{a}_{\mathbf{k}}^{[\tilde{j};j]} \sum_{0 \prec_{H^{[\tilde{j}]}} \mathbf{l} \leq \mathbf{k}} \binom{\mathbf{k}}{\mathbf{l}} He_{\mathbf{k}-\mathbf{l}} \zeta^{\mathbf{l}}$$

$$\text{where:} \quad \mathbf{a}_{\mathbf{k}}^{[\tilde{j};j]} = \frac{1}{\sqrt{2\pi}\mathbf{k}!} \int_{\zeta \in \mathbb{R}^{H^{[\tilde{j}]}}} \left(\tau^{[\tilde{j};j]}(\zeta) - \tau^{[\tilde{j};j]}(\mathbf{0})\right) He_{\mathbf{k}}(\zeta) e^{-\frac{1}{2}\|\zeta\|_2^2} d\zeta$$

(3)

From this we define the magnitude functions $s_\eta^{[\tilde{j};j]} : \mathbb{R}_+ \to \mathbb{R}_+$ (where $\eta \in \mathbb{R}_+$):

$$s_\eta^{[\tilde{j};j]}(\zeta) = \eta^2 \left\|\tau^{[\tilde{j};j]}(\mathbf{0})\right\|_2^2 + \sum_{\mathbf{k}\succ_{H^{[\tilde{j}]}}0} \left\|\mathbf{a}_{\mathbf{k}}^{[\tilde{j};j]}\right\|_1 \left((1+\eta\zeta)^{|\mathbf{k}|} - 1\right)$$

(4)

which are monotonically increasing and superadditive on $\mathbb{R}_+$. Note that while the Hermite transform terms and magnitude functions play an important role in the construction of our model, they play no role in our subsequent analysis of Rademacher complexity (they vanish in our analysis in the limit $\eta \to 0^+$). Thus, for our purposes, beyond their existence (which is guaranteed), their exact value/form does not matter here. Nevertheless, see Appendix B for a full analysis of the ReLU activation.

Regarding assumptions 4, if $\tau^{[\tilde{j};j]}|_a$ ($\tau^{[\tilde{j};j]}$ restricted to a ball of radius $a$) is Lipschitz then we denote the Lipschitz constant by $L_a^{[\tilde{j};j]}$. Conversely, if $\tau^{[\tilde{j};j]}$ is absolutely bounded, we denote the bound $B^{[\tilde{j};j]}$, where $|\tau^{[\tilde{j};j]}(\zeta)| \leq B^{[\tilde{j};j]} \ \forall \zeta$. While assumption 4 is not required to construct our bilinear dual representation we find it useful to include $L_a^{[\tilde{j};j]}$ here to simplify later results. When $L_a^{[\tilde{j};j]}$ is ill-defined we use the nominal value $L_a^{[\tilde{j};j]} = B^{[\tilde{j};j]}/\phi^{[\tilde{j}]}$ in the bounded case, or $L_a^{[\tilde{j};j]} = 1$ if assumption 4 is not satisfied. Table 3 provides Lipschitz constants/bounds for common neural activations.

---

[2]These are conditionally convergent series in general, so ordering of multi-indices $\mathbf{k}, \mathbf{l}$ in sums and vectors must be enforced consistently and must be compatible with the semi-ordering imposed by $\preceq_{H^{[\tilde{j}]}}$.

Figure 4: Recursive definition of the bilinear representation $\mathbf{f}(\boldsymbol{x};\Theta) = \langle \boldsymbol{\Psi}(\Theta), \boldsymbol{\phi}(\boldsymbol{x})]_{\mathbf{g}}$. The upper figure is a schematic representation of the formal definition (5), where the bilinear representation of the output of each node is obtained, using (5a), from the bilinear representations of the inputs $\widetilde{\jmath} \in \mathbb{A}^{[j]}$ to that node. Subsequently the bilinear representation of the network is defined recursively in terms of the (trivial) bilinear representation of $\boldsymbol{x} = \langle \mathbf{I}, \boldsymbol{x} ]_{\mathbf{1}}$. $\eta \in \mathbb{R}_+$ is an (arbitrary) constant that will be helpful in our Rademacher complexity analysis. With regard notation, we recall that node $j$ is characterized by its operation $\bigcirc^{[j]} \in \{\oplus, \sum, \otimes, \langle\!\langle \cdot \rangle\!\rangle\}$, and subsequently the form of the feature-map recursion (5a) depends on this operation as specified by the operators $\square^{[j]}$ (weight map operator), $\boxdot^{[j]}$ (data map/metric operator) and $\odot^{[j]}$ (norm bound operator) defined. For non-Lipschitz neural activations we set $L_a^{[\widetilde{\jmath};j]} = B^{[\widetilde{\jmath};j]}/\phi^{[\widetilde{\jmath}]}$ in the bounded case and $L_a^{[\widetilde{\jmath};j]} = 1$ if assumption 4 is not satisfied.

# 3 Neural Networks in Reproducing Kernel Banach Space

As noted in our introduction, a recurring theme in the machine learning (most famously kernel methods) is the use of bilinear (dual) representations to cleanly separate data and model parameters, ie:

$$\mathbf{f}(\boldsymbol{x};\Theta) = \langle \boldsymbol{\Psi}(\Theta), \boldsymbol{\phi}(\boldsymbol{x})]$$

Here the set of network parameters $\Theta$, and the data $\boldsymbol{x}$, are mapped entirely independently into distinct feature spaces by, respectively, $\boldsymbol{\Psi} : \mathbb{W} \to \mathcal{W}$ (weights and biases) and $\boldsymbol{\phi} : \mathbb{X} \to \mathcal{X}$ (data). The bilinear product $\langle \cdot, \cdot ] : \mathcal{W} \times \mathcal{X} \to \mathbb{R}^m$ generalizes the inner product of eg SVMs (Cortes & Vapnik, 1995; Burges, 1998; Cristianini & Shawe-Taylor, 2005; Steinwart & Christman, 2008) without losing the very useful property of bilinearity that makes this formalism so convenient to work with. Apart from the potential for constructing a representor theory (kernelization), if the bilinear product is continuous (ie. if $\exists C, C' \in \mathbb{R}_+$ so that $\langle \boldsymbol{\Psi}, \boldsymbol{\phi} ] \leq C \|\boldsymbol{\phi}\| \forall \boldsymbol{\Psi}$ or $\langle \boldsymbol{\Psi}, \boldsymbol{\phi} ] \leq C' \|\boldsymbol{\Psi}\| \forall \boldsymbol{\phi}$) then the existence of such a model significantly simplifies the development of Rademacher complexity bounds. A model of this type was developed in (Shilton et al., 2023) using a recursive Taylor series expansion of the neural activations - in brief, noting that $\langle \mathbf{a}, \mathbf{b} ]_{\mathbf{g}}^n = \langle \mathbf{a}^{\otimes n}, \mathbf{b}^{\otimes n} ]_{\mathbf{g}^{\otimes n}}$, if the input to a neuron can be represented bilinearly then so too could the output, which recursion defines the model. Unfortunately this approach only works for continuous neural activations, and even then only within the RoC of the Taylor expansion, rendering it inapplicable for common activations such as ReLU. Alternatively, in this paper we propose using a Hermite polynomial expansion, which has two benefits, precisely (1) the Hermite polynomial expansion exist for all finite-energy activations and is convergent everywhere (applicability), and (2) as the Hermite polynomials are constructed from monomials we can also use $\langle \mathbf{a}, \mathbf{b} ]_{\mathbf{g}}^n = \langle \mathbf{a}^{\otimes n}, \mathbf{b}^{\otimes n} ]_{\mathbf{g}^{\otimes n}}$ to construct our model (practicality).

We begin by constructing our dual representation:

**Theorem 1.** *Let* $\mathbf{f} : \mathbb{X} \times \mathbb{W} \to \mathbb{R}^m$ *be a neural network (1) satisfying assumptions 1- 3. Assume nominal bounds* $\|\mathbf{W}^{[\tilde{\jmath}:j]}\|_2 \leq \omega^{[\tilde{\jmath}:j]} < \infty$ *and* $\|\mathbf{b}^{[j]}\|_2 \leq \beta^{[j]} < \infty \; \forall j \in \mathbb{Z}_E, \tilde{\jmath} \in \mathbb{A}^{[j]}$. *Let* $\eta \in \mathbb{R}_+$. *Defining feature maps* $\boldsymbol{\Psi} : \mathbb{W} \to \mathcal{W} \subset \mathbb{R}^{\infty \times m}$ *(weights and biases) and* $\boldsymbol{\phi} : \mathbb{X} \to \mathcal{X} \subset \mathbb{R}^\infty$ *(data) and metric* $\mathbf{g} \in \mathbb{R}^\infty$ *as per (5) (Figure 4), the network may be written in bilinear form:*

$$\mathbf{f}(\boldsymbol{x};\Theta) = \langle \boldsymbol{\Psi}(\Theta), \boldsymbol{\phi}(\boldsymbol{x}) ]_{\mathbf{g}} \tag{6}$$

*where* $\|\boldsymbol{\Psi}(\Theta)\|_F \leq \psi_\eta < \infty \; \forall \Theta \in \mathbb{W}$, $\|\boldsymbol{\phi}(\boldsymbol{x})\|_2 \leq \phi_\eta < \infty \; \forall \boldsymbol{x} \in \mathbb{X}$, *(the constants* $\psi_\eta, \phi_\eta$ *are provided in Appendix C.1), where* $\lim_{\eta \to 0^+} \psi_\eta = \psi$ *and* $\lim_{\eta \to 0^+} \phi_\eta = \phi$; *and we note that* $\lim_{\eta \to 0^+} \|\boldsymbol{\phi}(\boldsymbol{x})\|_2 = \phi \; \forall \boldsymbol{x} \in \partial \mathbb{X}$ *(ie. if* $\|\boldsymbol{x}\|_2 = 1$*), and* $\lim_{\eta \to 0^+} \|\boldsymbol{\phi}(\boldsymbol{x})\|_2 > 0 \; \forall \boldsymbol{x} \neq \mathbf{0}$.

A full inductive proof can be found in Appendix C.1. To summarize, picking a layer $\jmath \in \mathbb{Z}_D$, we assume all nodes $\tilde{\jmath} \in \mathbb{L}^{[\jmath-1]}$ in the preceding layer may be written $\boldsymbol{x}^{[\tilde{\jmath}]} = \langle \boldsymbol{\Psi}^{[\tilde{\jmath}]}(\Theta), \boldsymbol{\phi}^{[\tilde{\jmath}]}(\boldsymbol{x}) ]_{\mathbf{g}^{[\tilde{\jmath}]}}$, which is trivial for the base case $\jmath = 0$. Then, using $(\mathbf{A}^{\mathrm{T}}\mathbf{b})^{\odot p} = (\mathbf{A}^{\otimes^{\ddagger} p})^{\mathrm{T}}(\mathbf{b}^{\otimes p})$ in combination with the Hermite (number) expansion of the neural activation function, we write the incoming edge activations $\boldsymbol{x}^{[\tilde{\jmath}:j]}$ as bilinear products $\boldsymbol{x}^{[\tilde{\jmath}:j]} = \langle \boldsymbol{\Psi}^{[\tilde{\jmath}:j]}(\Theta), \boldsymbol{\phi}^{[\tilde{\jmath}:j]}(\boldsymbol{x}) ]_{\mathbf{g}^{[\tilde{\jmath}:j]}}$ (see Appendix for full definitions). This, combined with the observation that $\bigcirc_{\tilde{\jmath} \in \mathbb{A}^{[j]}}^{[j]} \mathbf{W}^{[\tilde{\jmath}:j]\mathrm{T}} \boldsymbol{x}^{[\tilde{\jmath}:j]} = (\square_{\tilde{\jmath} \in \mathbb{A}^{[j]}}^{[j]} \mathbf{W}^{[\tilde{\jmath}:j]})^{\mathrm{T}} (\boxdot_{\tilde{\jmath} \in \mathbb{A}^{[j]}}^{[j]} \mathbf{x}^{[\tilde{\jmath}:j]})$, suffices to show that $\boldsymbol{x}^{[j]} = \langle \boldsymbol{\Psi}^{[j]}(\Theta), \boldsymbol{\phi}^{[j]}(\boldsymbol{x}) ]_{\mathbf{g}^{[j]}}$ as given, and the result follows by induction.

As alluded to in section 2, we can readily incorporate non-trivial nodes into this framework. In the recursive construction of the feature maps, (5a) is effectively a recipe for converting the bilinear expansion of the inputs to that node to a bilinear expansion of the node's output. As stated, (5a) is for a *trivial* node of the type shown in Figure 1, but alternatively we could wrap an entire sub-network or block *inside* this node (eg. an attention block - Figure 2b) and replace (5a) with the overall recipe for converting bilinear expansions of its input to a bilinear expansion of its output. Thus we may reasonably speak of an "attention node" in a Transformer network without needless clutter. For example Figure 2d includes a table detailing calculations for $\phi$ for attention, residual and LayerNorm blocks (nodes) (derivations for these can be found in Appendix D.1, D.2, D.3 and D.4).

Unfortunately the dual representation (6) is insufficient for Rademacher complexity analysis without assumption 4, which requires that the neural activations be Lipschitz or bounded (and in the latter case that $\|\boldsymbol{x}\|_2 = 1$). This assumption is central to casting the dual model (6) into RKBS, precisely:

**Definition 1** (Reproducing kernel Banach space (RKBS)). A RKBS on $\mathbb{X}$ is a Banach space $\mathcal{B}$ of functions $f : \mathbb{X} \to \mathbb{Y}$, where $\mathbb{Y}$ is normed, for which the point evaluation functionals $\boldsymbol{\delta}_{\boldsymbol{x}}(f) = f(\boldsymbol{x})$ on $\mathcal{B}$ are continuous (i.e. $\forall \boldsymbol{x} \in \mathbb{X} \; \exists C_{\boldsymbol{x}} \in \mathbb{R}_+$ such that $\|\boldsymbol{\delta}_{\boldsymbol{x}}(f)\| \leq C_{\boldsymbol{x}} \|f\|_{\mathcal{B}} \; \forall f \in \mathcal{B}$).

This is somewhat generic, so following (Lin et al., 2022) we focus on the special case:

$$\mathcal{B} = \left\{ f\left(\cdot;\Theta\right) = \left\langle \boldsymbol{\Psi}\left(\Theta\right), \boldsymbol{\phi}\left(\cdot\right)\right]_{\mathcal{W}\times\mathcal{X}} \middle| \Theta \in \mathbb{W} \right\} \tag{7}$$

where $\boldsymbol{\phi} : \mathbb{X} \to \mathcal{X}$ is a data feature map, $\boldsymbol{\Psi} : \mathbb{W} \to \mathcal{W}$ is a weight feature map, $\mathcal{X}$ and $\mathcal{W}$ are Banach spaces, and $\langle \cdot, \cdot]_{\mathcal{W}\times\mathcal{X}} : \mathcal{W} \times \mathcal{X} \to \mathbb{R}^m$ is a continuous bilinear form. Given this prequel we have:

**Corollary 2.** *The set $\mathcal{F}$ of networks (2) satisfying assumptions 1-4 with* Lipschitz *neural activations and weights and biases bounded as per Theorem 1 is an RKBS with* $\|\mathbf{f}(\cdot;\Theta)\|_{\mathcal{F}} \triangleq \lim_{\eta\to 0^+} \|\boldsymbol{\Psi}(\Theta)\|_F \leq \psi < \infty$ *and* $\|\mathbf{f}(\boldsymbol{x};\Theta)\|_2 \leq C_{\boldsymbol{x}}\|\mathbf{f}(\cdot;\Theta)\|_{\mathcal{F}}$, *where* $C_{\boldsymbol{x}} = 1 \ \forall \boldsymbol{x} \in \mathbb{X}$.

**Corollary 3.** *The set $\mathcal{F}^\star$ of networks (2) satisfying assumptions 1-4 with* Lipschitz *or bounded neural activations and with weights and biases bounded as per Theorem 1 is an RKBS with* $\|\mathbf{f}(\boldsymbol{x};\cdot)\|_{\mathcal{F}^\star} \triangleq \lim_{\eta\to 0^+} \|\boldsymbol{\phi}(\boldsymbol{x})\|_2 \leq \phi < \infty$ *and* $\|\mathbf{f}(\boldsymbol{x};\Theta)\|_2 \leq C_{\Theta}\|\mathbf{f}(\boldsymbol{x};\cdot)\|_{\mathcal{F}^\star}$, *where* $C_{\Theta} = 1 \ \forall \Theta \in \mathbb{W}$.

See Appendix C.3 for proofs (the structure of which minics that of the proof of Theorem 1). It follows from this that the model presented in Theorem 1 suffices to achieve our primary goal. Note that this result applies to a very wide range of networks, including feedforward ReLU networks, convolutional networks, residual networks (ResNet), and Transformer networks (see later discussion). We observe that the conditions for $\mathcal{F}$ to be an RKBS are stricter than the conditions for $\mathcal{F}^\star$ to be an RKBS, as non-Lipschitz neural activations appears incompatible with $\mathcal{F}$ being an RKBS. However as we will see that we only require $\mathcal{F}^\star$ be an RKBS to proceed with our Rademacher complexity analysis.

## 4 Rademacher Complexity Bounds

We now address our secondary goal, namely using our dual model to bound the Rademacher complexity of neural networks. For $h : \mathbb{R}^m \to \mathbb{R}$, the Rademacher complexity is defined as:

$$\mathcal{R}_N\left(h \circ \mathcal{F}\right) = \mathbb{E}_\nu \mathbb{E}_\epsilon \left[ \sup_{\mathbf{f}\in\mathcal{F}} \frac{1}{N} \sum_{i=1}^N \epsilon_i h\left(\mathbf{f}\left(\boldsymbol{x}_i\right)\right) \right]$$

for Rademacher random variables $\epsilon_i \in \{\pm 1\}$, where $\boldsymbol{x} \sim \nu$. We have:

**Theorem 4.** *Let $\mathcal{F}$ be the set of networks (2) satisfying assumptions 1-4 with weights and biases bounded as per Theorem 1, and let $h : \mathbb{R}^m \to \mathbb{R}$ be L-Lipschitz. Then:*

$$\mathcal{R}_N\left(h \circ \mathcal{F} : \left\|\mathbf{W}^{[\tilde{\jmath}:j]}\right\|_2 \leq \omega^{[\tilde{\jmath}:j]}, \left\|\mathbf{b}^{[j]}\right\|_2 \leq \beta^{[j]}\right) \leq \frac{H_m \phi}{\sqrt{N}} \tag{8}$$

*where $H_1 = 1$ if $h = \mathrm{id}$, $H_m = \sqrt{2m}L$ otherwise, and $\phi$ is defined in Figure 4.*

The proof follows the usual template for RKHS models (see eg. (Bartlett & Mendelson, 2002)) using our feature map; replacing the Cauchy-Schwarz inequality with $\|\mathbf{f}(\boldsymbol{x};\Theta)\|_2 \leq C_{\Theta}\|\boldsymbol{\phi}(\boldsymbol{x})\|_2$; taking the limit $\eta \to 0^+$; and recalling that $C_{\Theta} = 1$ and $\lim_{\eta\to 0^+} \|\boldsymbol{\phi}(\boldsymbol{x})\|_2 \leq \phi$, so $\|\mathbf{f}(\boldsymbol{x};\Theta)\|_2 \leq C_{\Theta}\|\boldsymbol{\phi}(\boldsymbol{x})\|_2 \leq \phi$. See Appendix F for full details.

Considering this Rademacher complexity bound, we recall that typically neural network weights and biases are initialized with magnitude proportional to $\frac{1}{\sqrt{H^{[j]}}}$ (LeCun initialization) or $\frac{1}{\sqrt{H^{[\tilde{\jmath}:j]}}}$ (He initialization), and stay close to their initial values in the wide limit, assuming a convex objective. Thus we would expect that $\|\mathbf{W}^{[\tilde{\jmath}:j]}\|_2$ (and hence its upper bound $\omega^{[\tilde{\jmath}:j]}$) should be *independent* of network width, rendering the complexity bound in Theorem 4 (effectively) width-independent. We also observe that the complexity bound does not contain any *explicitly* depth-dependent terms (nuisance terms that are often present in such bounds as discussed in (Golowich et al., 2018)); however the bound will in general grow exponentially with depth due to the multiplicative build-up of terms in $\phi$ from input to output, which is typical of such results (Neyshabur et al., 2015; Golowich et al., 2018; Truong, 2022). For a scalar-valued, unbiased, Lipschitz network with 1 node $j = \jmath$ per layer, (8) becomes:[3]

$$\mathcal{R}_N\left(\mathcal{F} : \left\|\mathbf{W}^{[j-1:j]}\right\|_2 \leq \omega^{[j-1:j]}\right) \leq \frac{\prod_{j=1}^D L^{[j-1:j]}\omega^{[j-1:j]}}{\sqrt{N}} \tag{9}$$

While this bound is depth-exponential in general, we can use to to derive conditions (on the weights) under which this exponentiality can be (in effect) neutralised. Motivated by this, the following result gives general, non-trivial threshold conditions for depth-independent Rademacher complexity:

---

[3]Conversely, we know that $\mathcal{R}_N(\mathcal{F} : \|\mathbf{W}^{[j-1:j]}\|_2 \leq \omega^{[j-1:j]}) \sim \Omega(\frac{1}{\sqrt{N}} \prod_{j=1}^D \omega^{[j-1:j]})$ (Golowich et al., 2018, Theorem 7), and hence $\mathcal{R}_N(\mathcal{F} : \left\|\mathbf{W}^{[j-1:j]}\right\|_2 \leq \omega^{[j-1:j]}) \asymp \frac{\prod_{j=1}^D \omega^{[j-1:j]}}{\sqrt{N}}$.

| Node or Block Type | Depth-Independence Condition | Notes |
|---|---|---|
| Trivial | $\left\|\mathbf{b}^{[j]}\right\|_2^2 + \circledcirc^{[j]}_{\widetilde{\jmath}\in\mathbb{A}^{[j]}} L_1^{[\widetilde{\jmath}:j]2}\left\|\mathbf{W}^{[\widetilde{\jmath}:j]}\right\|_2^2 \le 1$ | See Figure 1 and equation (10). |
| Residual | $\prod_{q\in\mathbb{Z}_{d_j}}\left\|\mathbf{W}^{[\widetilde{\jmath}:j]q}\right\|_2 \le s$ | See Figure 2c. In this bound we denote the weight matrix for (internal) layer $q$ as $\mathbf{W}^{[\widetilde{\jmath}:j]q}$. See Appendix D.1 for the complete derivation. |
| Single-Query Attention | $\lambda\left\|\mathbf{W}_V\right\|_2\left\|\mathbf{W}_Q\right\|_2\left\|\mathbf{W}_K\right\|_2 \le 1$ | See Figure 2a. In this bound $\lambda$ is the heat parameter for the softmax. See Appendix D.3 for a complete derivation. |
| Single- and Multi-Head Attention | $\lambda\sqrt{d_{\text{model}}}\left\|\mathbf{W}_V\right\|_2\left\|\mathbf{W}_Q\right\|_2\left\|\mathbf{W}_K\right\|_2 \le 1$ | See Figure 2b. In this bound $\lambda$ is the heat parameter for the softmax. Here $d_{\text{model}}$ is the product of the number of queries and the number of heads. See Appendix D.4 for a complete derivation. |

Figure 5: Conditions for Depth Independent Rademacher Complexity Bounds for Typical Nodes.

**Corollary 5.** *Let $\mathcal{F}$ be the set of networks (2) satisfying our assumptions with weights and biases bounded as per Theorem 1, and let $h : \mathbb{R}^m \to \mathbb{R}$ be $L$-Lipschitz. If:*

$$\left\|\boldsymbol{\phi}^{[\widetilde{\jmath}]}(\boldsymbol{x})\right\|_2 \le \phi^{[\widetilde{\jmath}]} = 1\forall\widetilde{\jmath}\in\mathbb{A}^{[j]} \implies \left\|\boldsymbol{\phi}^{[j]}(\boldsymbol{x})\right\|_2 \le \phi^{[j]} = 1 \tag{10}$$

*for all nodes $j \in \mathbb{Z}_E$, then $\mathcal{R}_N(h\circ\mathcal{F}) \le \frac{H_m}{\sqrt{N}}$, independent of both width and depth.*

This follows from the recursive definition of $\phi$ in (5) (Figure 4) as a sufficient condition to ensure that $\phi^{[j]} = 1$ given $\phi^{[\widetilde{\jmath}]} = 1$ for all nodes $j \in \mathbb{Z}_E$, $\widetilde{\jmath} \in \mathbb{A}^{[j]}$, and subsequently (recursively) $\phi = \phi^{[E]} = 1$. In practice the interpretation of this result is node specific. Conditions for various nodes (in the Lipschitz case) can be found in Table 5, where derivations may be found in the appendices noted. The general, non-Lipschitz (bounded) case is somewhat more complicated. Recall that if there are non-Lipschitz neural activations in the network we assume that $\boldsymbol{x} \in \partial\mathbb{X}$ or, equivalently, $\|\boldsymbol{x}\|_2 = 1$; and for non-Lipschitz, bounded neural activations $\boldsymbol{\tau}^{[\widetilde{\jmath}:j]}$, we set $L_a^{[\widetilde{\jmath}:j]} = B^{[\widetilde{\jmath}:j]}/\phi^{[\widetilde{\jmath}]}$. Considering one such non-Lipschitz neural activation $\boldsymbol{\tau}^{[\widetilde{\jmath}:j]}$, in the recursive definition of the norm-bound $\phi$ in (5), the corresponding term in the sum becomes $L_a^{[\widetilde{\jmath}:j]}\phi^{[\widetilde{\jmath}]} = B^{[\widetilde{\jmath}:j]}$ - so, for example, for a LayerNorm block (Figure 2d) $j \in \mathbb{Z}_E$ we see that $\phi^{[j]} = \sqrt{H^{[\widetilde{\jmath}:j]}}$ (for full derivation see Appendix D.2), and moreover if this is the only node in its layer then the Rademacher complexity bound will be independent of all layers preceeding it. However we would advise caution here; the assumption $\boldsymbol{x} \in \partial\mathbb{X}$ is quite strong and may not be realistic in general. We will discuss how this assumption may be relaxed, along with what impact this relaxation has on our Rademacher complexity bound, in section 4.1.

### 4.1 Generalizations and Standard Toplogies

In this section, we consider two more realistic relaxions assumption 1 - firstly expanding the bounds on $\|\boldsymbol{x}\|_2$, and secondly considering $\boldsymbol{x} \sim \mathcal{X}$ drawn from an unbounded distribution $\mathcal{X}$ such that it lies in the bounded of case 1 with high probability (whp). Using these, we conclude the paper by analysing a range of standard network topologies. Formally, we consider two generalization of assumption 1:

**Strictly Bounded:** $\boldsymbol{x} \in \mathbb{X}_{\rho,r} = \{\boldsymbol{x} \in \mathbb{R}^n : \rho \le \|\boldsymbol{x}\|_2 \le r\}$, where $0 \le \rho \le r \in \mathbb{R}_+$ and $\rho > 0$ if the network contains non-Lipschitz neural activations.

**Distributional:** $\boldsymbol{x} \sim \mathcal{X}$ for a distribution $\mathcal{X}$ for which there exists $0 \le \rho \le r \in \mathbb{R}_+$ ($\rho > 0$ if the network contains non-Lipschitz neural activations) such that $\boldsymbol{x} \in \mathcal{X}_{\rho,r}$ with high probability $\ge 1 - \epsilon$.

In both cases we consider a mild modification of our feature map (5), precisely:[4]

$$\boldsymbol{\Psi}^{[0]}(\Theta) = r\mathbf{I}_n, \boldsymbol{\phi}^{[0]}(\boldsymbol{x}) = \boldsymbol{x}, \mathbf{g}^{[0]} = \tfrac{1}{r}\mathbf{1}_n, \psi^{[0]} = r, \phi_\downarrow^{[0]} = \rho, \phi^{[0]} = r, \quad \phi_\downarrow = \phi_\downarrow^{[E]}$$

$$\phi_\downarrow^{[j]2} = \beta^{[j]2} + \circledcirc^{[j]}_{\widetilde{\jmath}\in\mathbb{A}^{[j]}}\omega^{[\widetilde{\jmath}:j]2}\begin{cases} L_{\phi^{[\widetilde{\jmath}]}}^{[\widetilde{\jmath}:j]2}\phi_\downarrow^{[\widetilde{\jmath}]2} & \text{if } \boldsymbol{\tau}^{[\widetilde{\jmath}:j]} \text{ is Lipschitz} \\ B^{[\widetilde{\jmath}:j]2} & \text{otherwise} \end{cases} \quad \forall j \in \mathbb{Z}_E, \tag{11}$$

and moreover for non-Lipschitz, bounded neural activations $\boldsymbol{\tau}^{[\widetilde{\jmath}:j]}$, we set $L_a^{[\widetilde{\jmath}:j]} = B^{[\widetilde{\jmath}:j]}/\phi_\downarrow^{[\widetilde{\jmath}]}$. For a full discussion of this generalization see Appendix C. Observe that, in the limit $\eta \to 0^+$:

$$\phi^{[j]2} = \beta^{[j]2} + \circledcirc^{[j]}_{\widetilde{\jmath}\in\mathbb{A}^{[j]}}\omega^{[\widetilde{\jmath}:j]2}\begin{cases} L_{\phi^{[\widetilde{\jmath}]}}^{[\widetilde{\jmath}:j]2}\phi^{[\widetilde{\jmath}]2} & \text{if } \boldsymbol{\tau}^{[\widetilde{\jmath}:j]} \text{ is Lipschitz} \\ B^{[\widetilde{\jmath}:j]2}\frac{\phi^{[\widetilde{\jmath}]2}}{\phi_\downarrow^{[\widetilde{\jmath}]2}} & \text{otherwise} \end{cases} \quad \forall j \in \mathbb{Z}_E \tag{12}$$

$$\phi_\downarrow^{[j]} \le \lim_{\eta\to 0^+}\left\|\boldsymbol{\phi}^{[j]}(\boldsymbol{x})\right\|_2 \le \phi^{[j]} \quad \forall\boldsymbol{x} \in \mathbb{X}_{\rho,r}$$

---

[4]In both the cases $\rho = 0$, $r = 1$ (the fully Lipschitz variant of assumption 1) and $\rho = r = 1$ (the non-Lipschitz variant of assumption 1) this reduces to the standard feature map (5).

The Rademacher complexity bound (Theorem 8) takes the same form as usual. The exact impact of letting $r \neq 1$ is dependent on the network topology. For a simple, layerwise, fully Lipschitz neural network with 1 trivial node $j = \jmath$ per layer, as demonstrated in Appendix E.1:[5]

$$\mathcal{R}_N \left( h \circ \mathcal{F} : \left\| \mathbf{W}^{[\jmath-1:\jmath]} \right\|_2 \leq \omega^{[\jmath-1:\jmath]} \right) \leq r \frac{H_m \prod_{\jmath=1}^{D} L^{[\jmath-1:\jmath]} \omega^{[\jmath-1:\jmath]}}{\sqrt{N}}$$

This bound is exponential in depth, as discussed previously. As a mild generalization of this scenario, if we allow non-Lipschitz neural activations (for example LayerNorm blocks) in this simple network, with the last such at layer $\jmath = D_\downarrow$, then, using (12) and noting that $\phi^{[\jmath]} / \phi_\downarrow^{[\jmath]} = \frac{r}{\rho} \forall \jmath \in \mathbb{Z}_D \cup \{0\}$:

$$\mathcal{R}_N \left( h \circ \mathcal{F} : \left\| \mathbf{W}^{[\jmath-1:\jmath]} \right\|_2 \leq \omega^{[\jmath-1:\jmath]} \right) \leq \frac{r}{\rho} \frac{H_m B^{[D_\downarrow-1:D_\downarrow]} \omega^{[D_\downarrow-1:D_\downarrow]} \prod_{\jmath=D_\downarrow+1}^{D} L^{[\jmath-1:\jmath]} \omega^{[\jmath-1:\jmath]}}{\sqrt{N}}$$

where we note that this bound is exponential in the depth to the non-Lipschitz node $D - D_\downarrow$ and proportional to $\frac{r}{\rho}$. The independence from the weights of layers preceeding $D_\downarrow$ is noteworthy, but if we consider as an example a ReLU network terminated by a LayerNorm and observe that the scale of these weights is entirely arbitrary, it perhaps not surprising. The $\frac{1}{\rho}$ term reflects the need to assume that, in the worst-case, small inputs will be "amplified" (e.g. by LayerNorm) to the largest possible output.

The transformer can be similarly analysed. The catch in this case is that the attention block is multiplicative. In particular (see Appendices D.3, D.4 for details), for an attention block:

$$\frac{\phi_{\text{out}}}{\phi_{\text{out}\downarrow}} = \frac{\phi_{\text{out},Q}}{\phi_{\text{out},Q\downarrow}} \frac{\phi_{\text{out},K}}{\phi_{\text{out},K\downarrow}} \frac{\phi_{\text{in},V}}{\phi_{\text{in},V\downarrow}}$$

so, unlike the simpler case considered above, each attention block will cause polynomial growth in the ratio $\frac{\phi}{\phi_\downarrow}$. Subsequently, as shown in Appendix E.3, the overall bound (due to the final LayerNorm) is:

$$\mathcal{R}_N \left( h \circ \mathcal{F} : \left\| \mathbf{W}_{\text{out}} \right\|_2 \leq \omega \right) \leq \left( \frac{\rho}{r} \right)^{3^{3M-1}} \frac{H_m \sqrt{d_{\text{model}}} \omega}{\sqrt{N}}$$

where $\mathbf{W}_{\text{out}}$ are the weights for the linear output layer of the transformer.[6] If $\rho = r$ (that is, $\boldsymbol{x} \in \partial \mathbb{X}$ as in assumption 1) this collapses to $\frac{H_m \sqrt{d_{\text{model}}} \omega}{\sqrt{N}}$, but in general, despite being independent of the weights in all but the output layer of the network, this bound grows doubly-exponentially in depth, dependent on the ratio $\frac{r}{\rho}$ of smallest/largest inputs.

Finally, bounds for the distributional case follow the strictly bounded case, but only whp $\geq 1 - \epsilon$. For example, in Appendix C.4 we consider $\boldsymbol{x} \sim \mathcal{X} = \mathcal{N}(\mathbf{0}_n, \sigma^2 \mathbf{I}_n)$, showing that $\rho \leq \|\boldsymbol{x}\|_2 \leq r$, where:

$$\rho = 0, \ r = \sqrt{2n \ln\left(\frac{2}{\epsilon}\right)} \sigma \qquad \text{or} \qquad \frac{r}{\rho} = \frac{\sqrt{n \ln\left(\frac{4}{\epsilon}\right)}}{\left(\Gamma\left(\frac{n}{2}+1\right)\frac{\epsilon}{2}\right)^{\frac{1}{n}}}$$

whp $\geq 1 - \epsilon$ which apply, respectively, for the *purely Lipchitz* and *bounded* cases. In particular, the latter result allows one to explore Rademacher complexity bounds in the general case without giving $\rho$ or $r$ (the bounds on $\|\boldsymbol{x}\|_2$, where $\rho$ in particular may be difficult to quantify intuitively) a-priori.

## 5   Conclusions

In this paper we have constructed a dual model of a very general set of feedforward neural networks that re-expresses them as a continuous bilinear product between a weight/bias feature map and a data feature map - that is, a reproducing kernel Banach space (RKBS) model. This model is *exact*, with no approximation or assumptions beyond bounded (norm) inputs, bounded (spectral norm) weights and biases, and finite-energy neural activations, and incorporates networks ranging from simple layerwise models (ReLU etc) to ResNet and Transformers. Subsequently, we have applied this model to the analysis of the Rademacher complexity analysis of neural networks, giving a simple recursive bound for the Rademacher complexity of all models neural network topologies covered by our model. This bound is exact (non-asymptotic) and does not include depth- or width- dependent nuisance factors. Moreover it is width-independent and, while exponential in depth (due to the multiplicative build-up of terms through the layers of the networks), enables us to derive straightforward (spectral) threshold conditions under which depth-dependence may be removed entirely.

---

[5]This also applies to ResNet, where for residual blocks $\jmath$ with $d$ internal layers we let $\omega^{[\jmath-1:\jmath]2} = (\omega^{[\bar{\jmath}-1:\bar{\jmath}]_d 2} \ldots \omega^{[\bar{\jmath}-1:\bar{\jmath}]_2 2} \omega^{[\bar{\jmath}-1:\bar{\jmath}]_1 2} + 1 - s^2$ as also described in Appendix E.1.

[6]$M$ here is the size of the encoder/decoder stacks. We use $M$ here rather than $N$ as used in (Vaswani et al., 2017) to avoid a notational ambiguity within our paper.

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

# A  Properties of Hermite polynomials

## A.1  Univariate Case

The (probabilist's) Hermite polynomials are given by (Abramowitz et al., 1972; Morse & Feshbach, 1953; Olver et al., 2010; Courant & Hilbert, 1937):

$$He_k(\zeta) = (-1)^k e^{\frac{\zeta^2}{2}} \frac{d^k}{d\zeta^k} e^{-\frac{\zeta^2}{2}} \qquad \forall k \in \mathbb{N}$$

or, explicitly:

$$He_k(\zeta) = k! \sum_{0 \le 2p \le k} \frac{(-1)^p}{2^p p!(k-2p)!} \zeta^{k-2p} \qquad \forall k \in \mathbb{N} \tag{13}$$

and form an orthogonal basis of $L^2(\mathbb{R}, e^{-x^2})$. For any $f \in L^2(\mathbb{R}, e^{-x^2})$ there exist Hermite coefficients $a_0, a_1, \ldots \in \mathbb{R}$ (the Hermite transform of $f$) such that:

$$f(\zeta) = \sum_{k \in \mathbb{N}} a_k He_k(\zeta) \quad \forall \xi, \zeta \in \mathbb{R}$$

where:

$$a_k = \frac{1}{k!\sqrt{2\pi}} \int_{-\infty}^{\infty} f(\xi + \zeta) e^{-\frac{\zeta^2}{2}} He_k(\zeta) d\zeta$$

and moreover the series representation converges everywhere on the real line.[7]

The Hermite numbers derive from the Hermite polynomials:[8]

$$\mathrm{He}_k \triangleq He_k(0) = \begin{cases} \frac{(-1)^{k/2}k!}{k!!} & \text{if } k \text{ even} \\ 0 & \text{otherwise} \end{cases} \tag{15}$$

where $k!! = k(k-2)(k-4)\ldots$ is the double-factorial. It is well known that (see eg. (Morse & Feshbach, 1953)):

$$He_k(\xi + \zeta) = \sum_{0 \le l \le k} \binom{k}{l} He_{k-l}(\xi) \zeta^l$$

and so:

$$He_k(\zeta) = \sum_{0 \le l \le k} \binom{k}{l} \mathrm{He}_{k-l} \zeta^l$$

It follows that, taking care not to change or order of summation (remember this is an alternating series, so convergence depends on the order of the summation):

$$f(\zeta) - f(0) = \sum_{k=1}^{\infty} a_k \sum_{l=1}^{k} \binom{k}{l} \mathrm{He}_{k-l} \zeta^l$$

For later reference we also note that the Hermite polynomials satisfy the well-known recursion and derivative relation for $k > 1$:

$$\begin{aligned} \zeta H_k(\zeta) &= \tfrac{1}{2} H_{k+1}(\zeta) + \tfrac{1}{2} H'_k(\zeta) \\ &= \tfrac{1}{2} H_{k+1}(\zeta) + k H_{k-1}(\zeta) \end{aligned} \tag{16}$$

---

[7]Hille (1940); Boyd (1980) show that this series converges on a strip $\mathbb{X}_\rho = \{z \in \mathbb{C} : -\rho < \mathrm{Im}(z) < \rho\}$ of width $\rho$ about the real axis in the complex plane, where (note that Hille (1940); Boyd (1980) use the normalized physicist's Hermite polynomials. The additional scale factor here arises in the translation to the un-normalized probabilist's Hermite polynomials used here):

$$\rho = -\limsup_{k \to \infty} \frac{1}{\sqrt{2k+1}} \log \left( \left| \frac{a_k}{\sqrt{k!\sqrt{\pi}}} \right| \right) \tag{14}$$

[8]Typically the Hermite numbers are defined from the physicist's Hermite polynomials, but as we use the Probabilist's form as we find these more convenient for our purposes.

## A.2 Multivariate Case

The multivariate Hermite polynomials $He_{\mathbf{k}} : \mathbb{R}^n \to \mathbb{R}$, $\mathbf{k} \in \mathbb{N}^n$, are the functions (Rahman, 2017):

$$He_{\mathbf{k}}(\boldsymbol{\zeta}) = (-1)^{|\mathbf{k}|} \exp\left(\tfrac{1}{2}\boldsymbol{\zeta}^{\mathrm{T}}\boldsymbol{\zeta}\right) \frac{\partial^{\mathbf{k}}}{\partial \boldsymbol{\zeta}^{\mathbf{k}}} \exp\left(-\tfrac{1}{2}\boldsymbol{\zeta}^{\mathrm{T}}\boldsymbol{\zeta}\right) = \prod_i He_{k_i}(\zeta_i)$$

where we use multi-index notation $|\mathbf{k}| = \sum_i k_i$, $\mathbf{a}^{\mathbf{k}} = \prod_i a_i^{k_i}$, $\mathbf{k}! = \prod_i k_i!$, $\mathbf{k}!! = \prod_i k_i!!$, and $\frac{\partial^{\mathbf{k}}}{\partial \boldsymbol{\zeta}^{\mathbf{k}}} = \prod_i \frac{\partial^{k_i}}{\partial \zeta_i^{k_i}}$. For any $f \in L^2(\mathbb{R}^n, e^{-\boldsymbol{\zeta}^{\mathrm{T}}\boldsymbol{\zeta}})$ there exists coefficients $a_{\mathbf{k}} \in \mathbb{R} : \mathbf{k} \succeq_n i$ (the Hermite transform of $f$), where $\mathbf{k} \succeq_n i$ means $\mathbf{k} \in \{\mathbf{k} \in \mathbb{N}^n : |\mathbf{k}| \geq i\}$, such that:

$$f(\boldsymbol{\zeta}) = \sum_{\mathbf{k} \succeq_n 0} a_{\mathbf{k}} He_{\mathbf{k}}(\boldsymbol{\zeta}) \quad \forall \boldsymbol{\zeta} \in \mathbb{R}^n$$

where:

$$
\begin{aligned}
a_{\boldsymbol{k}} &= \frac{1}{\mathbf{k}!(2\pi)^{\frac{n}{2}}} \int_{-\infty}^{\infty} f(\boldsymbol{\zeta}) e^{-\frac{\boldsymbol{\zeta}^{\mathrm{T}}\boldsymbol{\zeta}}{2}} He_{\mathbf{k}}(\boldsymbol{\zeta}) \, d\boldsymbol{\zeta} \\
&= \frac{1}{k_1!\sqrt{2\pi}} \int_{\zeta_1 \in \mathbb{R}} e^{-\frac{\zeta_1^2}{2}} He_{k_1}(\zeta_1) \frac{1}{k_2!\sqrt{2\pi}} \int_{\zeta_2 \in \mathbb{R}} e^{-\frac{\zeta_2^2}{2}} He_{k_2}(\zeta_2) \dots f(\boldsymbol{\zeta}) \dots d\zeta_2 d\zeta_1
\end{aligned}
$$

and the series representation converges everywhere on $\mathbb{R}^n$.

As in the univariate case, the multivariate Hermite numbers are defined as:

$$\mathrm{He}_{\mathbf{k}} \triangleq He_{\mathbf{k}}(\mathbf{0}) = \prod_i \mathrm{He}_{k_i} = \begin{cases} \frac{(-1)^{|\mathbf{k}|/2}\mathbf{k}!}{\mathbf{k}!!} & \text{if } k_0, k_1, \dots \text{ are all even} \\ 0 & \text{otherwise} \end{cases}$$

where in the final step we have used (15). Subsequently:

$$f(\boldsymbol{\zeta}) - f(\mathbf{0}) = \sum_{\mathbf{k} \succ_n 0} a_{\mathbf{k}} \sum_{0 \prec_n \mathbf{1} \leq \mathbf{k}} \binom{\mathbf{k}}{\mathbf{1}} \mathrm{He}_{\mathbf{k}-\mathbf{1}} \boldsymbol{\zeta}^{\mathbf{1}}$$

where $\mathbf{k} \succ_n i$ means $\mathbf{k} \in \{\mathbf{k} \in \mathbb{N}^n : |\mathbf{k}| > i\}$.

Finally, if we consider a vector-valued function $\mathbf{f} : \mathbb{R}^n \to \mathbb{R}^m$ then it is not hard to see that scalar-valued expansion can be extended to:

$$\mathbf{f}(\boldsymbol{\zeta}) - \mathbf{f}(\mathbf{0}) = \sum_{\mathbf{k} \succ_n 0} \mathbf{a}_{\mathbf{k}} \sum_{0 \prec_n \mathbf{1} \leq \mathbf{k}} \binom{\mathbf{k}}{\mathbf{1}} \mathrm{He}_{\mathbf{k}-\mathbf{1}} \boldsymbol{\zeta}^{\mathbf{1}} \tag{17}$$

where $a_{\mathbf{k},i}$ are the Hermite coefficients of $f_i$. We note that if $n = m$ and $\mathbf{f}(\boldsymbol{\zeta}) = [g(\zeta_i)]_i$ acts elementwise (for example a neural activation that acts elementwise) then:

$$a_{\mathbf{k},i} = \delta_{|\mathbf{k}|,k_i} b_{|\mathbf{k}|} \tag{18}$$

where $b_0, b_1, \dots$ are the (univariate) Hermite coefficients of $g : \mathbb{R} \to \mathbb{R}$.

# B ReLU Activation Function Analysis

In this section we derive the Hermite-polynomial expansion of the ReLU activation function:

$$\tau^{[\mathrm{ReLU}]}(\zeta) = [\zeta]_+$$

We find it convenient to work in terms of the physicists Hermite polynomials $H_k$ to suit (Gradshteyn & Ryzhik, 2000). So:

$$
\begin{aligned}
b_k^{[\mathrm{ReLU}]} &= \frac{1}{\sqrt{2\pi}k!} \int_0^{\infty} \zeta e^{-\frac{\zeta^2}{2}} He_k(\zeta) \, d\zeta \\
&= \frac{1}{\sqrt{2\pi}k!} \int_0^{\infty} \zeta e^{-\frac{\zeta^2}{2}} \frac{1}{\sqrt{2}^k} H_k\left(\frac{\zeta}{\sqrt{2}}\right) d\zeta \\
&= \sqrt{\frac{2}{\pi}} \frac{1}{k!} \int_0^{\infty} \frac{\zeta}{\sqrt{2}} e^{-\left(\frac{\zeta}{\sqrt{2}}\right)^2} \frac{1}{\sqrt{2}^k} H_k\left(\frac{\zeta}{\sqrt{2}}\right) d\frac{\zeta}{\sqrt{2}} \\
&= \sqrt{\frac{2}{\pi}} \frac{1}{\sqrt{2}^k k!} \int_0^{\infty} \zeta e^{-\zeta^2} H_k(\zeta) \, d\zeta
\end{aligned}
$$

and hence, using (16) and (Gradshteyn & Ryzhik, 2000, (7.373)):

$$b_k^{[\mathrm{ReLU}]} = \frac{k+1}{\sqrt{\pi}} \frac{1}{\sqrt{2}^{k+1}(k+1)!} \int_0^{\infty} e^{-\zeta^2} H_{k+1}(\zeta) \, d\zeta + \frac{1}{\sqrt{\pi}} \frac{1}{\sqrt{2}^{k-1}(k-1)!} \int_0^{\infty} e^{-\zeta^2} H_{k-1}(\zeta) \, d\zeta$$

and using (Gradshteyn & Ryzhik, 2000, (7.373)) again:

$$b_k^{[\text{ReLU}]} = \frac{1}{\sqrt{\pi}} \frac{1}{\sqrt{2}^{k+1}(k+1)!}(k+1)\left(e^0 H_k(0) - e^{-\frac{\infty^2}{2}} H_k\left(\frac{\infty}{\sqrt{2}}\right)\right) + \dots$$

$$\dots \frac{1}{\sqrt{\pi}} \frac{1}{\sqrt{2}^{k-1}(k-1)!}\left(e^0 H_{k-2}(0) - e^{-\frac{\infty^2}{2}} H_{k-2}\left(\frac{\infty}{\sqrt{2}}\right)\right)$$

$$= \frac{1}{\sqrt{2\pi}}\left(\frac{k+1}{\sqrt{2}^k(k+1)!} H_k(0) + \frac{1}{\sqrt{2}^{k-2}(k-1)!} H_{k-2}(0)\right)$$

If $k = 2p$ and $p > 0$ then, noting that $H_k(0) = \sqrt{2}^k \text{He}_k$:

$$b_{2p}^{[\text{ReLU}]} = \frac{1}{\sqrt{2\pi}}\left(\frac{1}{\sqrt{2}^{2p}(2p+1)!}(2p+1)H_{2p}(0) + \frac{1}{\sqrt{2}^{2p-2}(2p-1)!}H_{2p-2}(0)\right)$$

$$= \frac{1}{\sqrt{2\pi}}\left(\frac{1}{\sqrt{2}^{2p}(2p+1)!}(2p+1)H_{2p}(0) + \frac{1}{\sqrt{2}^{2p-2}(2p-1)!}H_{2p-2}(0)\right)$$

$$= \frac{(-1)^{p+1}}{\sqrt{2\pi}(2p-1)2^p p!}\left(\frac{(-1)^{p+1}(2p-1)p!}{(2p)!}H_{2p}(0) + \frac{(-1)^{p+1}2p!}{(2(p-1))!}H_{2p-2}(0)\right)$$

$$= \frac{(-1)^{p+1}}{\sqrt{2\pi}(2p-1)2^p p!}e^{-\frac{\xi^2}{2}}\left(\frac{(-1)^{p+1}(2p-1)p!}{(2p)!}2^p\frac{(-1)^p(2p)!}{2^p p!} + \frac{(-1)^{p+1}2p!}{(2(p-1))!}2^{p-1}\frac{(-1)^{p+1}(2p-2)!}{(p-1)!2^{p-1}}\right)$$

$$= \frac{(-1)^{p+1}}{\sqrt{2\pi}(2p-1)2^p p!}$$

If $k = 2p + 1$ and $p > 0$ then:

$$b_{2p+1}^{[\text{ReLU}]} = \frac{1}{\sqrt{2\pi}}\left(\frac{1}{\sqrt{2}^{2p+1}(2p+2)!}(2p+2)H_{2p+1}(0) + \frac{1}{\sqrt{2}^{2p-1}(2p)!}H_{2p-1}(0)\right)$$

$$= \frac{1}{\sqrt{2\pi}}\left(-\frac{1}{\sqrt{2}^{2p+1}(2p+2)!}(2p+2)H_{2p+1}(0) - \frac{1}{\sqrt{2}^{2p-1}(2p)!}H_{2p-1}(0)\right)$$

$$= \frac{1}{\sqrt{2\pi}}\left(-\frac{\sqrt{2}}{2^{p+1}(2p+1)!}H_{2p+1}(0) - \frac{\sqrt{2}}{2^p(2p)!}H_{2p-1}(0)\right)$$

$$= 0$$

For the cases $k = 0, 1$ we use the result:

$$\int_a^b \zeta^m e^{-\zeta^2} d\zeta = \frac{1}{2}\Gamma\left(\frac{m+1}{2}, a^2\right) - \frac{1}{2}\Gamma\left(\frac{m+1}{2}, b^2\right)$$

and so:

$$\int_a^\infty \zeta^m e^{-\zeta^2} d\zeta = \frac{1}{2}\Gamma\left(\frac{m+1}{2}, a^2\right)$$

In the case $k = 0$:

$$b_0^{[\text{ReLU}]} = \sqrt{\frac{2}{\pi}}\int_0^\infty \zeta e^{-\zeta^2} d\zeta$$

$$= \frac{1}{\sqrt{2\pi}}\Gamma(1, 0)$$

$$= \frac{1}{\sqrt{2\pi}}$$

and in the case $k = 1$:

$$b_1^{[\text{ReLU}]} = \frac{2}{\sqrt{\pi}}\int_0^\infty \zeta^2 e^{-\zeta^2} d\zeta$$

$$= \frac{1}{\sqrt{\pi}}\Gamma\left(\frac{3}{2}, 0\right)$$

$$= \frac{1}{2}$$

Subsequently, for the elementwise ReLU neural activaiton, using (18):

$$\mathbf{a_k} = \left[\ \delta_{|\mathbf{k}|, k_i} b_{|\mathbf{k}|}\ \right]_i \tag{19}$$

Next we derive the magnitude functions for the ReLU. Using integration by parts, we see that:

$$\frac{1}{\sqrt{2\pi}}\int_c^\zeta \frac{1}{\xi^2}\left(e^{\frac{1}{2}\xi^2} - 1\right)d\xi = \frac{1}{\sqrt{2\pi}}\frac{1}{\sqrt{2}}\int_c^\zeta \frac{2}{\xi^2}\left(e^{\frac{1}{2}\xi^2} - 1\right)d\frac{\xi}{\sqrt{2}}$$

$$= \frac{1}{\sqrt{2\pi}}\frac{1}{\sqrt{2}}\int_{\frac{c}{\sqrt{2}}}^{\frac{\zeta}{\sqrt{2}}} \frac{1}{\xi^2}\left(e^{\xi^2} - 1\right)d\xi$$

$$= -\frac{1}{\sqrt{2\pi}}\frac{1}{\zeta}\left(e^{\frac{1}{2}\zeta^2} - 1\right) + \frac{1}{\sqrt{2\pi}}\frac{1}{c}\left(e^{\frac{1}{2}c^2} - 1\right) + \frac{1}{\sqrt{\pi}}\int_{\frac{c}{\sqrt{2}}}^{\frac{\zeta}{\sqrt{2}}} e^{\xi^2} d\xi$$

$$= -\frac{1}{\sqrt{2\pi}}\frac{1}{\zeta}\left(e^{\frac{1}{2}\zeta^2} - 1\right) + \frac{1}{\sqrt{2\pi}}\frac{1}{c}\left(e^{\frac{1}{2}c^2} - 1\right) + \frac{1}{2}\frac{2}{\sqrt{\pi}}\int_{\frac{c}{\sqrt{2}}}^{\frac{\zeta}{\sqrt{2}}} e^{\xi^2} d\xi$$

$$= -\frac{1}{\sqrt{2\pi}}\frac{1}{\zeta}\left(e^{\frac{1}{2}\zeta^2} - 1\right) + \frac{1}{2}\text{erfi}\left(\frac{\zeta}{\sqrt{2}}\right) - \frac{1}{2}\left(\text{erfi}\left(\frac{c}{\sqrt{2}}\right) - \frac{1}{\sqrt{2\pi}}\frac{2}{c}\left(e^{\frac{1}{2}c^2} - 1\right)\right)$$

So:

$$\sum_{k=1}^{\infty} \left| b_k^{[\text{ReLU}]} \right| \zeta^k = \tfrac{1}{2}\zeta + \tfrac{1}{\sqrt{2\pi}} \sum_{p=1}^{\infty} \frac{\zeta^{2p}}{(2p-1)2^p p!}$$

$$= \tfrac{1}{2}\zeta + \tfrac{1}{\sqrt{2\pi}}\zeta \sum_{p=1}^{\infty} \frac{\zeta^{2p-1}}{(2p-1)2^p p!}$$

$$= \tfrac{1}{2}\zeta + \tfrac{1}{\sqrt{2\pi}}\zeta \int_c^{\zeta} \left( \frac{\partial}{\partial \xi} \sum_{p=1}^{\infty} \frac{\xi^{2p-1}}{(2p-1)2^p p!} \right) d\xi$$

$$= \tfrac{1}{2}\zeta + \tfrac{1}{\sqrt{2\pi}}\zeta \int_c^{\zeta} \left( \sum_{p=1}^{\infty} \frac{\xi^{2p-2}}{2^p p!} \right) d\xi$$

$$= \tfrac{1}{2}\zeta + \tfrac{1}{2\sqrt{2\pi}}\zeta \int_c^{\zeta} \left( \sum_{p=1}^{\infty} \frac{1}{p!}\left(\tfrac{1}{2}\xi^2\right)^{p-1} \right) d\xi$$

$$= \tfrac{1}{2}\zeta + \tfrac{1}{\sqrt{2\pi}}\zeta \int_c^{\zeta} \tfrac{1}{\xi^2} \left( \sum_{p=1}^{\infty} \frac{1}{p!}\left(\tfrac{1}{2}\xi^2\right)^{p} \right) d\xi$$

$$= \tfrac{1}{2}\zeta + \tfrac{1}{\sqrt{2\pi}}\zeta \int_c^{\zeta} \tfrac{1}{\xi^2} \left( e^{\frac{1}{2}\xi^2} - 1 \right) d\xi$$

$$= \tfrac{1}{2}\zeta \left( \text{erfi}\left(\tfrac{\zeta}{\sqrt{2}}\right) + 1 - \text{erfi}\left(\tfrac{c}{\sqrt{2}}\right) + \tfrac{1}{\sqrt{2\pi}}\tfrac{2}{c}\left( e^{\frac{1}{2}c^2} - 1 \right) \right) + \tfrac{1}{\sqrt{2\pi}} \left( 1 - e^{\frac{1}{2}\zeta^2} \right)$$

Select $c$ so that the first derivative is $\tfrac{1}{2}\zeta$:

$$-\text{erfi}\left(\tfrac{c}{\sqrt{2}}\right) + \tfrac{1}{\sqrt{2\pi}}\tfrac{2}{c}\left( e^{\frac{1}{2}c^2} - 1 \right) = 0 \text{ if } c = 0$$

Hence:

$$s_\eta^{[\text{ReLU}]}(\zeta) \triangleq \sum_{k=1}^{\infty} \left| b_k^{[\text{ReLU}]} \right| (1+\eta\zeta)^k - \sum_{k=1}^{\infty} \left| b_k^{[\text{ReLU}]} \right|$$

$$= \tfrac{1}{2}(1+\eta\zeta)\left( \text{erfi}\left(\tfrac{1+\eta\zeta}{\sqrt{2}}\right) + 1 \right) + \tfrac{1}{\sqrt{2\pi}}\left( 1 - e^{\frac{1}{2}(1+\eta\zeta)^2} \right) - \tfrac{1}{2}\left( \text{erfi}\left(\tfrac{1}{\sqrt{2}}\right) + 1 \right) - \ldots$$
$$\ldots - \tfrac{1}{\sqrt{2\pi}}\left( 1 - e^{\frac{1}{2}} \right) \tag{20}$$

$$= \tfrac{1}{2}\eta\zeta\left( \text{erfi}\left(\tfrac{1+\eta\zeta}{\sqrt{2}}\right) + 1 \right) + \tfrac{1}{\sqrt{2\pi}}\left( e^{\frac{1}{2}} - e^{\frac{1}{2}(1+\eta\zeta)^2} \right) + \tfrac{1}{2}\left( \text{erfi}\left(\tfrac{1+\eta\zeta}{\sqrt{2}}\right) - \text{erfi}\left(\tfrac{1}{\sqrt{2}}\right) \right)$$

## C  Bilinear Representation - Proofs, Bounds and Generalizations

In this section we present proof of theorems, bounds and generalizations related to the bilinear representation. To avoid repeating work we consider a mild generalization of the map presented in the body of the paper, as shown in Figure 6. The key generalizations here over the main body of the paper are:

1. We let $x \in \mathbb{X}_{\rho,r} = \{ x \in \mathbb{R}^n : \rho \leq \|x\|_2 \leq r \}$ for some $0 \leq \rho \leq r \in \mathbb{R}_+$. In the main body of the paper we let $\rho = 0, r = 1$ for simplicity when all neural activations are Lipschitz, and $\rho = r = 1$ otherwise. In general we require $\rho > 0$ when considering a network containing non-Lipschitz neural activations.

2. We use base-case $\mathbf{\Psi}^{[0]}(\Theta) = r\mathbf{I}_n$, $\mathbf{g}^{[0]} = \tfrac{1}{r}\mathbf{1}_n$ here (recall $r = 1$ in the main body).

3. We use $L_{\psi_\eta^{[\tilde{j}]}}^{[\tilde{j}:j]}, L_{\phi_\eta^{[\tilde{j}]}}^{[\tilde{j}:j]}$ to scale the feature map here rather than $L_{\psi^{[\tilde{j}]}}^{[\tilde{j}:j]}$ and $L_{\phi^{[\tilde{j}]}}^{[\tilde{j}:j]}$. Note, however, that (as we demonstrate) $\lim_{\eta \to 0} \psi_\eta^{[\tilde{j}]} = \psi^{[\tilde{j}]}$ and $\lim_{\eta \to 0} \phi_\eta^{[\tilde{j}]} = \phi^{[\tilde{j}]}$, so the definitions coincide in the limit $\eta \to 0^+$, which is the case we are primarily concerned with (as it is used in our Rademacher complexity bound).

4. For non-Lipschitz, bounded neural activations (edges), we let $L^{[\tilde{j}:j]} = \frac{B^{[\tilde{j}:j]}}{\phi_{\downarrow\eta}^{[\tilde{j}]2}}$, where $\phi_{\downarrow\eta}^{[\tilde{j}]}$ is a lower bound on $\|\phi^{[\tilde{j}]}(x)\|_2$ (recall that $\rho = 1$ in the main body of the paper, and note that we will prove that $\phi_{\downarrow\eta}^{[\tilde{j}]} = \phi_\eta^{[\tilde{j}]}$ in this case). More generally for neural activations that are neither bounded or Lipschitz we let $L^{[\tilde{j}:j]} = 1$. Note, however, that we cannot prove continuity of our bilinear product in this case, so the relevant parts of the proof do not apply for this.

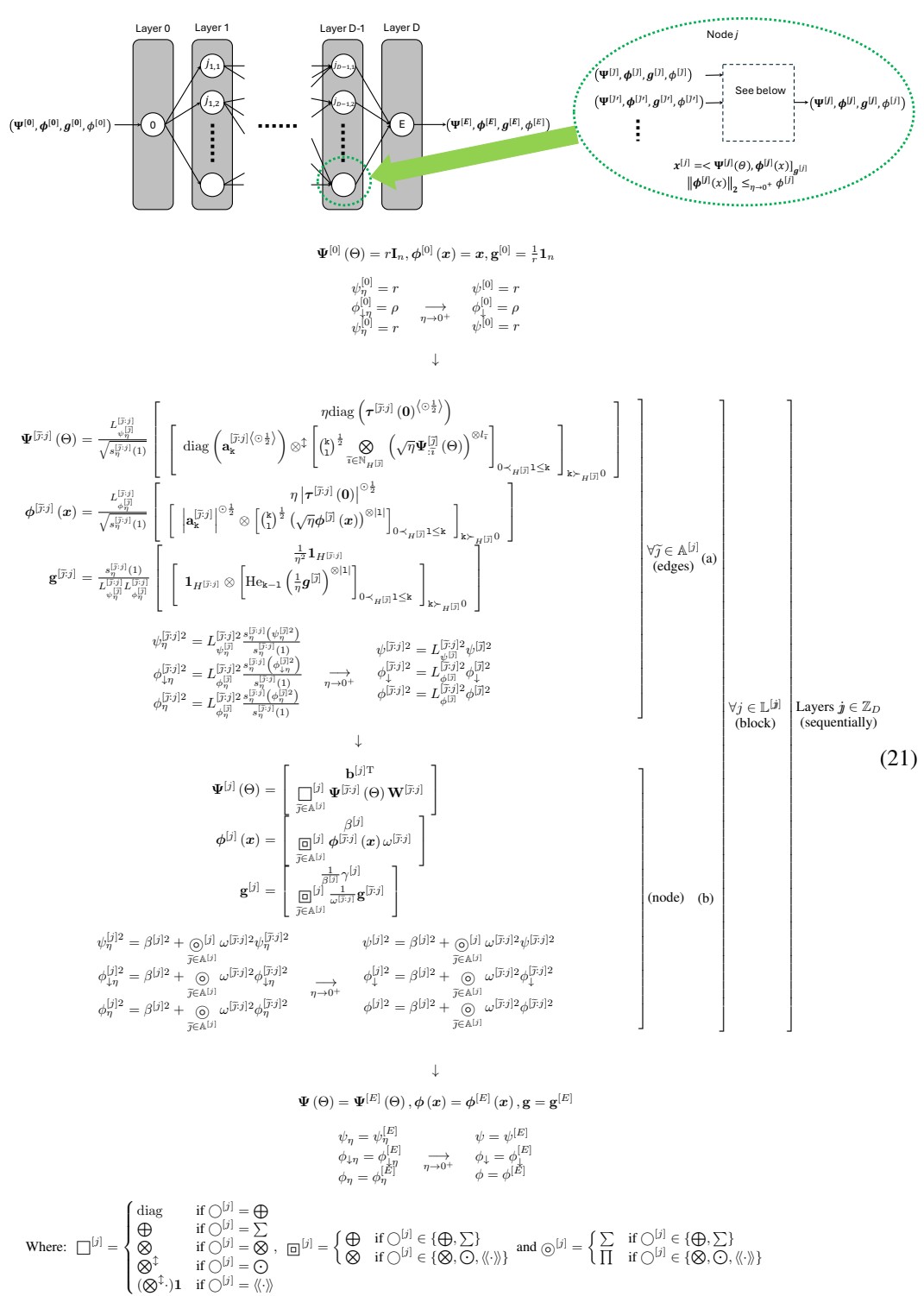

Figure 6: Complete version of Figure 4 (recursive definition of bilinear representation) splitting edge/node maps, showing limits and including correct weights (the main body uses simplified weights that are correct in the limit and sets $r = 1$, $\rho = 0$ (or $\rho = 1$ if non-Lipschitz neurons are present). For non-Lipschitz, bounded neural activations $\boldsymbol{\tau}^{[\widetilde{j}:j]}$ we set $L_a^{[\widetilde{j}:j]} = B^{[\widetilde{j}:j]}/\phi_{\downarrow\eta}^{[\widetilde{j}]2}$, and for non-Lipschitz and unbounded neural activations we set $L_a^{[\widetilde{j}:j]} = 1$.

Note that for each $j \in \mathbb{Z}_E$ the feature map construction is split into two steps - a construction (21a) for the incoming edges $[\widetilde{\jmath} : j]$, which we refer to as the **edge case**; and a construction (21b) for the (core of the) node itself, which we refer to as the **node case**. This split will simplify our proofs and improve clarity by separating the key steps therein. As in the main body of the paper the overall representation is:

$$\mathbf{f}(\boldsymbol{x}; \Theta) = \langle \boldsymbol{\Psi}(\Theta), \boldsymbol{\phi}(\boldsymbol{x})]_{\mathbf{g}} \tag{22}$$

We will also show that:

$$\begin{aligned}
\boldsymbol{x}^{[\widetilde{\jmath}:j]} &= \left\langle \boldsymbol{\Psi}^{[\widetilde{\jmath}:j]}(\Theta), \boldsymbol{\phi}^{[\widetilde{\jmath}:j]}(\boldsymbol{x}) \right]_{\mathbf{g}^{[\widetilde{\jmath}:j]}} \quad \forall j \in \mathbb{Z}_E, \widetilde{\jmath} \in \mathbb{A}^{[j]} \\
\boldsymbol{x}^{[j]} &= \left\langle \boldsymbol{\Psi}^{[j]}(\Theta), \boldsymbol{\phi}^{[j]}(\boldsymbol{x}) \right]_{\mathbf{g}^{[j]}} \qquad \forall j \in \mathbb{Z}_E \cup \{0\}
\end{aligned} \tag{23}$$

where the following bounds hold:

$$\begin{aligned}
\left\| \boldsymbol{\Psi}^{[\widetilde{\jmath}:j]}(\Theta) \right\|_F &\leq \psi_\eta^{[\widetilde{\jmath}:j]} & \forall j \in \mathbb{Z}_E, \widetilde{\jmath} \in \mathbb{A}^{[j]}, \Theta \in \mathbb{W} \\
\left\| \boldsymbol{\phi}^{[\widetilde{\jmath}:j]}(\boldsymbol{x}) \right\|_2 &\in \left[ \phi_{\eta\downarrow}^{[\widetilde{\jmath}:j]}, \phi_\eta^{[\widetilde{\jmath}:j]} \right] & \forall j \in \mathbb{Z}_E, \widetilde{\jmath} \in \mathbb{A}^{[j]}, \boldsymbol{x} \in \mathbb{W} \\
\left\| \boldsymbol{\Psi}^{[j]}(\Theta) \right\|_F &\leq \psi_\eta^{[j]} & \forall j \in \mathbb{Z}_E \cup \{0\}, \Theta \in \mathbb{W} \\
\left\| \boldsymbol{\phi}^{[j]}(\boldsymbol{x}) \right\|_2 &\in \left[ \phi_{\eta\downarrow}^{[j]}, \phi_\eta^{[j]} \right] & \forall j \in \mathbb{Z}_E \cup \{0\}, \boldsymbol{x} \in \mathbb{X}
\end{aligned} \tag{24}$$

noting that $\phi_{\eta\downarrow}^{[\widetilde{\jmath}:j]}, \phi_{\eta\downarrow}^{[j]} > 0$ if $\rho > 0$ and:

$$\begin{aligned}
\left\| \boldsymbol{\phi}^{[\widetilde{\jmath}:j]}(\boldsymbol{x}) \right\|_2 &= \phi_{\eta\downarrow}^{[\widetilde{\jmath}:j]} & \forall j \in \mathbb{Z}_E, \widetilde{\jmath} \in \mathbb{A}^{[j]}, \boldsymbol{x} \in \mathbb{W} : \|\boldsymbol{x}\|_2 = \rho \\
\left\| \boldsymbol{\phi}^{[\widetilde{\jmath}:j]}(\boldsymbol{x}) \right\|_2 &= \phi_\eta^{[\widetilde{\jmath}:j]} & \forall j \in \mathbb{Z}_E, \widetilde{\jmath} \in \mathbb{A}^{[j]}, \boldsymbol{x} \in \mathbb{W} : \|\boldsymbol{x}\|_2 = r \\
\left\| \boldsymbol{\phi}^{[j]}(\boldsymbol{x}) \right\|_2 &= \phi_{\eta\downarrow}^{[j]} & \forall j \in \mathbb{Z}_E \cup \{0\}, \boldsymbol{x} \in \mathbb{X} : \|\boldsymbol{x}\|_2 = \rho \\
\left\| \boldsymbol{\phi}^{[j]}(\boldsymbol{x}) \right\|_2 &= \phi_\eta^{[j]} & \forall j \in \mathbb{Z}_E \cup \{0\}, \boldsymbol{x} \in \mathbb{X} : \|\boldsymbol{x}\|_2 = r
\end{aligned} \tag{25}$$

## C.1 Proof of Theorem 1 - Bilinear Representation

Recalling that the network is arranged in layers $\jmath = 0, 1, 2, \ldots, D$, and given that we know the feature map representation for the input layer $\jmath = 0$ is, tivially:

$$\boldsymbol{x}^{[0]} = \left\langle \boldsymbol{\Psi}^{[0]}(\Theta), \boldsymbol{\phi}^{[0]}(\boldsymbol{x}) \right]_{\mathbf{g}^{[0]}}$$

where $\boldsymbol{\Psi}^{[0]}(\Theta) = r\mathbf{I}$, $\boldsymbol{\phi}^{[0]}(\boldsymbol{x}) = \boldsymbol{x}$ and $\mathbf{g}^{[0]} = \frac{1}{r}\mathbf{1}$, it suffices to show that *if* all outputs of all nodes $\widetilde{\jmath} \in \mathbb{L}^{[\jmath-1]}$ ($\mathbb{L}^{[0]} = \{0\}$) in layer $\jmath - 1$ can be expressed in terms of bilinear products:

$$\boldsymbol{x}^{[\widetilde{\jmath}]} = \left\langle \boldsymbol{\Psi}^{[\widetilde{\jmath}]}(\Theta), \boldsymbol{\phi}^{[\widetilde{\jmath}]}(\boldsymbol{x}) \right]_{\mathbf{g}^{[\widetilde{\jmath}]}} \tag{26}$$

*then* all nodes $j \in \mathbb{L}^{[\jmath]}$, using the definitions given, can be written:

$$\boldsymbol{x}^{[\widetilde{\jmath}:j]} = \left\langle \boldsymbol{\Psi}^{[\widetilde{\jmath}:j]}(\Theta), \boldsymbol{\phi}^{[\widetilde{\jmath}:j]}(\boldsymbol{x}) \right]_{\mathbf{g}^{[\widetilde{\jmath}:j]}} \quad \forall \widetilde{\jmath} \in \mathbb{A}^{[j]} \tag{27}$$

and:

$$\boldsymbol{x}^{[j]} = \left\langle \boldsymbol{\Psi}^{[j]}(\Theta), \boldsymbol{\phi}^{[j]}(\boldsymbol{x}) \right]_{\mathbf{g}^{[j]}} \tag{28}$$

We call (27) the edge case and (28) the node case, and will treat them separately.

**Edge case:** We are given that (26) is correct. Substituting (21a) into the bilinear product and using (26), (3) and (17), we find that:

$$\begin{aligned}
\left\langle \boldsymbol{\Psi}^{[\widetilde{\jmath}:j]}(\Theta), \boldsymbol{\phi}^{[\widetilde{\jmath}:j]}(\boldsymbol{x}) \right]_{\mathbf{g}^{[\widetilde{\jmath}:j]}} &= \left\langle \boldsymbol{\Psi}^{[\widetilde{\jmath}:j]}(\Theta), \boldsymbol{\phi}^{[\widetilde{\jmath}:j]}(\boldsymbol{x}) \right]_{\mathbf{g}^{[\widetilde{\jmath}:j]}} \\
&= \boldsymbol{\tau}^{[\widetilde{\jmath}:j]}(\mathbf{0}) + \sum_{\mathbf{k} \succ_{H[\widetilde{\jmath}]} \mathbf{0}} \mathbf{a}_\mathbf{k}^{[\widetilde{\jmath}:j]} \sum_{0 \prec_{H[\widetilde{\jmath}]} \mathbf{1} \leq \mathbf{k}} \binom{\mathbf{k}}{\mathbf{l}} \mathrm{He}_{\mathbf{k}-\mathbf{1}} \left[ \left\langle \bigotimes_{\widetilde{\imath} \in \mathbb{N}_{H[\widetilde{\jmath}]}} \boldsymbol{\Psi}_{:i_{\widetilde{\jmath}}}^{[\widetilde{\jmath}]}(\Theta)^{\otimes l_{i_{\widetilde{\jmath}}}}, \boldsymbol{\phi}^{[\widetilde{\jmath}]}(\boldsymbol{x})^{\otimes|\mathbf{1}|} \right]_{\mathbf{g}^{[\widetilde{\jmath}] \otimes |\mathbf{1}|}} \right]_{i_{\widetilde{\jmath}}} \\
&= \boldsymbol{\tau}^{[\widetilde{\jmath}:j]}(\mathbf{0}) + \sum_{\mathbf{k} \succ_{H[\widetilde{\jmath}]} \mathbf{0}} \mathbf{a}_\mathbf{k}^{[\widetilde{\jmath}:j]} \sum_{0 \prec_{H[\widetilde{\jmath}]} \mathbf{1} \leq \mathbf{k}} \binom{\mathbf{k}}{\mathbf{l}} \mathrm{He}_{\mathbf{k}-\mathbf{1}} \left\langle \boldsymbol{\Psi}^{[\widetilde{\jmath}]}(\Theta), \boldsymbol{\phi}^{[\widetilde{\jmath}]}(\boldsymbol{x}) \right]_{\mathbf{g}^{[\widetilde{\jmath}]}}^{\mathbf{1}} \\
&= \boldsymbol{\tau}^{[\widetilde{\jmath}:j]}(\mathbf{0}) + \sum_{\mathbf{k} \succ_{H[\widetilde{\jmath}]} \mathbf{0}} \mathbf{a}_\mathbf{k}^{[\widetilde{\jmath}:j]} \sum_{0 \prec_{H[\widetilde{\jmath}]} \mathbf{1} \leq \mathbf{k}} \binom{\mathbf{k}}{\mathbf{l}} \mathrm{He}_{\mathbf{k}-\mathbf{1}} \boldsymbol{x}^{[\widetilde{\jmath}]\mathbf{1}} \\
&= \boldsymbol{x}^{[\widetilde{\jmath}:j]}
\end{aligned}$$

which is the desired result (27).

**Node case:** We have shown that (27) is correct. Substituting (21b) into the bilinear product and using (27), we find that, for columnar concatenation nodes $\bigcirc^{[j]} = \bigoplus$ (so $\square^{[j]} = \mathrm{diag}$, $\boxdot^{[j]} = \bigoplus$):

$$
\begin{aligned}
\left[\boldsymbol{\Psi}^{[j]}(\Theta), \phi^{[j]}(\boldsymbol{x})\right]_{\mathbf{g}^{[j]}} &= \gamma^{[j]}\mathbf{b}^{[j]} + \left\langle \underset{\widetilde{\jmath}\in\mathbb{A}^{[j]}}{\square^{[j]}} \frac{1}{\omega^{[\widetilde{\jmath}:j]}} \boldsymbol{\Psi}^{[\widetilde{\jmath}:j]}(\Theta)\, \mathbf{W}^{[\widetilde{\jmath}:j]}, \underset{\widetilde{\jmath}\in\mathbb{A}^{[j]}}{\boxdot^{[j]}} \omega^{[\widetilde{\jmath}:j]}\phi^{[\widetilde{\jmath}:j]}(\boldsymbol{x}) \right\rangle_{\underset{\widetilde{\jmath}\in\mathbb{A}^{[j]}}{\boxdot^{[j]}} \frac{1}{\omega^{[\widetilde{\jmath}:j]}} \mathbf{g}^{[\widetilde{\jmath}:j]}} \\
&= \gamma^{[j]}\mathbf{b}^{[j]} + \left\langle \begin{bmatrix} \frac{1}{\omega^{[\widetilde{\jmath}_1:j]}}\boldsymbol{\Psi}^{[\widetilde{\jmath}_1:j]}(\Theta)\mathbf{W}^{[\widetilde{\jmath}_1:j]} & \mathbf{0} & \cdots \\ \mathbf{0} & \frac{1}{\omega^{[\widetilde{\jmath}_2:j]}}\boldsymbol{\Psi}^{[\widetilde{\jmath}_2:j]}(\Theta)\mathbf{W}^{[\widetilde{\jmath}_2:j]} & \cdots \\ \vdots & \vdots & \ddots \end{bmatrix}, \begin{bmatrix} \omega^{[\widetilde{\jmath}_1:j]}\phi^{[\widetilde{\jmath}_1:j]}(\boldsymbol{x}) \\ \omega^{[\widetilde{\jmath}_2:j]}\phi^{[\widetilde{\jmath}_1:j]}(\boldsymbol{x}) \\ \vdots \end{bmatrix} \right\rangle \begin{bmatrix} \frac{1}{\omega^{[\widetilde{\jmath}_1:j]}}\mathbf{g}^{[\widetilde{\jmath}_1:j]} \\ \frac{1}{\omega^{[\widetilde{\jmath}_2:j]}}\mathbf{g}^{[\widetilde{\jmath}_2:j]} \\ \vdots \end{bmatrix} \\
&= \gamma^{[j]}\mathbf{b}^{[j]} + \begin{bmatrix} \left[\boldsymbol{\Psi}^{[\widetilde{\jmath}_1:j]}(\Theta)\mathbf{W}^{[\widetilde{\jmath}_1:j]}, \phi^{[\widetilde{\jmath}_1:j]}(\boldsymbol{x})\right]_{\mathbf{g}^{[\widetilde{\jmath}_1:j]}} \\ \left[\boldsymbol{\Psi}^{[\widetilde{\jmath}_2:j]}(\Theta)\mathbf{W}^{[\widetilde{\jmath}_2:j]}, \phi^{[\widetilde{\jmath}_2:j]}(\boldsymbol{x})\right]_{\mathbf{g}^{[\widetilde{\jmath}_2:j]}} \\ \vdots \end{bmatrix} \\
&= \gamma^{[j]}\mathbf{b}^{[j]} + \begin{bmatrix} \mathbf{W}^{[\widetilde{\jmath}_1:j]\mathrm{T}}\boldsymbol{x}^{[\widetilde{\jmath}_1:j]} \\ \mathbf{W}^{[\widetilde{\jmath}_2:j]\mathrm{T}}\boldsymbol{x}^{[\widetilde{\jmath}_2:j]} \\ \vdots \end{bmatrix} \\
&= \gamma^{[j]}\mathbf{b}^{[j]} + \underset{\widetilde{\jmath}\in\widetilde{\mathbb{P}}^{[j]}}{\bigoplus} \mathbf{W}^{[\widetilde{\jmath}:j]\mathrm{T}}\boldsymbol{x}^{[\widetilde{\jmath}:j]} = \boldsymbol{x}^{[j]}
\end{aligned}
$$

For additive nodes $\bigcirc^{[j]} = \sum$ (so $\square^{[j]} = \bigoplus$, $\boxdot^{[j]} = \bigoplus$):

$$
\begin{aligned}
\left[\boldsymbol{\Psi}^{[j]}(\Theta), \phi^{[j]}(\boldsymbol{x})\right]_{\mathbf{g}^{[j]}} &= \gamma^{[j]}\mathbf{b}^{[j]} + \left\langle \underset{\widetilde{\jmath}\in\mathbb{A}^{[j]}}{\square^{[j]}} \frac{1}{\omega^{[\widetilde{\jmath}:j]}} \boldsymbol{\Psi}^{[\widetilde{\jmath}:j]}(\Theta)\, \mathbf{W}^{[\widetilde{\jmath}:j]}, \underset{\widetilde{\jmath}\in\mathbb{A}^{[j]}}{\boxdot^{[j]}} \omega^{[\widetilde{\jmath}:j]}\phi^{[\widetilde{\jmath}:j]}(\boldsymbol{x}) \right\rangle_{\underset{\widetilde{\jmath}\in\mathbb{A}^{[j]}}{\boxdot^{[j]}} \frac{1}{\omega^{[\widetilde{\jmath}:j]}} \mathbf{g}^{[\widetilde{\jmath}:j]}} \\
&= \gamma^{[j]}\mathbf{b}^{[j]} + \left\langle \begin{bmatrix} \frac{1}{\omega^{[\widetilde{\jmath}_1:j]}}\boldsymbol{\Psi}^{[\widetilde{\jmath}_1:j]}(\Theta)\mathbf{W}^{[\widetilde{\jmath}_1:j]} \\ \frac{1}{\omega^{[\widetilde{\jmath}_2:j]}}\boldsymbol{\Psi}^{[\widetilde{\jmath}_2:j]}(\Theta)\mathbf{W}^{[\widetilde{\jmath}_2:j]} \\ \vdots \end{bmatrix}, \begin{bmatrix} \omega^{[\widetilde{\jmath}_1:j]}\phi^{[\widetilde{\jmath}_1:j]}(\boldsymbol{x}) \\ \omega^{[\widetilde{\jmath}_2:j]}\phi^{[\widetilde{\jmath}_1:j]}(\boldsymbol{x}) \\ \vdots \end{bmatrix} \right\rangle \begin{bmatrix} \frac{1}{\omega^{[\widetilde{\jmath}_1:j]}}\mathbf{g}^{[\widetilde{\jmath}_1:j]} \\ \frac{1}{\omega^{[\widetilde{\jmath}_2:j]}}\mathbf{g}^{[\widetilde{\jmath}_2:j]} \\ \vdots \end{bmatrix} \\
&= \gamma^{[j]}\mathbf{b}^{[j]} + \sum_{\widetilde{\jmath}\in\mathbb{A}^{[j]}} \left\langle \boldsymbol{\Psi}^{[\widetilde{\jmath}:j]}(\Theta)\mathbf{W}^{[\widetilde{\jmath}:j]}, \phi^{[\widetilde{\jmath}:j]}(\boldsymbol{x}) \right\rangle_{\mathbf{g}^{[\widetilde{\jmath}:j]}} \\
&= \gamma^{[j]}\mathbf{b}^{[j]} + \sum_{\widetilde{\jmath}\in\widetilde{\mathbb{P}}^{[j]}} \mathbf{W}^{[\widetilde{\jmath}:j]\mathrm{T}}\boldsymbol{x}^{[\widetilde{\jmath}:j]} = \boldsymbol{x}^{[j]}
\end{aligned}
$$

For Kronecker-product nodes $\bigcirc^{[j]} = \bigotimes$ (so $\square^{[j]} = \bigotimes$, $\boxdot^{[j]} = \bigotimes$):

$$
\begin{aligned}
\left[\boldsymbol{\Psi}^{[j]}(\Theta), \phi^{[j]}(\boldsymbol{x})\right]_{\mathbf{g}^{[j]}} &= \gamma^{[j]}\mathbf{b}^{[j]} + \left\langle \underset{\widetilde{\jmath}\in\mathbb{A}^{[j]}}{\square^{[j]}} \frac{1}{\omega^{[\widetilde{\jmath}:j]}} \boldsymbol{\Psi}^{[\widetilde{\jmath}:j]}(\Theta)\, \mathbf{W}^{[\widetilde{\jmath}:j]}, \underset{\widetilde{\jmath}\in\mathbb{A}^{[j]}}{\boxdot^{[j]}} \omega^{[\widetilde{\jmath}:j]}\phi^{[\widetilde{\jmath}:j]}(\boldsymbol{x}) \right\rangle_{\underset{\widetilde{\jmath}\in\mathbb{A}^{[j]}}{\boxdot^{[j]}} \frac{1}{\omega^{[\widetilde{\jmath}:j]}} \mathbf{g}^{[\widetilde{\jmath}:j]}} \\
&= \gamma^{[j]}\mathbf{b}^{[j]} + \left\langle \underset{\widetilde{\jmath}\in\mathbb{A}^{[j]}}{\bigotimes} \frac{1}{\omega^{[\widetilde{\jmath}:j]}} \boldsymbol{\Psi}^{[\widetilde{\jmath}:j]}(\Theta)\mathbf{W}^{[\widetilde{\jmath}:j]}, \underset{\widetilde{\jmath}\in\mathbb{A}^{[j]}}{\bigotimes} \omega^{[\widetilde{\jmath}:j]}\phi^{[\widetilde{\jmath}:j]}(\boldsymbol{x}) \right\rangle_{\underset{\widetilde{\jmath}\in\mathbb{A}^{[j]}}{\bigotimes} \frac{1}{\omega^{[\widetilde{\jmath}:j]}} \mathbf{g}^{[\widetilde{\jmath}:j]}} \\
&= \gamma^{[j]}\mathbf{b}^{[j]} + \left( \underset{\widetilde{\jmath}\in\mathbb{A}^{[j]}}{\bigotimes} \frac{1}{\omega^{[\widetilde{\jmath}:j]}} \boldsymbol{\Psi}^{[\widetilde{\jmath}:j]}(\Theta)\mathbf{W}^{[\widetilde{\jmath}:j]} \right)^{\mathrm{T}} \left( \left( \underset{\widetilde{\jmath}\in\mathbb{A}^{[j]}}{\bigotimes} \phi^{[\widetilde{\jmath}:j]}(\boldsymbol{x}) \right) \odot \left( \underset{\widetilde{\jmath}\in\mathbb{A}^{[j]}}{\bigotimes} \mathbf{g}^{[\widetilde{\jmath}:j]} \right) \right) \\
&= \gamma^{[j]}\mathbf{b}^{[j]} + \underset{\widetilde{\jmath}\in\mathbb{A}^{[j]}}{\bigotimes} \left( \frac{1}{\omega^{[\widetilde{\jmath}:j]}} \boldsymbol{\Psi}^{[\widetilde{\jmath}:j]}(\Theta)\mathbf{W}^{[\widetilde{\jmath}:j]} \right)^{\mathrm{T}} \left( \phi^{[\widetilde{\jmath}:j]}(\boldsymbol{x}) \odot \mathbf{g}^{[\widetilde{\jmath}:j]} \right) \\
&= \gamma^{[j]}\mathbf{b}^{[j]} + \underset{\widetilde{\jmath}\in\mathbb{A}^{[j]}}{\bigotimes} \mathbf{W}^{[\widetilde{\jmath}:j]\mathrm{T}} \left\langle \boldsymbol{\Psi}^{[\widetilde{\jmath}:j]}(\Theta), \phi^{[\widetilde{\jmath}:j]}(\boldsymbol{x}) \right\rangle_{\mathbf{g}^{[\widetilde{\jmath}:j]}} \\
&= \gamma^{[j]}\mathbf{b}^{[j]} + \underset{\widetilde{\jmath}\in\widetilde{\mathbb{P}}^{[j]}}{\bigotimes} \mathbf{W}^{[\widetilde{\jmath}:j]\mathrm{T}}\boldsymbol{x}^{[\widetilde{\jmath}:j]} = \boldsymbol{x}^{[j]}
\end{aligned}
$$

For Hadamard product nodes $\bigcirc^{[j]} = \odot$ (so $\square^{[j]} = \bigotimes^{\updownarrow}$, $\boxdot^{[j]} = \bigotimes$):

$$\left\langle \boldsymbol{\Psi}^{[j]}\left(\Theta\right), \boldsymbol{\phi}^{[j]}\left(\boldsymbol{x}\right)\right]_{\mathbf{g}^{[j]}} = \gamma^{[j]}\mathbf{b}^{[j]} + \left\langle \underset{\widetilde{\jmath}\in\mathbb{A}^{[j]}}{\square^{[j]}} \tfrac{1}{\omega^{[\widetilde{\jmath};j]}} \boldsymbol{\Psi}^{[\widetilde{\jmath};j]}\left(\Theta\right)\mathbf{W}^{[\widetilde{\jmath};j]}, \underset{\widetilde{\jmath}\in\mathbb{A}^{[j]}}{\boxdot^{[j]}} \omega^{[\widetilde{\jmath};j]}\boldsymbol{\phi}^{[\widetilde{\jmath};j]}\left(\boldsymbol{x}\right)\right]_{\underset{\widetilde{\jmath}\in\mathbb{A}^{[j]}}{\boxdot^{[j]}} \tfrac{1}{\omega^{[\widetilde{\jmath};j]}}\mathbf{g}^{[\widetilde{\jmath};j]}}$$

$$= \gamma^{[j]}\mathbf{b}^{[j]} + \left\langle \underset{\widetilde{\jmath}\in\mathbb{A}^{[j]}}{\bigotimes^{\updownarrow}} \tfrac{1}{\omega^{[\widetilde{\jmath};j]}} \boldsymbol{\Psi}^{[\widetilde{\jmath};j]}\left(\Theta\right)\mathbf{W}^{[\widetilde{\jmath};j]}, \underset{\widetilde{\jmath}\in\mathbb{A}^{[j]}}{\bigotimes} \omega^{[\widetilde{\jmath};j]}\boldsymbol{\phi}^{[\widetilde{\jmath};j]}\left(\boldsymbol{x}\right)\right]_{\underset{\widetilde{\jmath}\in\mathbb{A}^{[j]}}{\bigotimes} \tfrac{1}{\omega^{[\widetilde{\jmath};j]}}\mathbf{g}^{[\widetilde{\jmath};j]}}$$

$$= \gamma^{[j]}\mathbf{b}^{[j]} + \left(\underset{\widetilde{\jmath}\in\mathbb{A}^{[j]}}{\bigotimes^{\updownarrow}} \tfrac{1}{\omega^{[\widetilde{\jmath};j]}} \boldsymbol{\Psi}^{[\widetilde{\jmath};j]}\left(\Theta\right)\mathbf{W}^{[\widetilde{\jmath};j]}\right)^{\mathrm{T}} \left(\left(\underset{\widetilde{\jmath}\in\mathbb{A}^{[j]}}{\bigotimes} \boldsymbol{\phi}^{[\widetilde{\jmath};j]}\left(\boldsymbol{x}\right)\right) \odot \left(\underset{\widetilde{\jmath}\in\mathbb{A}^{[j]}}{\bigotimes} \mathbf{g}^{[\widetilde{\jmath};j]}\right)\right)$$

$$= \gamma^{[j]}\mathbf{b}^{[j]} + \left[\left(\underset{\widetilde{\jmath}\in\mathbb{A}^{[j]}}{\bigotimes} \left(\boldsymbol{\Psi}^{[\widetilde{\jmath};j]}\left(\Theta\right)\mathbf{W}^{[\widetilde{\jmath};j]}_{:i_j}\right)\right)^{\mathrm{T}} \left(\underset{\widetilde{\jmath}\in\mathbb{A}^{[j]}}{\bigotimes} \left(\boldsymbol{\phi}^{[\widetilde{\jmath};j]}\left(\boldsymbol{x}\right) \odot \mathbf{g}^{[\widetilde{\jmath};j]}\right)\right)\right]_{i_j}$$

$$= \gamma^{[j]}\mathbf{b}^{[j]} + \left[\prod_{\widetilde{\jmath}\in\mathbb{A}^{[j]}} \left(\boldsymbol{\Psi}^{[\widetilde{\jmath};j]}\left(\Theta\right)\mathbf{W}^{[\widetilde{\jmath};j]}_{:i_j}\right)^{\mathrm{T}} \left(\boldsymbol{\phi}^{[\widetilde{\jmath};j]}\left(\boldsymbol{x}\right) \odot \mathbf{g}^{[\widetilde{\jmath};j]}\right)\right]_{i_j}$$

$$= \gamma^{[j]}\mathbf{b}^{[j]} + \left[\prod_{\widetilde{\jmath}\in\mathbb{A}^{[j]}} \mathbf{W}^{[\widetilde{\jmath};j]\mathrm{T}}_{:i_j}\boldsymbol{\Psi}^{[\widetilde{\jmath};j]}\left(\Theta\right)^{\mathrm{T}} \left(\boldsymbol{\phi}^{[\widetilde{\jmath};j]}\left(\boldsymbol{x}\right) \odot \mathbf{g}^{[\widetilde{\jmath};j]}\right)\right]_{i_j}$$

$$= \gamma^{[j]}\mathbf{b}^{[j]} + \left[\prod_{\widetilde{\jmath}\in\mathbb{A}^{[j]}} \mathbf{W}^{[\widetilde{\jmath};j]\mathrm{T}}_{:i_j}\left\langle \boldsymbol{\Psi}^{[\widetilde{\jmath};j]}\left(\Theta\right), \boldsymbol{\phi}^{[\widetilde{\jmath};j]}\left(\boldsymbol{x}\right)\right]_{\mathbf{g}^{[\widetilde{\jmath};j]}}\right]_{i_j}$$

$$= \gamma^{[j]}\mathbf{b}^{[j]} + \left[\prod_{\widetilde{\jmath}\in\mathbb{A}^{[j]}} \mathbf{W}^{[\widetilde{\jmath};j]\mathrm{T}}_{:i_j}\boldsymbol{x}^{[\widetilde{\jmath};j]}\right]_{i_j}$$

$$= \gamma^{[j]}\mathbf{b}^{[j]} + \underset{\widetilde{\jmath}\in\widetilde{\mathbb{P}}^{[j]}}{\odot} \mathbf{W}^{[\widetilde{\jmath};j]\mathrm{T}}\boldsymbol{x}^{[\widetilde{\jmath};j]} = \boldsymbol{x}^{[j]}$$

For multi-inner-product nodes $\bigcirc^{[j]} = \langle\!\langle\cdot\rangle\!\rangle$ (so $\square^{[j]} = \bigotimes^{\updownarrow}\left(\cdot\right)\mathbf{1}$, $\boxdot^{[j]} = \bigotimes$):

$$\left\langle \boldsymbol{\Psi}^{[j]}\left(\Theta\right), \boldsymbol{\phi}^{[j]}\left(\boldsymbol{x}\right)\right]_{\mathbf{g}^{[j]}} = \gamma^{[j]}\mathbf{b}^{[j]} + \left\langle \underset{\widetilde{\jmath}\in\mathbb{A}^{[j]}}{\square^{[j]}} \tfrac{1}{\omega^{[\widetilde{\jmath};j]}} \boldsymbol{\Psi}^{[\widetilde{\jmath};j]}\left(\Theta\right)\mathbf{W}^{[\widetilde{\jmath};j]}, \underset{\widetilde{\jmath}\in\mathbb{A}^{[j]}}{\boxdot^{[j]}} \omega^{[\widetilde{\jmath};j]}\boldsymbol{\phi}^{[\widetilde{\jmath};j]}\left(\boldsymbol{x}\right)\right]_{\underset{\widetilde{\jmath}\in\mathbb{A}^{[j]}}{\boxdot^{[j]}} \tfrac{1}{\omega^{[\widetilde{\jmath};j]}}\mathbf{g}^{[\widetilde{\jmath};j]}}$$

$$= \gamma^{[j]}\mathbf{b}^{[j]} + \left\langle \underset{\widetilde{\jmath}\in\mathbb{A}^{[j]}}{\bigotimes^{\updownarrow}} \tfrac{1}{\omega^{[\widetilde{\jmath};j]}} \boldsymbol{\Psi}^{[\widetilde{\jmath};j]}\left(\Theta\right)\mathbf{W}^{[\widetilde{\jmath};j]}\mathbf{1}, \underset{\widetilde{\jmath}\in\mathbb{A}^{[j]}}{\bigotimes} \omega^{[\widetilde{\jmath};j]}\boldsymbol{\phi}^{[\widetilde{\jmath};j]}\left(\boldsymbol{x}\right)\right]_{\underset{\widetilde{\jmath}\in\mathbb{A}^{[j]}}{\bigotimes} \tfrac{1}{\omega^{[\widetilde{\jmath};j]}}\mathbf{g}^{[\widetilde{\jmath};j]}}$$

$$= \gamma^{[j]}\mathbf{b}^{[j]} + \left(\underset{\widetilde{\jmath}\in\mathbb{A}^{[j]}}{\bigotimes^{\updownarrow}} \tfrac{1}{\omega^{[\widetilde{\jmath};j]}} \boldsymbol{\Psi}^{[\widetilde{\jmath};j]}\left(\Theta\right)\mathbf{W}^{[\widetilde{\jmath};j]}\mathbf{1}\right)^{\mathrm{T}} \left(\left(\underset{\widetilde{\jmath}\in\mathbb{A}^{[j]}}{\bigotimes} \boldsymbol{\phi}^{[\widetilde{\jmath};j]}\left(\boldsymbol{x}\right)\right) \odot \left(\underset{\widetilde{\jmath}\in\mathbb{A}^{[j]}}{\bigotimes} \mathbf{g}^{[\widetilde{\jmath};j]}\right)\right)$$

$$= \gamma^{[j]}\mathbf{b}^{[j]} + \mathbf{1}^{\mathrm{T}}\left[\left(\underset{\widetilde{\jmath}\in\mathbb{A}^{[j]}}{\bigotimes} \left(\boldsymbol{\Psi}^{[\widetilde{\jmath};j]}\left(\Theta\right)\mathbf{W}^{[\widetilde{\jmath};j]}_{:i_j}\right)\right)^{\mathrm{T}} \left(\underset{\widetilde{\jmath}\in\mathbb{A}^{[j]}}{\bigotimes} \left(\boldsymbol{\phi}^{[\widetilde{\jmath};j]}\left(\boldsymbol{x}\right) \odot \mathbf{g}^{[\widetilde{\jmath};j]}\right)\right)\right]_{i_j}$$

$$= \gamma^{[j]}\mathbf{b}^{[j]} + \mathbf{1}^{\mathrm{T}}\left[\prod_{\widetilde{\jmath}\in\mathbb{A}^{[j]}} \left(\boldsymbol{\Psi}^{[\widetilde{\jmath};j]}\left(\Theta\right)\mathbf{W}^{[\widetilde{\jmath};j]}_{:i_j}\right)^{\mathrm{T}} \left(\boldsymbol{\phi}^{[\widetilde{\jmath};j]}\left(\boldsymbol{x}\right) \odot \mathbf{g}^{[\widetilde{\jmath};j]}\right)\right]_{i_j}$$

$$= \gamma^{[j]}\mathbf{b}^{[j]} + \mathbf{1}^{\mathrm{T}}\left[\prod_{\widetilde{\jmath}\in\mathbb{A}^{[j]}} \mathbf{W}^{[\widetilde{\jmath};j]\mathrm{T}}_{:i_j}\boldsymbol{\Psi}^{[\widetilde{\jmath};j]}\left(\Theta\right)^{\mathrm{T}} \left(\boldsymbol{\phi}^{[\widetilde{\jmath};j]}\left(\boldsymbol{x}\right) \odot \mathbf{g}^{[\widetilde{\jmath};j]}\right)\right]_{i_j}$$

$$= \gamma^{[j]}\mathbf{b}^{[j]} + \mathbf{1}^{\mathrm{T}}\left[\prod_{\widetilde{\jmath}\in\mathbb{A}^{[j]}} \mathbf{W}^{[\widetilde{\jmath};j]\mathrm{T}}_{:i_j}\left\langle \boldsymbol{\Psi}^{[\widetilde{\jmath};j]}\left(\Theta\right), \boldsymbol{\phi}^{[\widetilde{\jmath};j]}\left(\boldsymbol{x}\right)\right]_{\mathbf{g}^{[\widetilde{\jmath};j]}}\right]_{i_j}$$

$$= \gamma^{[j]}\mathbf{b}^{[j]} + \mathbf{1}^{\mathrm{T}}\left[\prod_{\widetilde{\jmath}\in\mathbb{A}^{[j]}} \mathbf{W}^{[\widetilde{\jmath};j]\mathrm{T}}_{:i_j}\boldsymbol{x}^{[\widetilde{\jmath};j]}\right]_{i_j}$$

$$= \gamma^{[j]}\mathbf{b}^{[j]} + \mathbf{1}^{\mathrm{T}}\underset{\widetilde{\jmath}\in\widetilde{\mathbb{P}}^{[j]}}{\odot} \mathbf{W}^{[\widetilde{\jmath};j]\mathrm{T}}\boldsymbol{x}^{[\widetilde{\jmath};j]} = \boldsymbol{x}^{[j]}$$

$$= \gamma^{[j]}\mathbf{b}^{[j]} + \underset{\widetilde{\jmath}\in\widetilde{\mathbb{P}}^{[j]}}{\langle\!\langle\cdot\rangle\!\rangle} \mathbf{W}^{[\widetilde{\jmath};j]\mathrm{T}}\boldsymbol{x}^{[\widetilde{\jmath};j]} = \boldsymbol{x}^{[j]}$$

where on the final line we use $\langle\!\langle\cdot\rangle\!\rangle$ as an operator (see notation section). So, in all cases:

$$\left\langle \boldsymbol{\Psi}^{[j]}\left(\Theta\right), \boldsymbol{\phi}^{[j]}\left(\boldsymbol{x}\right)\right]_{\mathbf{g}^{[j]}} = \gamma\mathbf{b}^{[j]} + \underset{\widetilde{\jmath}\in\widetilde{\mathbb{P}}^{[j]}}{\bigcirc^{[j]}} \mathbf{W}^{[\widetilde{\jmath};j]\mathrm{T}}\boldsymbol{x}^{[\widetilde{\jmath};j]} = \boldsymbol{x}^{[j]}$$

which is the desired result (28) for the node case.

## C.2    Proof of Theorem 1 - Norm-Bounds

Recalling that the network is arranged in layers $\jmath = 0, 1, 2, \ldots, D$, and noting that for the input layer $\jmath = 0$ is, tivially from our assumptions:

$$\left\| \boldsymbol{\Psi}^{[0]}\left(\Theta\right)\right\|_F = \psi_\eta^{[0]}$$
$$\phi_{\downarrow\eta}^{[0]} \leq \left\| \boldsymbol{\phi}^{[0]}\left(\boldsymbol{x}\right)\right\|_F = \phi_\eta^{[0]}$$

where $\psi_\eta^{[0]} = r$, $\phi_{\downarrow\eta}^{[0]} = \rho$ and $\phi_\eta^{[0]} = r$, it suffices to show that *if* all outputs of all nodes $\widetilde{\jmath} \in \mathbb{L}^{[\jmath-1]}$ in layer $\jmath - 1$ satisfy:

$$\left\| \boldsymbol{\Psi}^{[\widetilde{\jmath}]}\left(\Theta\right)\right\|_F = \psi_\eta^{[\widetilde{\jmath}]}$$
$$\phi_{\downarrow\eta}^{[\widetilde{\jmath}]} \leq \left\| \boldsymbol{\phi}^{[\widetilde{\jmath}]}\left(\boldsymbol{x}\right)\right\|_F = \phi_\eta^{[\widetilde{\jmath}]} \tag{29}$$

*then* all nodes $j \in \mathbb{L}^{[\jmath]}$, using the definitions given, satisfy:

$$\left\| \boldsymbol{\Psi}^{[\widetilde{\jmath};j]}\left(\Theta\right)\right\|_F = \psi_\eta^{[\widetilde{\jmath};j]}$$
$$\phi_{\downarrow\eta}^{[\widetilde{\jmath};j]} \leq \left\| \boldsymbol{\phi}^{[\widetilde{\jmath};j]}\left(\boldsymbol{x}\right)\right\|_F = \phi_\eta^{[\widetilde{\jmath};j]} \qquad \forall \widetilde{\jmath} \in \mathbb{A}^{[j]} \tag{30}$$

and:

$$\left\| \boldsymbol{\Psi}^{[j]}\left(\Theta\right)\right\|_F = \psi_\eta^{[j]}$$
$$\phi_{\downarrow\eta}^{[j]} \leq \left\| \boldsymbol{\phi}^{[j]}\left(\boldsymbol{x}\right)\right\|_F = \phi_\eta^{[j]} \tag{31}$$

We call (30) the edge case and (31) the node case, and will treat them separately.

**Edge case:** We are given that (29) is correct. By direct calculation, for incoming edges (21a), using the multinomial theorem at step $(*)$:

$$
\begin{aligned}
\left\|\boldsymbol{\Psi}^{[\widetilde{\jmath};j]}(\Theta)\right\|_F^2 &= \frac{L_{\psi_\eta^{[\widetilde{\jmath}]}}^{[\widetilde{\jmath};j]2}}{s_\eta^{[\widetilde{\jmath};j]}(1)} \left( \eta^2 \left\|\boldsymbol{\tau}^{[\widetilde{\jmath};j]}(\boldsymbol{0})\right\|_2^2 + \left\| \left[ \operatorname{diag}\left(\mathbf{a}_{\mathbf{k}}^{[\widetilde{\jmath};j]\langle\odot\frac{1}{2}\rangle}\right) \otimes^{\updownarrow} \left[ \binom{\mathbf{k}}{\mathbf{1}}^{\frac{1}{2}} \bigotimes_{\widetilde{i}\in\mathbb{N}_{H[\widetilde{\jmath}]}} \left(\sqrt{\eta}\boldsymbol{\Psi}_{:\widetilde{i}}^{[\widetilde{\jmath}]}(\Theta)\right)^{\otimes l_{\widetilde{i}}} \right]_{\substack{1\succ_{H[\widetilde{\jmath}]}0,\\1\leq\mathbf{k}}} \right]_{\mathbf{k}\succ_{H[\widetilde{\jmath}]}0} \right\|_F^2 \right) \\
&= \frac{L_{\psi_\eta^{[\widetilde{\jmath}]}}^{[\widetilde{\jmath};j]2}}{s_\eta^{[\widetilde{\jmath};j]}(1)} \left( \eta^2 \left\|\boldsymbol{\tau}^{[\widetilde{\jmath};j]}(\boldsymbol{0})\right\|_2^2 + \sum_{\mathbf{k}\succ_{H[\widetilde{\jmath}]}0} \left\|\mathbf{a}_{\mathbf{k}}^{[\widetilde{\jmath};j]}\right\|_1 \left\| \left[ \binom{\mathbf{k}}{\mathbf{1}}^{\frac{1}{2}} \bigotimes_{\widetilde{i}\in\mathbb{N}_{H[\widetilde{\jmath}]}} \left(\sqrt{\eta}\boldsymbol{\Psi}_{:\widetilde{i}}^{[\widetilde{\jmath}]}(\Theta)\right)^{\otimes l_{\widetilde{i}}} \right]_{\substack{1\succ_{H[\widetilde{\jmath}]}0,\\1\leq\mathbf{k}}} \right\|_2^2 \right) \\
&= \frac{L_{\psi_\eta^{[\widetilde{\jmath}]}}^{[\widetilde{\jmath};j]2}}{s_\eta^{[\widetilde{\jmath};j]}(1)} \left( \eta^2 \left\|\boldsymbol{\tau}^{[\widetilde{\jmath};j]}(\boldsymbol{0})\right\|_2^2 + \sum_{\mathbf{k}\succ_{H[\widetilde{\jmath}]}0} \left\|\mathbf{a}_{\mathbf{k}}^{[\widetilde{\jmath};j]}\right\|_1 \sum_{0\prec_{H[\widetilde{\jmath}]}1\leq\mathbf{k}} \binom{\mathbf{k}}{\mathbf{1}} \left\| \bigotimes_{\widetilde{i}\in\mathbb{N}_{H[\widetilde{\jmath}]}} \left(\sqrt{\eta}\boldsymbol{\Psi}_{:\widetilde{i}}^{[\widetilde{\jmath}]}(\Theta)\right)^{\otimes l_{\widetilde{i}}} \right\|_2^2 \right) \\
&= \frac{L_{\psi_\eta^{[\widetilde{\jmath}]}}^{[\widetilde{\jmath};j]2}}{s_\eta^{[\widetilde{\jmath};j]}(1)} \left( \eta^2 \left\|\boldsymbol{\tau}^{[\widetilde{\jmath};j]}(\boldsymbol{0})\right\|_2^2 + \sum_{\mathbf{k}\succ_{H[\widetilde{\jmath}]}0} \left\|\mathbf{a}_{\mathbf{k}}^{[\widetilde{\jmath};j]}\right\|_1 \sum_{0\prec_{H[\widetilde{\jmath}]}1\leq\mathbf{k}} \binom{\mathbf{k}}{\mathbf{1}} \prod_{\widetilde{i}\in\mathbb{N}_{H[\widetilde{\jmath}]}} \left(\sqrt{\eta}\left\|\boldsymbol{\Psi}_{:\widetilde{i}}^{[\widetilde{\jmath}]}(\Theta)\right\|_2\right)^{2l_{\widetilde{i}}} \right) \\
&= \frac{L_{\psi_\eta^{[\widetilde{\jmath}]}}^{[\widetilde{\jmath};j]}}{s_\eta^{[\widetilde{\jmath};j]}(1)} \left( \eta^2 \left\|\boldsymbol{\tau}^{[\widetilde{\jmath};j]}(\boldsymbol{0})\right\|_2^2 + \sum_{\mathbf{k}\succ_{H[\widetilde{\jmath}]}0} \left\|\mathbf{a}_{\mathbf{k}}^{[\widetilde{\jmath};j]}\right\|_1 \left( \left( \sum_{0\preceq_{H[\widetilde{\jmath}]}1\leq\mathbf{k}} \binom{\mathbf{k}}{\mathbf{1}} 1^{|\mathbf{k}|-|\mathbf{1}|} \prod_{\widetilde{i}\in\mathbb{N}_{H[\widetilde{\jmath}]}} \left(\eta\left\|\boldsymbol{\Psi}_{:\widetilde{i}}^{[\widetilde{\jmath}]}(\Theta)\right\|_2^2\right)^{l_{\widetilde{i}}} \right) - 1 \right) \right) \\
&\stackrel{(*)}{=} \frac{L_{\psi_\eta^{[\widetilde{\jmath}]}}^{[\widetilde{\jmath};j]2}}{s_\eta^{[\widetilde{\jmath};j]}(1)} \left( \eta^2 \left\|\boldsymbol{\tau}^{[\widetilde{\jmath};j]}(\boldsymbol{0})\right\|_2^2 + \sum_{\mathbf{k}\succ_{H[\widetilde{\jmath}]}0} \left\|\mathbf{a}_{\mathbf{k}}^{[\widetilde{\jmath};j]}\right\|_1 \left( \left(1+\sum_{\widetilde{i}\in\mathbb{N}_{H[\widetilde{\jmath}]}} \eta\left\|\boldsymbol{\Psi}_{:\widetilde{i}}^{[\widetilde{\jmath}]}(\Theta)\right\|_2^2\right)^{|\mathbf{k}|} - 1 \right) \right) \\
&= \frac{L_{\psi_\eta^{[\widetilde{\jmath}]}}^{[\widetilde{\jmath};j]2}}{s_\eta^{[\widetilde{\jmath};j]}(1)} \left( \eta^2 \left\|\boldsymbol{\tau}^{[\widetilde{\jmath};j]}(\boldsymbol{0})\right\|_2^2 + \sum_{\mathbf{k}\succ_{H[\widetilde{\jmath}]}0} \left\|\mathbf{a}_{\mathbf{k}}^{[\widetilde{\jmath};j]}\right\|_1 \left( \left(1+\eta\left\|\boldsymbol{\Psi}^{[\widetilde{\jmath}]}(\Theta)\right\|_F^2\right)^{|\mathbf{k}|} - 1 \right) \right) \\
&= L_{\psi_\eta^{[\widetilde{\jmath}]}}^{[\widetilde{\jmath};j]2} \frac{s_\eta^{[\widetilde{\jmath};j]}\left(\left\|\boldsymbol{\Psi}^{[\widetilde{\jmath}]}(\Theta)\right\|_F^2\right)}{s_\eta^{[\widetilde{\jmath};j]}(1)}
\end{aligned}
$$

and:

$$\left\|\boldsymbol{\phi}^{[\widetilde{\jmath};j]}(\boldsymbol{x})\right\|_2^2 = \frac{L_{\phi_\eta^{[\widetilde{\jmath}]}}^{[\widetilde{\jmath};j]2}}{s_\eta^{[\widetilde{\jmath};j]}(1)}\left(\eta^2\left\|\boldsymbol{\tau}^{[\widetilde{\jmath};j]}(\mathbf{0})\right\|_2^2 + \left\|\left[\left.\left|\mathbf{a}_{\mathrm{k}}^{[\widetilde{\jmath};j]}\right|^{\odot\frac{1}{2}}\otimes\left[\binom{\mathrm{k}}{1}^{\frac{1}{2}}\left(\sqrt{\eta}\boldsymbol{\phi}^{[\widetilde{\jmath}]}(\boldsymbol{x})\right)^{\otimes|1|}\right]_{\substack{1\succ_{H[\widetilde{\jmath}]}0,\\1\leq\mathrm{k}}}\right]_{\mathrm{k}\succ_{H[\widetilde{\jmath}]}0}\right\|_2^2\right)$$

$$= \frac{L_{\phi_\eta^{[\widetilde{\jmath}]}}^{[\widetilde{\jmath};j]2}}{s_\eta^{[\widetilde{\jmath};j]}(1)}\left(\eta^2\left\|\boldsymbol{\tau}^{[\widetilde{\jmath};j]}(\mathbf{0})\right\|_2^2 + \sum_{\mathrm{k}\succ_{H[\widetilde{\jmath}]}0}\left\|\mathbf{a}_{\mathrm{k}}^{[\widetilde{\jmath};j]}\right\|_1\left\|\left[\binom{\mathrm{k}}{1}^{\frac{1}{2}}\left(\sqrt{\eta}\boldsymbol{\phi}^{[\widetilde{\jmath}]}(\boldsymbol{x})\right)^{\otimes|1|}\right]_{\substack{1\succ_{H[\widetilde{\jmath}]}0,\\1\leq\mathrm{k}}}\right\|_2^2\right)$$

$$= \frac{L_{\phi_\eta^{[\widetilde{\jmath}]}}^{[\widetilde{\jmath};j]2}}{s_\eta^{[\widetilde{\jmath};j]}(1)}\left(\eta^2\left\|\boldsymbol{\tau}^{[\widetilde{\jmath};j]}(\mathbf{0})\right\|_2^2 + \sum_{\mathrm{k}\succ_{H[\widetilde{\jmath}]}0}\left\|\mathbf{a}_{\mathrm{k}}^{[\widetilde{\jmath};j]}\right\|_1\sum_{0\prec_{H[\widetilde{\jmath}]}1\leq\mathrm{k}}\binom{\mathrm{k}}{1}\left\|\left(\sqrt{\eta}\boldsymbol{\phi}^{[\widetilde{\jmath}]}(\boldsymbol{x})\right)^{\otimes|1|}\right\|_2^2\right)$$

$$= \frac{L_{\phi_\eta^{[\widetilde{\jmath}]}}^{[\widetilde{\jmath};j]2}}{s_\eta^{[\widetilde{\jmath};j]}(1)}\left(\eta^2\left\|\boldsymbol{\tau}^{[\widetilde{\jmath};j]}(\mathbf{0})\right\|_2^2 + \sum_{\mathrm{k}\succ_{H[\widetilde{\jmath}]}0}\left\|\mathbf{a}_{\mathrm{k}}^{[\widetilde{\jmath};j]}\right\|_1\sum_{0\prec_{H[\widetilde{\jmath}]}1\leq\mathrm{k}}\binom{\mathrm{k}}{1}\left(\sqrt{\eta}\left\|\boldsymbol{\phi}^{[\widetilde{\jmath}]}(\boldsymbol{x})\right\|_2\right)^{2|1|}\right)$$

$$=^{(*)} \frac{L_{\phi_\eta^{[\widetilde{\jmath}]}}^{[\widetilde{\jmath};j]2}}{s_\eta^{[\widetilde{\jmath};j]}(1)}\left(\eta^2\left\|\boldsymbol{\tau}^{[\widetilde{\jmath};j]}(\mathbf{0})\right\|_2^2 + \sum_{\mathrm{k}\succ_{H[\widetilde{\jmath}]}0}\left\|\mathbf{a}_{\mathrm{k}}^{[\widetilde{\jmath};j]}\right\|_1\left(\left(1+\eta\left\|\boldsymbol{\phi}^{[\widetilde{\jmath}]}(\boldsymbol{x})\right\|_2^2\right)^{|\mathrm{k}|}-1\right)\right)$$

$$= L_{\phi_\eta^{[\widetilde{\jmath}]}}^{[\widetilde{\jmath};j]2}\frac{s_\eta^{[\widetilde{\jmath};j]}\left(\left\|\boldsymbol{\phi}^{[\widetilde{\jmath}]}(\boldsymbol{x})\right\|_2^2\right)}{s_\eta^{[\widetilde{\jmath};j]}(1)}$$

which we may bound as (using that $s_\eta^{[\widetilde{\jmath};j]}$ is increasing on $\mathbb{R}^+$):

$$\left\|\boldsymbol{\Psi}^{[\widetilde{\jmath};j]}(\Theta)\right\|_F^2 = L_{\psi_\eta^{[\widetilde{\jmath}]}}^{[\widetilde{\jmath};j]2}\frac{s_\eta^{[\widetilde{\jmath};j]}\left(\left\|\boldsymbol{\Psi}^{[\widetilde{\jmath}]}(\Theta)\right\|_F^2\right)}{s_\eta^{[\widetilde{\jmath};j]}(1)} \leq \psi_\eta^{[\widetilde{\jmath};j]2}$$

$$\left\|\boldsymbol{\phi}^{[\widetilde{\jmath};j]}(\boldsymbol{x})\right\|_2^2 = L_{\phi_\eta^{[\widetilde{\jmath}]}}^{[\widetilde{\jmath};j]2}\frac{s_\eta^{[\widetilde{\jmath};j]}\left(\left\|\boldsymbol{\phi}^{[\widetilde{\jmath}]}(\boldsymbol{x})\right\|_2^2\right)}{s_\eta^{[\widetilde{\jmath};j]}(1)} \in \left[\phi_{\downarrow\eta}^{[\widetilde{\jmath};j]}, \phi_\eta^{[\widetilde{\jmath};j]2}\right]$$

which is the desired result (30).

**Node case:** We have shown that (30) is correct. For columnar concatenation nodes $\bigcirc^{[j]} = \bigoplus$ (so $\square^{[j]} = \mathrm{diag}$, $\boxdot^{[j]} = \bigoplus$):

$$\left\|\boldsymbol{\Psi}^{[j]}(\Theta)\right\|_F^2 = \left\|\mathbf{b}^{[j]}\right\|_2^2 + \sum_{\widetilde{\jmath}\in\mathbb{A}^{[j]}}L_{\psi_\eta^{[\widetilde{\jmath}]}}^{[\widetilde{\jmath};j]2}\frac{s_\eta^{[\widetilde{\jmath};j]}\left(\left\|\boldsymbol{\Psi}^{[\widetilde{\jmath}]}(\Theta)\right\|_F^2\right)}{s_\eta^{[\widetilde{\jmath};j]}(1)}\left\|\mathbf{W}^{[\widetilde{\jmath};j]}\right\|_2^2$$

$$\left\|\boldsymbol{\phi}^{[j]}(\boldsymbol{x})\right\|_2^2 = \beta^{[j]2} + \sum_{\widetilde{\jmath}\in\mathbb{A}^{[j]}}L_{\phi_\eta^{[\widetilde{\jmath}]}}^{[\widetilde{\jmath};j]2}\frac{s_\eta^{[\widetilde{\jmath};j]}\left(\left\|\boldsymbol{\phi}^{[\widetilde{\jmath}]}(\boldsymbol{x})\right\|_2^2\right)}{s_\eta^{[\widetilde{\jmath};j]}(1)}\omega^{[\widetilde{\jmath};j]2}$$

For additive nodes $\bigcirc^{[j]} = \sum$ (so $\square^{[j]} = \bigoplus$, $\boxdot^{[j]} = \bigoplus$):

$$\left\|\boldsymbol{\Psi}^{[j]}(\Theta)\right\|_F^2 = \left\|\mathbf{b}^{[j]}\right\|_2^2 + \sum_{\widetilde{\jmath}\in\mathbb{A}^{[j]}}L_{\psi_\eta^{[\widetilde{\jmath}]}}^{[\widetilde{\jmath};j]2}\frac{s_\eta^{[\widetilde{\jmath};j]}\left(\left\|\boldsymbol{\Psi}^{[\widetilde{\jmath}]}(\Theta)\right\|_F^2\right)}{s_\eta^{[\widetilde{\jmath};j]}(1)}\left\|\mathbf{W}^{[\widetilde{\jmath};j]}\right\|_2^2$$

$$\left\|\boldsymbol{\phi}^{[j]}(\boldsymbol{x})\right\|_2^2 = \beta^{[j]2} + \sum_{\widetilde{\jmath}\in\mathbb{A}^{[j]}}L_{\phi_\eta^{[\widetilde{\jmath}]}}^{[\widetilde{\jmath};j]2}\frac{s_\eta^{[\widetilde{\jmath};j]}\left(\left\|\boldsymbol{\phi}^{[\widetilde{\jmath}]}(\boldsymbol{x})\right\|_2^2\right)}{s_\eta^{[\widetilde{\jmath};j]}(1)}\omega^{[\widetilde{\jmath};j]2}$$

For Kronecker-product nodes $\bigcirc^{[j]} = \bigotimes$ (so $\square^{[j]} = \bigotimes$, $\boxdot^{[j]} = \bigotimes$):

$$\left\|\boldsymbol{\Psi}^{[j]}(\Theta)\right\|_F^2 = \left\|\mathbf{b}^{[j]}\right\|_2^2 + \prod_{\widetilde{\jmath}\in\mathbb{A}^{[j]}}L_{\psi_\eta^{[\widetilde{\jmath}]}}^{[\widetilde{\jmath};j]2}\frac{s_\eta^{[\widetilde{\jmath};j]}\left(\left\|\boldsymbol{\Psi}^{[\widetilde{\jmath}]}(\Theta)\right\|_F^2\right)}{s_\eta^{[\widetilde{\jmath};j]}(1)}\left\|\mathbf{W}^{[\widetilde{\jmath};j]}\right\|_2^2$$

$$\left\|\boldsymbol{\phi}^{[j]}(\boldsymbol{x})\right\|_2^2 = \beta^{[j]2} + \prod_{\widetilde{\jmath}\in\mathbb{A}^{[j]}}L_{\phi_\eta^{[\widetilde{\jmath}]}}^{[\widetilde{\jmath};j]2}\frac{s_\eta^{[\widetilde{\jmath};j]}\left(\left\|\boldsymbol{\phi}^{[\widetilde{\jmath}]}(\boldsymbol{x})\right\|_2^2\right)}{s_\eta^{[\widetilde{\jmath};j]}(1)}\omega^{[\widetilde{\jmath};j]2}$$

For Hadamard product nodes $\bigcirc^{[j]} = \bigodot$ (so $\square^{[j]} = \bigotimes^{\ddagger}$, $\boxdot^{[j]} = \bigotimes$):

$$\left\|\boldsymbol{\Psi}^{[j]}(\Theta)\right\|_F^2 \leq \left\|\mathbf{b}^{[j]}\right\|_2^2 + \prod_{\widetilde{\jmath}\in\mathbb{A}^{[j]}}L_{\psi_\eta^{[\widetilde{\jmath}]}}^{[\widetilde{\jmath};j]2}\frac{s_\eta^{[\widetilde{\jmath};j]}\left(\left\|\boldsymbol{\Psi}^{[\widetilde{\jmath}]}(\Theta)\right\|_F^2\right)}{s_\eta^{[\widetilde{\jmath};j]}(1)}\left\|\mathbf{W}^{[\widetilde{\jmath};j]}\right\|_2^2$$

$$\left\|\boldsymbol{\phi}^{[j]}(\boldsymbol{x})\right\|_2^2 = \beta^{[j]2} + \prod_{\widetilde{\jmath}\in\mathbb{A}^{[j]}}L_{\phi_\eta^{[\widetilde{\jmath}]}}^{[\widetilde{\jmath};j]2}\frac{s_\eta^{[\widetilde{\jmath};j]}\left(\left\|\boldsymbol{\phi}^{[\widetilde{\jmath}]}(\boldsymbol{x})\right\|_2^2\right)}{s_\eta^{[\widetilde{\jmath};j]}(1)}\omega^{[\widetilde{\jmath};j]2}$$

For multi-inner-product nodes $\bigcirc^{[j]} = \langle\!\langle \cdot \rangle\!\rangle$ (so $\square^{[j]} = \bigotimes^{\updownarrow}(\cdot)\,\mathbf{1}$, $\boxdot^{[j]} = \bigotimes$):

$$\left\|\mathbf{\Psi}^{[j]}\left(\Theta\right)\right\|_F^2 \leq \left\|\mathbf{b}^{[j]}\right\|_2^2 + \prod_{\widetilde{\jmath}\in\mathbb{A}^{[j]}} L_{\psi_\eta^{[\widetilde{\jmath}]}}^{[\widetilde{\jmath}:j]2}\frac{s_\eta^{[\widetilde{\jmath}:j]}\left(\left\|\mathbf{\Psi}^{[\widetilde{\jmath}]}(\Theta)\right\|_F^2\right)}{s_\eta^{[\widetilde{\jmath}:j]}(1)}\left\|\mathbf{W}^{[\widetilde{\jmath}:j]}\right\|_2^2$$

$$\left\|\boldsymbol{\phi}^{[j]}\left(\boldsymbol{x}\right)\right\|_2^2 = \beta^{[j]2} + \prod_{\widetilde{\jmath}\in\mathbb{A}^{[j]}} L_{\phi_\eta^{[\widetilde{\jmath}]}}^{[\widetilde{\jmath}:j]2}\frac{s_\eta^{[\widetilde{\jmath}:j]}\left(\left\|\boldsymbol{\phi}^{[\widetilde{\jmath}]}(\boldsymbol{x})\right\|_2^2\right)}{s_\eta^{[\widetilde{\jmath}:j]}(1)}\omega^{[\widetilde{\jmath}:j]2}$$

Thus in general, for all nodes considered here:

$$\left\|\mathbf{\Psi}^{[j]}\left(\Theta\right)\right\|_F^2 \leq \left\|\mathbf{b}^{[j]}\right\|_2^2 + \prod_{\widetilde{\jmath}\in\mathbb{A}^{[j]}} L_{\psi_\eta^{[\widetilde{\jmath}]}}^{[\widetilde{\jmath}:j]2}\frac{s_\eta^{[\widetilde{\jmath}:j]}\left(\left\|\mathbf{\Psi}^{[\widetilde{\jmath}]}(\Theta)\right\|_F^2\right)}{s_\eta^{[\widetilde{\jmath}:j]}(1)}\left\|\mathbf{W}^{[\widetilde{\jmath}:j]}\right\|_2^2$$

$$\left\|\boldsymbol{\phi}^{[j]}\left(\boldsymbol{x}\right)\right\|_2^2 = \beta^{[j]2} + \prod_{\widetilde{\jmath}\in\mathbb{A}^{[j]}} L_{\phi_\eta^{[\widetilde{\jmath}]}}^{[\widetilde{\jmath}:j]2}\frac{s_\eta^{[\widetilde{\jmath}:j]}\left(\left\|\boldsymbol{\phi}^{[\widetilde{\jmath}]}(\boldsymbol{x})\right\|_2^2\right)}{s_\eta^{[\widetilde{\jmath}:j]}(1)}\omega^{[\widetilde{\jmath}:j]2}$$

which we may bound as:

$$\left\|\mathbf{\Psi}^{[j]}\left(\Theta\right)\right\|_F^2 \leq \psi_\eta^{[j]2}$$
$$\left\|\boldsymbol{\phi}^{[j]}\left(\boldsymbol{x}\right)\right\|_2^2 \in \left[\phi_{\downarrow\eta}^{[j]2}, \phi_\eta^{[j]2}\right]$$

which is the desired result (31) for the node case.

We observe that the data-feature-map bound is tight:

$$\left\|\boldsymbol{\phi}\left(\boldsymbol{x}\right)\right\|_2^2 = \phi_{\downarrow\eta}^2 \text{ if } \left\|\boldsymbol{x}\right\|_2 = \rho, \text{ and } \phi_{\downarrow\eta}^2 > 0 \text{ if } \rho > 0$$

$$\left\|\boldsymbol{\phi}\left(\boldsymbol{x}\right)\right\|_2^2 = \phi_\eta^2 \text{ if } \left\|\boldsymbol{x}\right\|_2 = r$$

In the limit $\eta \to 0$, identifying $\psi^{[j]} = \psi_{0+}^{[j]}$, $\phi_\downarrow^{[j]} = \phi_{\downarrow 0+}^{[j]}$, $\phi^{[j]} = \phi_{0+}^{[j]}$; $\psi = \psi_{0+}$, $\phi_\downarrow = \phi_{\downarrow 0+}$, $\phi = \phi_{0+}$; $\psi = \psi^{[E]}$, $\phi_\downarrow = \phi_\downarrow^{[E]}$, $\phi = \phi^{[E]}$; where, recursively $\forall j \in \mathbb{Z}_E$:

$$\begin{aligned}
\psi^{[j]2} &\triangleq \beta^{[j]2} + \bigodot_{\widetilde{\jmath}\in\mathbb{A}^{[j]}}^{[j]} \omega^{[\widetilde{\jmath}:j]2} \begin{cases} L_{\psi^{[\widetilde{\jmath}]}}^{[\widetilde{\jmath}:j]2}\psi^{[\widetilde{\jmath}]2} & \text{if } \tau^{[\widetilde{\jmath}:j]} \text{ Lipschitz} \\ B_{\psi^{[\widetilde{\jmath}]}}^{[\widetilde{\jmath}:j]2}\frac{\psi^{[\widetilde{\jmath}]2}}{\phi_\downarrow^{[\widetilde{\jmath}]2}} & \text{otherwise} \end{cases} \\
\phi_\downarrow^{[j]2} &\triangleq \beta^{[j]2} + \bigodot_{\widetilde{\jmath}\in\mathbb{A}^{[j]}} \omega^{[\widetilde{\jmath}:j]2} \begin{cases} L_{\psi^{[\widetilde{\jmath}]}}^{[\widetilde{\jmath}:j]2}\phi_\downarrow^{[\widetilde{\jmath}]2} & \text{if } \tau^{[\widetilde{\jmath}:j]} \text{ Lipschitz} \\ B_{\psi^{[\widetilde{\jmath}]}}^{[\widetilde{\jmath}:j]2} & \text{otherwise} \end{cases} \\
\phi^{[j]2} &\triangleq \beta^{[j]2} + \bigodot_{\widetilde{\jmath}\in\mathbb{A}^{[j]}} \omega^{[\widetilde{\jmath}:j]2} \begin{cases} L_{\phi^{[\widetilde{\jmath}]}}^{[\widetilde{\jmath}:j]2}\phi^{[\widetilde{\jmath}]2} & \text{if } \tau^{[\widetilde{\jmath}:j]} \text{ Lipschitz} \\ B_{\phi^{[\widetilde{\jmath}]}}^{[\widetilde{\jmath}:j]2}\frac{\phi^{[\widetilde{\jmath}]2}}{\phi_\downarrow^{[\widetilde{\jmath}]2}} & \text{otherwise} \end{cases}
\end{aligned} \tag{32}$$

(here we have used that $\lim_{\eta\to 0}\frac{s_\eta^{[\widetilde{\jmath}:j]}(z)}{s_\eta^{[\widetilde{\jmath}:j]}(1)} = z$ by observation of the definition), which justifies our simplification in the main body of the paper.

## C.3 Proof of Corollaries 2 and 3 - Continuity Bounds

Our approach here mimics the previous two proofs. For the input node $j = 0$, for a given $\Theta \in \mathbb{W}$:

$$\sup_{\boldsymbol{x}\in\mathbb{X}} \frac{\left\|\left[\langle \mathbf{\Psi}^{[0]}(\Theta),\boldsymbol{\phi}^{[0]}(\boldsymbol{x})\rangle\right]_{\mathbf{g}^{[0]}}\right\|_2^2}{\left\|\boldsymbol{\phi}^{[0]}(\boldsymbol{x})\right\|_2^2} = \sup_{\boldsymbol{x}\in\mathbb{X}}\frac{\|\boldsymbol{x}\|_2^2}{\|\boldsymbol{x}\|_2^2} = C_{\Theta,\eta}^{[0]2} \triangleq 1 \quad \text{for given } \Theta \in \mathbb{W}$$

$$\sup_{\boldsymbol{x}\in\mathbb{X}} \frac{\left\|\left[\langle \mathbf{\Psi}^{[0]}(\Theta),\boldsymbol{\phi}^{[0]}(\boldsymbol{x})\rangle\right]_{\mathbf{g}^{[0]}}\right\|_2^2}{\left\|\boldsymbol{\phi}^{[0]}(\boldsymbol{x})\right\|_2^2} = \sup_{\boldsymbol{x}\in\mathbb{X}}\frac{\|\boldsymbol{x}\|_2^2}{\|\boldsymbol{x}\|_2^2} = C_{\mathbb{W},\eta}^{[0]2} \triangleq 1 \quad \forall\Theta \in \mathbb{W}$$

$$\sup_{\Theta\in\mathbb{W}} \frac{\left\|\left[\langle \mathbf{\Psi}^{[0]}(\Theta),\boldsymbol{\phi}^{[0]}(\boldsymbol{x})\rangle\right]_{\mathbf{g}^{[0]}}\right\|_2^2}{\left\|\mathbf{\Psi}^{[0]}(\Theta)\right\|_F^2} = \frac{\|\boldsymbol{x}\|_2^2}{r^2} \leq C_{\boldsymbol{x},\eta}^{[0]2} \triangleq 1 \quad \text{for given } \boldsymbol{x} \in \mathbb{X}$$

$$\sup_{\Theta\in\mathbb{W}} \frac{\left\|\left[\langle \mathbf{\Psi}^{[0]}(\Theta),\boldsymbol{\phi}^{[0]}(\boldsymbol{x})\rangle\right]_{\mathbf{g}^{[0]}}\right\|_2^2}{\left\|\mathbf{\Psi}^{[0]}(\Theta)\right\|_F^2} = \frac{\|\boldsymbol{x}\|_2^2}{r^2} \leq C_{\mathbb{X},\eta}^{[0]2} \triangleq 1 \quad \forall\boldsymbol{x} \in \mathbb{X}$$

As in the previous section consider a single node $j \in \mathbb{L}^{[\dot{j}]}$ in layer $\dot{j}$. Assume that, for all nodes in the previous layer $\widetilde{j} \in \mathbb{L}^{[\dot{j}-1]}$:

$$
\left.
\begin{array}{l}
\displaystyle \sup_{\boldsymbol{x}\in\mathbb{X}} \frac{\left\|\left\langle \boldsymbol{\Psi}^{[\bar{j}]}(\Theta),\boldsymbol{\phi}^{[\bar{j}]}(\boldsymbol{x})\right\rangle_{\mathbf{g}^{[\bar{j}]}}\right\|_2^2}{\left\|\boldsymbol{\phi}^{[\bar{j}]}(\boldsymbol{x})\right\|_2^2} \leq C_{\Theta,\eta}^{[\bar{j}]2} \quad \text{for given } \Theta \in \mathbb{W} \\[2em]
\displaystyle \sup_{\boldsymbol{x}\in\mathbb{X}} \frac{\left\|\left\langle \boldsymbol{\Psi}^{[\bar{j}]}(\Theta),\boldsymbol{\phi}^{[\bar{j}]}(\boldsymbol{x})\right\rangle_{\mathbf{g}^{[\bar{j}]}}\right\|_2^2}{\left\|\boldsymbol{\phi}^{[\bar{j}]}(\boldsymbol{x})\right\|_2^2} \leq C_{\mathbb{W},\eta}^{[\bar{j}]2} \quad \forall \Theta \in \mathbb{W}
\end{array}
\right\} \quad C_{\Theta,\eta}^{[\bar{j}]2} \leq C_{\mathbb{W},\eta}^{[\bar{j}]2}
$$

$$
\left.
\begin{array}{l}
\displaystyle \sup_{\Theta\in\mathbb{W}} \frac{\left\|\left\langle \boldsymbol{\Psi}^{[\bar{j}]}(\Theta),\boldsymbol{\phi}^{[\bar{j}]}(\boldsymbol{x})\right\rangle_{\mathbf{g}^{[\bar{j}]}}\right\|_2^2}{\left\|\boldsymbol{\Psi}^{[\bar{j}]}(\Theta)\right\|_F^2} \leq C_{\boldsymbol{x},\eta}^{[\bar{j}]2} \quad \text{for given } \boldsymbol{x} \in \mathbb{X} \\[2em]
\displaystyle \sup_{\Theta\in\mathbb{W}} \frac{\left\|\left\langle \boldsymbol{\Psi}^{[\bar{j}]}(\Theta),\boldsymbol{\phi}^{[\bar{j}]}(\boldsymbol{x})\right\rangle_{\mathbf{g}^{[\bar{j}]}}\right\|_2^2}{\left\|\boldsymbol{\Psi}^{[\bar{j}]}(\Theta)\right\|_F^2} \leq C_{\mathbb{X},\eta}^{[\bar{j}]2} \quad \forall \boldsymbol{x} \in \mathbb{X}
\end{array}
\right\} \quad C_{\boldsymbol{x},\eta}^{[\bar{j}]2} \leq C_{\mathbb{X},\eta}^{[\bar{j}]2}
$$

**Edge case:** for a Lipschitz neural activation $\boldsymbol{\tau}^{[\widetilde{j};j]}$, for incoming edges (21a), for fixed $\Theta \in \mathbb{W}$:

$$
\begin{aligned}
\sup_{\boldsymbol{x}\in\mathbb{X}} \frac{\left\|\left\langle \boldsymbol{\Psi}^{[\widetilde{j};j]}(\Theta),\boldsymbol{\phi}^{[\widetilde{j};j]}(\boldsymbol{x})\right\rangle_{\mathbf{g}^{[\widetilde{j};j]}}\right\|_2^2}{\left\|\boldsymbol{\phi}^{[\widetilde{j};j]}(\boldsymbol{x})\right\|_2^2}
&= \sup_{\boldsymbol{x}\in\mathbb{X}} \frac{\left\|\boldsymbol{\tau}^{[\widetilde{j};j]}\!\left(\boldsymbol{x}^{[\widetilde{j}]}\right)\right\|_2^2}{\left\|\boldsymbol{\phi}^{[\widetilde{j};j]}(\boldsymbol{x})\right\|_2^2} \\[1em]
&= \sup_{\boldsymbol{x}\in\mathbb{X}} \frac{\left\|\boldsymbol{\tau}^{[\widetilde{j};j]}\!\left(\boldsymbol{x}^{[\widetilde{j}]}\right)\right\|_2^2}{L_{\phi_\eta^{[\widetilde{j}]}}^{[\widetilde{j};j]2}\, \frac{s_\eta^{[\widetilde{j};j]}\left(\left\|\boldsymbol{\phi}^{[\widetilde{j}]}(\boldsymbol{x})\right\|_2^2\right)}{s_\eta^{[\widetilde{j};j]}(1)}} \\[1em]
&\leq \sup_{\boldsymbol{x}\in\mathbb{X}} \frac{L_{C_{\Theta,\eta}^{[\widetilde{j}]}\phi_\eta^{[\widetilde{j}]}}^{[\widetilde{j};j]2}\left\|\boldsymbol{x}^{[\widetilde{j}]}\right\|_2^2}{L_{\phi_\eta^{[\widetilde{j}]}}^{[\widetilde{j};j]2}\, \frac{s_\eta^{[\widetilde{j};j]}\left(\left\|\boldsymbol{\phi}^{[\widetilde{j}]}(\boldsymbol{x})\right\|_2^2\right)}{s_\eta^{[\widetilde{j};j]}(1)}} \\[1em]
&\leq \frac{L_{C_{\Theta,\eta}^{[\widetilde{j}]}\phi_\eta^{[\widetilde{j}]}}^{[\widetilde{j};j]2}}{L_{\phi_\eta^{[\widetilde{j}]}}^{[\widetilde{j};j]2}} \left(\sup_{\boldsymbol{x}\in\mathbb{X}} \frac{\left\|\boldsymbol{\phi}^{[\widetilde{j}]}(\boldsymbol{x})\right\|_2^2}{\frac{s_\eta^{[\widetilde{j};j]}\left(\left\|\boldsymbol{\phi}^{[\widetilde{j}]}(\boldsymbol{x})\right\|_2^2\right)}{s_\eta^{[\widetilde{j};j]}(1)}}\right) \left(\sup_{\boldsymbol{x}\in\mathbb{X}} \frac{\left\|\left\langle \boldsymbol{\Psi}^{[\widetilde{j}]}(\Theta),\boldsymbol{\phi}^{[\widetilde{j}]}(\boldsymbol{x})\right\rangle_{\mathbf{g}^{[\widetilde{j}]}}\right\|_2^2}{\left\|\boldsymbol{\phi}^{[\widetilde{j}]}(\boldsymbol{x})\right\|_2^2}\right) \\[1em]
&\leq \frac{L_{C_{\Theta,\eta}^{[\widetilde{j}]}\phi_\eta^{[\widetilde{j}]}}^{[\widetilde{j};j]2}}{L_{\phi_\eta^{[\widetilde{j}]}}^{[\widetilde{j};j]2}} \sup_{\boldsymbol{x}\in\mathbb{X}} \frac{\left\|\left\langle \boldsymbol{\Psi}^{[\widetilde{j}]}(\Theta),\boldsymbol{\phi}^{[\widetilde{j}]}(\boldsymbol{x})\right\rangle_{\mathbf{g}^{[\widetilde{j}]}}\right\|_2^2}{\left\|\boldsymbol{\phi}^{[\widetilde{j}]}(\boldsymbol{x})\right\|_2^2}
\end{aligned}
$$

and similarly for fixed $\boldsymbol{x} \in \mathbb{X}$:

$$
\begin{aligned}
\sup_{\Theta\in\mathbb{W}} \frac{\left\|\left\langle \boldsymbol{\Psi}^{[\widetilde{j};j]}(\Theta),\boldsymbol{\phi}^{[\widetilde{j};j]}(\boldsymbol{x})\right\rangle_{\mathbf{g}^{[\widetilde{j};j]}}\right\|_2^2}{\left\|\boldsymbol{\Psi}^{[\widetilde{j};j]}(\Theta)\right\|_F^2}
&= \sup_{\Theta\in\mathbb{W}} \frac{\left\|\boldsymbol{\tau}^{[\widetilde{j};j]}\!\left(\boldsymbol{x}^{[\widetilde{j}]}\right)\right\|_2^2}{\left\|\boldsymbol{\Psi}^{[\widetilde{j};j]}(\Theta)\right\|_F^2} \\[1em]
&= \sup_{\Theta\in\mathbb{W}} \frac{\left\|\boldsymbol{\tau}^{[\widetilde{j};j]}\!\left(\boldsymbol{x}^{[\widetilde{j}]}\right)\right\|_2^2}{L_{\psi_\eta^{[\widetilde{j}]}}^{[\widetilde{j};j]2}\, \frac{s_\eta^{[\widetilde{j};j]}\left(\left\|\boldsymbol{\Psi}^{[\widetilde{j}]}(\Theta)\right\|_F^2\right)}{s_\eta^{[\widetilde{j};j]}(1)}} \\[1em]
&\leq \sup_{\Theta\in\mathbb{W}} \frac{L_{C_{\boldsymbol{x},\eta}^{[\widetilde{j}]}\psi_\eta^{[\widetilde{j}]}}^{[\widetilde{j};j]2}}{L_{\psi_\eta^{[\widetilde{j}]}}^{[\widetilde{j};j]2}}\, \frac{\left\|\boldsymbol{x}^{[\widetilde{j}]}\right\|_2^2}{\frac{s_\eta^{[\widetilde{j};j]}\left(\left\|\boldsymbol{\Psi}^{[\widetilde{j}]}(\Theta)\right\|_F^2\right)}{s_\eta^{[\widetilde{j};j]}(1)}} \\[1em]
&\leq \frac{L_{C_{\boldsymbol{x},\eta}^{[\widetilde{j}]}\psi_\eta^{[\widetilde{j}]}}^{[\widetilde{j};j]2}}{L_{\psi_\eta^{[\widetilde{j}]}}^{[\widetilde{j};j]2}} \left(\sup_{\Theta\in\mathbb{W}} \frac{\left\|\boldsymbol{\Psi}^{[\widetilde{j}]}(\Theta)\right\|_F^2}{\frac{s_\eta^{[\widetilde{j};j]}\left(\left\|\boldsymbol{\Psi}^{[\widetilde{j}]}(\Theta)\right\|_F^2\right)}{s_\eta^{[\widetilde{j};j]}(1)}}\right) \left(\sup_{\Theta\in\mathbb{W}} \frac{\left\|\left\langle \boldsymbol{\Psi}^{[\widetilde{j}]}(\Theta),\boldsymbol{\phi}^{[\widetilde{j}]}(\boldsymbol{x})\right\rangle_{\mathbf{g}^{[\widetilde{j}]}}\right\|_2^2}{\left\|\boldsymbol{\Psi}^{[\widetilde{j}]}(\Theta)\right\|_F^2}\right) \\[1em]
&\leq \frac{L_{C_{\boldsymbol{x},\eta}^{[\widetilde{j}]}\psi_\eta^{[\widetilde{j}]}}^{[\widetilde{j};j]2}}{L_{\psi_\eta^{[\widetilde{j}]}}^{[\widetilde{j};j]2}} \sup_{\Theta\in\mathbb{W}} \frac{\left\|\left\langle \boldsymbol{\Psi}^{[\widetilde{j}]}(\Theta),\boldsymbol{\phi}^{[\widetilde{j}]}(\boldsymbol{x})\right\rangle_{\mathbf{g}^{[\widetilde{j}]}}\right\|_2^2}{\left\|\boldsymbol{\Psi}^{[\widetilde{j}]}(\Theta)\right\|_F^2}
\end{aligned}
$$

Alternatively, for bounded (non-Lipschitz) neural activations:

$$\sup_{\boldsymbol{x}\in\mathbb{X}}\frac{\left\|\left\langle\boldsymbol{\Psi}^{[\bar{\jmath}:j]}(\Theta),\boldsymbol{\phi}^{[\bar{\jmath}:j]}(\boldsymbol{x})\right]_{\mathbf{g}^{[\bar{\jmath}:j]}}\right\|_2^2}{\left\|\boldsymbol{\phi}^{[\bar{\jmath}:j]}(\boldsymbol{x})\right\|_2^2}=\sup_{\boldsymbol{x}\in\mathbb{X}}\frac{\left\|\boldsymbol{\tau}^{[\bar{\jmath}:j]}\left(\boldsymbol{x}^{[\bar{\jmath}]}\right)\right\|_2^2}{\frac{B^{[\bar{\jmath}:j]2}}{\phi_{\downarrow\eta}^{[\bar{\jmath}]2}}\frac{s_\eta^{[\bar{\jmath}:j]}\left(\left\|\boldsymbol{\phi}^{[\bar{\jmath}]}(\boldsymbol{x})\right\|_2^2\right)}{s_\eta^{[\bar{\jmath}:j]}(1)}}$$

$$=\sup_{\boldsymbol{x}\in\mathbb{X}}\frac{\frac{1}{B^{[\bar{\jmath}:j]2}}\left\|\boldsymbol{\tau}^{[\bar{\jmath}:j]}\left(\boldsymbol{x}^{[\bar{\jmath}]}\right)\right\|_2^2}{\frac{1}{\phi_{\downarrow\eta}^{[\bar{\jmath}]2}}\frac{s_\eta^{[\bar{\jmath}:j]}\left(\left\|\boldsymbol{\phi}^{[\bar{\jmath}]}(\boldsymbol{x})\right\|_2^2\right)}{s_\eta^{[\bar{\jmath}:j]}(1)}}$$

$$\leq\sup_{\boldsymbol{x}\in\mathbb{X}}\frac{s_\eta^{[\bar{\jmath}:j]}(1)}{\frac{1}{\phi_{\downarrow\eta}^{[\bar{\jmath}]2}}s_\eta^{[\bar{\jmath}:j]}\left(\left\|\boldsymbol{\phi}^{[\bar{\jmath}]}(\boldsymbol{x})\right\|_2^2\right)}$$

$$=^{(*)}\frac{s_\eta^{[\bar{\jmath}:j]}(1)}{\frac{1}{\phi_{\downarrow\eta}^{[\bar{\jmath}]2}}s_\eta^{[\bar{\jmath}:j]}\left(\phi_{\downarrow\eta}^{[\bar{\jmath}]2}\right)}$$

which is finite; and:

$$\sup_{\Theta\in\mathbb{W}}\frac{\left\|\left\langle\boldsymbol{\Psi}^{[\bar{\jmath}:j]}(\Theta),\boldsymbol{\phi}^{[\bar{\jmath}:j]}(\boldsymbol{x})\right]_{\mathbf{g}^{[\bar{\jmath}:j]}}\right\|_2^2}{\left\|\boldsymbol{\Psi}^{[\bar{\jmath}:j]}(\Theta)\right\|_F^2}=\sup_{\Theta\in\mathbb{W}}\frac{\left\|\boldsymbol{\tau}^{[\bar{\jmath}:j]}\left(\boldsymbol{x}^{[\bar{\jmath}]}\right)\right\|_2^2}{\frac{B^{[\bar{\jmath}:j]2}}{\phi_{\downarrow\eta}^{[\bar{\jmath}]2}}\frac{s_\eta^{[\bar{\jmath}:j]}\left(\left\|\boldsymbol{\Psi}^{[\bar{\jmath}]}(\Theta)\right\|_F^2\right)}{s_\eta^{[\bar{\jmath}:j]}(1)}}$$

$$=\sup_{\Theta\in\mathbb{W}}\frac{\frac{1}{B^{[\bar{\jmath}:j]2}}\left\|\boldsymbol{\tau}^{[\bar{\jmath}:j]}\left(\boldsymbol{x}^{[\bar{\jmath}]}\right)\right\|_2^2}{\frac{1}{\phi_{\downarrow\eta}^{[\bar{\jmath}]2}}\frac{s_\eta^{[\bar{\jmath}:j]}\left(\left\|\boldsymbol{\Psi}^{[\bar{\jmath}]}(\Theta)\right\|_F^2\right)}{s_\eta^{[\bar{\jmath}:j]}(1)}}$$

$$\leq\sup_{\Theta\in\mathbb{W}}\frac{s_\eta^{[\bar{\jmath}:j]}(1)}{\frac{1}{\phi_{\downarrow\eta}^{[\bar{\jmath}]2}}s_\eta^{[\bar{\jmath}:j]}\left(\left\|\boldsymbol{\Psi}^{[\bar{\jmath}]}(\Theta)\right\|_F^2\right)}$$

which is unbounded in general. It follows that:

$$\sup_{\boldsymbol{x}\in\mathbb{X}}\frac{\left\|\left\langle\boldsymbol{\Psi}^{[\bar{\jmath}:j]}(\Theta),\boldsymbol{\phi}^{[\bar{\jmath}:j]}(\boldsymbol{x})\right]_{\mathbf{g}^{[\bar{\jmath}:j]}}\right\|_2^2}{\left\|\boldsymbol{\phi}^{[\bar{\jmath}:j]}(\boldsymbol{x})\right\|_2^2}\leq C_{\Theta,\eta}^{[\bar{\jmath}:j]2}\triangleq\begin{cases}\frac{L^{[\bar{\jmath}:j]2}}{C_{\Theta,\eta}^{[\bar{\jmath}]}\phi_\eta^{[\bar{\jmath}]}}\frac{L^{[\bar{\jmath}:j]2}}{L_{\phi_\eta^{[\bar{\jmath}]}}^{[\bar{\jmath}:j]2}}C_{\Theta,\eta}^{[\bar{\jmath}]2} & \text{if }\boldsymbol{\tau}^{[\bar{\jmath}:j]}\text{ is Lipschitz}\\[2ex]\frac{s_\eta^{[\bar{\jmath}:j]}(1)}{\frac{1}{\phi_{\downarrow\eta}^{[\bar{\jmath}]2}}s_\eta^{[\bar{\jmath}:j]}\left(\phi_{\downarrow\eta}^{[\bar{\jmath}]2}\right)} & \text{otherwise}\end{cases}\quad\text{for given }\Theta\in\mathbb{W}$$

$$\sup_{\boldsymbol{x}\in\mathbb{X}}\frac{\left\|\left\langle\boldsymbol{\Psi}^{[\bar{\jmath}:j]}(\Theta),\boldsymbol{\phi}^{[\bar{\jmath}:j]}(\boldsymbol{x})\right]_{\mathbf{g}^{[\bar{\jmath}:j]}}\right\|_2^2}{\left\|\boldsymbol{\phi}^{[\bar{\jmath}:j]}(\boldsymbol{x})\right\|_2^2}\leq C_{\mathbb{W},\eta}^{[\bar{\jmath}:j]2}\triangleq\begin{cases}\frac{L^{[\bar{\jmath}:j]2}}{C_{\mathbb{W},\eta}^{[\bar{\jmath}]}\phi_\eta^{[\bar{\jmath}]}}\frac{L^{[\bar{\jmath}:j]2}}{L_{\phi_\eta^{[\bar{\jmath}]}}^{[\bar{\jmath}:j]2}}C_{\mathbb{W},\eta}^{[\bar{\jmath}]2} & \text{if }\boldsymbol{\tau}^{[\bar{\jmath}:j]}\text{ is Lipschitz}\\[2ex]\frac{s_\eta^{[\bar{\jmath}:j]}(1)}{\frac{1}{\phi_{\downarrow\eta}^{[\bar{\jmath}]2}}s_\eta^{[\bar{\jmath}:j]}\left(\phi_{\downarrow\eta}^{[\bar{\jmath}]2}\right)} & \text{otherwise}\end{cases}\quad\forall\Theta\in\mathbb{W}$$

$$\left.\begin{array}{l}\end{array}\right\}C_{\Theta,\eta}^{[\bar{\jmath}:j]}\leq C_{\mathbb{W},\eta}^{[\bar{\jmath}]}$$

$$\sup_{\Theta\in\mathbb{W}}\frac{\left\|\left\langle\boldsymbol{\Psi}^{[\bar{\jmath}:j]}(\Theta),\boldsymbol{\phi}^{[\bar{\jmath}:j]}(\boldsymbol{x})\right]_{\mathbf{g}^{[\bar{\jmath}:j]}}\right\|_2^2}{\left\|\boldsymbol{\Psi}^{[\bar{\jmath}:j]}(\Theta)\right\|_F^2}\leq C_{\boldsymbol{x},\eta}^{[\bar{\jmath}:j]2}\triangleq\begin{cases}\frac{L^{[\bar{\jmath}:j]2}}{C_{\boldsymbol{x},\eta}^{[\bar{\jmath}]}\psi_\eta^{[\bar{\jmath}]}}\frac{L^{[\bar{\jmath}:j]2}}{L_{\psi_\eta^{[\bar{\jmath}]}}^{[\bar{\jmath}:j]2}}C_{\boldsymbol{x},\eta}^{[\bar{\jmath}]2} & \text{if }\boldsymbol{\tau}^{[\bar{\jmath}:j]}\text{ is Lipschitz}\\[2ex]\infty & \text{otherwise}\end{cases}\quad\text{for given }\boldsymbol{x}\in\mathbb{X}$$

$$\sup_{\Theta\in\mathbb{W}}\frac{\left\|\left\langle\boldsymbol{\Psi}^{[\bar{\jmath}:j]}(\Theta),\boldsymbol{\phi}^{[\bar{\jmath}:j]}(\boldsymbol{x})\right]_{\mathbf{g}^{[\bar{\jmath}:j]}}\right\|_2^2}{\left\|\boldsymbol{\Psi}^{[\bar{\jmath}:j]}(\Theta)\right\|_F^2}\leq C_{\mathbb{X},\eta}^{[\bar{\jmath}:j]2}\triangleq\begin{cases}\frac{L^{[\bar{\jmath}:j]2}}{C_{\mathbb{X},\eta}^{[\bar{\jmath}]}\psi_\eta^{[\bar{\jmath}]}}\frac{L^{[\bar{\jmath}:j]2}}{L_{\psi_\eta^{[\bar{\jmath}]}}^{[\bar{\jmath}:j]2}}C_{\mathbb{X},\eta}^{[\bar{\jmath}]2} & \text{if }\boldsymbol{\tau}^{[\bar{\jmath}:j]}\text{ is Lipschitz}\\[2ex]\infty & \text{otherwise}\end{cases}\quad\forall\boldsymbol{x}\in\mathbb{X}$$

$$\left.\begin{array}{l}\end{array}\right\}C_{\boldsymbol{x},\eta}^{[\bar{\jmath}:j]}\leq C_{\mathbb{X},\eta}^{[\bar{\jmath}:j]}$$

**Edge case:** using (21b), for columnar concatenation nodes $\bigcirc^{[j]} = \bigoplus$ (so $\square^{[j]} = \mathrm{diag}$, $\boxdot^{[j]} = \bigoplus$):

$$
\sup_{\boldsymbol{x}\in\mathbb{X}} \frac{\left\|\left\langle\boldsymbol{\Psi}^{[j]}(\Theta),\boldsymbol{\phi}^{[j]}(\boldsymbol{x})\right\rangle_{\mathbf{g}^{[j]}}\right\|_2^2}{\left\|\boldsymbol{\phi}^{[j]}(\boldsymbol{x})\right\|_2^2} = \sup_{\boldsymbol{x}\in\mathbb{X}} \frac{\left\|\boldsymbol{x}^{[j]}\right\|_2^2}{\left\|\boldsymbol{\phi}^{[j]}(\boldsymbol{x})\right\|_2^2} = \sup_{\boldsymbol{x}\in\mathbb{X}} \frac{\left\|\mathbf{b}^{[j]}+\bigoplus_{\widetilde{\jmath}\in\mathbb{A}^{[j]}}\mathbf{W}^{[\widetilde{\jmath};j]\mathrm{T}}\boldsymbol{x}^{[\widetilde{\jmath};j]}\right\|_2^2}{\left\|\boldsymbol{\phi}^{[j]}(\boldsymbol{x})\right\|_2^2}
$$

$$
= \sup_{\boldsymbol{x}\in\mathbb{X}} \frac{\left\|\mathbf{b}^{[j]}+\bigoplus_{\widetilde{\jmath}\in\mathbb{A}^{[j]}}\mathbf{W}^{[\widetilde{\jmath};j]\mathrm{T}}\boldsymbol{x}^{[\widetilde{\jmath};j]}\right\|_2^2}{\beta^{[j]2}+\sum_{\widetilde{\jmath}\in\mathbb{A}^{[j]}}\omega^{[\widetilde{\jmath};j]2}\left\|\boldsymbol{\phi}^{[\widetilde{\jmath};j]}(\boldsymbol{x})\right\|_2^2}
$$

$$
\leq \sup_{\boldsymbol{x}\in\mathbb{X}} \frac{\left\|\mathbf{b}^{[j]}\right\|_2^2+\sum_{\widetilde{\jmath}\in\mathbb{A}^{[j]}}\left\|\mathbf{W}^{[\widetilde{\jmath};j]\mathrm{T}}\boldsymbol{x}^{[\widetilde{\jmath};j]}\right\|_2^2}{\beta^{[j]2}+\sum_{\widetilde{\jmath}\in\mathbb{A}^{[j]}}\omega^{[\widetilde{\jmath};j]2}\left\|\boldsymbol{\phi}^{[\widetilde{\jmath};j]}(\boldsymbol{x})\right\|_2^2}
$$

$$
= \sup_{\boldsymbol{x}\in\mathbb{X}} \frac{\left\|\mathbf{b}^{[j]}\right\|_2^2+\sum_{\widetilde{\jmath}\in\mathbb{A}^{[j]}}\left\|\mathbf{W}^{[\widetilde{\jmath};j]\mathrm{T}}\left\langle\boldsymbol{\Psi}^{[\widetilde{\jmath};j]}(\Theta),\boldsymbol{\phi}^{[\widetilde{\jmath};j]}(\boldsymbol{x})\right\rangle_{\mathbf{g}^{[\widetilde{\jmath};j]}}\right\|_2^2}{\beta^{[j]2}+\sum_{\widetilde{\jmath}\in\mathbb{A}^{[j]}}\omega^{[\widetilde{\jmath};j]2}\left\|\boldsymbol{\phi}^{[\widetilde{\jmath};j]}(\boldsymbol{x})\right\|_2^2}
$$

$$
\leq \sup_{\boldsymbol{x}\in\mathbb{X}} \frac{\left\|\mathbf{b}^{[j]}\right\|_2^2+\sum_{\widetilde{\jmath}\in\mathbb{A}^{[j]}}\left\|\mathbf{W}^{[\widetilde{\jmath};j]}\right\|_2^2\left\|\boldsymbol{\phi}^{[\widetilde{\jmath};j]}(\boldsymbol{x})\right\|_2^2 C_{\Theta,\eta}^{[\widetilde{\jmath};j]2}}{\beta^{[j]2}+\sum_{\widetilde{\jmath}\in\mathbb{A}^{[j]}}\omega^{[\widetilde{\jmath};j]2}\left\|\boldsymbol{\phi}^{[\widetilde{\jmath};j]}(\boldsymbol{x})\right\|_2^2}
$$

$$
\leq \sup_{\boldsymbol{x}\in\mathbb{X}} \frac{\beta^{[j]2}+\sum_{\widetilde{\jmath}\in\mathbb{A}^{[j]}}\omega^{[\widetilde{\jmath};j]2}\left\|\boldsymbol{\phi}^{[\widetilde{\jmath};j]}(\boldsymbol{x})\right\|_2^2}{\beta^{[j]2}+\sum_{\widetilde{\jmath}\in\mathbb{A}^{[j]}}\omega^{[\widetilde{\jmath};j]2}\left\|\boldsymbol{\phi}^{[\widetilde{\jmath};j]}(\boldsymbol{x})\right\|_2^2}\max\left\{\gamma^{[j]},\max_{\widetilde{\jmath}\in\mathbb{A}^{[j]}}\left\{C_{\Theta,\eta}^{[\widetilde{\jmath};j]2}\right\}\right\}
$$

$$
= \max\left\{\gamma^{[j]},\max_{\widetilde{\jmath}\in\mathbb{A}^{[j]}}\left\{C_{\Theta,\eta}^{[\widetilde{\jmath};j]2}\right\}\right\}
$$

and:

$$
\sup_{\Theta\in\mathbb{W}} \frac{\left\|\left\langle\boldsymbol{\Psi}^{[j]}(\Theta),\boldsymbol{\phi}^{[j]}(\boldsymbol{x})\right\rangle_{\mathbf{g}^{[j]}}\right\|_2^2}{\left\|\boldsymbol{\Psi}^{[j]}(\Theta)\right\|_F^2} \leq \sup_{\Theta\in\mathbb{W}} \frac{\left\|\mathbf{b}^{[j]}\right\|_2^2+\sum_{\widetilde{\jmath}\in\mathbb{A}^{[j]}}\left\|\mathbf{W}^{[\widetilde{\jmath};j]}\right\|_2^2\left\|\boldsymbol{\Psi}^{[\widetilde{\jmath};j]}(\Theta)\right\|_F^2 C_{\boldsymbol{x},\eta}^{[\widetilde{\jmath};j]2}}{\left\|\mathbf{b}^{[j]}\right\|_2^2+\sum_{\widetilde{\jmath}\in\mathbb{A}^{[j]}}\left\|\mathbf{W}^{[\widetilde{\jmath};j]}\right\|_2^2\left\|\boldsymbol{\phi}^{[\widetilde{\jmath};j]}(\boldsymbol{x})\right\|_2^2}
$$

$$
\leq \frac{\left\|\mathbf{b}^{[j]}\right\|_2^2+\sum_{\widetilde{\jmath}\in\mathbb{A}^{[j]}}\left\|\mathbf{W}^{[\widetilde{\jmath};j]}\right\|_2^2\left\|\boldsymbol{\phi}^{[\widetilde{\jmath};j]}(\boldsymbol{x})\right\|_2^2}{\left\|\mathbf{b}^{[j]}\right\|_2^2+\sum_{\widetilde{\jmath}\in\mathbb{A}^{[j]}}\left\|\mathbf{W}^{[\widetilde{\jmath};j]}\right\|_2^2\left\|\boldsymbol{\phi}^{[\widetilde{\jmath};j]}(\boldsymbol{x})\right\|_2^2}\max\left\{\gamma^{[j]},\max_{\widetilde{\jmath}\in\mathbb{A}^{[j]}}\left\{C_{\boldsymbol{x},\eta}^{[\widetilde{\jmath};j]2}\right\}\right\}
$$

$$
\leq \max\left\{\gamma^{[j]},\max_{\widetilde{\jmath}\in\mathbb{A}^{[j]}}\left\{C_{\boldsymbol{x},\eta}^{[\widetilde{\jmath};j]2}\right\}\right\}
$$

For additive nodes $\bigcirc^{[j]} = \sum$ (so $\square^{[j]} = \bigoplus$, $\boxdot^{[j]} = \bigoplus$):

$$
\sup_{\boldsymbol{x}\in\mathbb{X}} \frac{\left\|\left\langle\boldsymbol{\Psi}^{[j]}(\Theta),\boldsymbol{\phi}^{[j]}(\boldsymbol{x})\right\rangle_{\mathbf{g}^{[j]}}\right\|_2^2}{\left\|\boldsymbol{\phi}^{[j]}(\boldsymbol{x})\right\|_2^2} = \sup_{\boldsymbol{x}\in\mathbb{X}} \frac{\left\|\boldsymbol{x}^{[j]}\right\|_2^2}{\left\|\boldsymbol{\phi}^{[j]}(\boldsymbol{x})\right\|_2^2} = \sup_{\boldsymbol{x}\in\mathbb{X}} \frac{\left\|\mathbf{b}^{[j]}+\sum_{\widetilde{\jmath}\in\mathbb{A}^{[j]}}\mathbf{W}^{[\widetilde{\jmath};j]\mathrm{T}}\boldsymbol{x}^{[\widetilde{\jmath};j]}\right\|_2^2}{\left\|\boldsymbol{\phi}^{[j]}(\boldsymbol{x})\right\|_2^2}
$$

$$
= \sup_{\boldsymbol{x}\in\mathbb{X}} \frac{\left\|\mathbf{b}^{[j]}+\sum_{\widetilde{\jmath}\in\mathbb{A}^{[j]}}\mathbf{W}^{[\widetilde{\jmath};j]\mathrm{T}}\boldsymbol{x}^{[\widetilde{\jmath};j]}\right\|_2^2}{\beta^{[j]2}+\sum_{\widetilde{\jmath}\in\mathbb{A}^{[j]}}\omega^{[\widetilde{\jmath};j]2}\left\|\boldsymbol{\phi}^{[\widetilde{\jmath};j]}(\boldsymbol{x})\right\|_2^2}
$$

$$
\leq \sup_{\boldsymbol{x}\in\mathbb{X}} \frac{\left\|\mathbf{b}^{[j]}\right\|_2^2+\sum_{\widetilde{\jmath}\in\mathbb{A}^{[j]}}\left\|\mathbf{W}^{[\widetilde{\jmath};j]\mathrm{T}}\boldsymbol{x}^{[\widetilde{\jmath};j]}\right\|_2^2}{\beta^{[j]2}+\sum_{\widetilde{\jmath}\in\mathbb{A}^{[j]}}\omega^{[\widetilde{\jmath};j]2}\left\|\boldsymbol{\phi}^{[\widetilde{\jmath};j]}(\boldsymbol{x})\right\|_2^2}
$$

$$
= \sup_{\boldsymbol{x}\in\mathbb{X}} \frac{\left\|\mathbf{b}^{[j]}\right\|_2^2+\sum_{\widetilde{\jmath}\in\mathbb{A}^{[j]}}\left\|\mathbf{W}^{[\widetilde{\jmath};j]\mathrm{T}}\left\langle\boldsymbol{\Psi}^{[\widetilde{\jmath};j]}(\Theta),\boldsymbol{\phi}^{[\widetilde{\jmath};j]}(\boldsymbol{x})\right\rangle_{\mathbf{g}^{[\widetilde{\jmath};j]}}\right\|_2^2}{\beta^{[j]2}+\sum_{\widetilde{\jmath}\in\mathbb{A}^{[j]}}\omega^{[\widetilde{\jmath};j]2}\left\|\boldsymbol{\phi}^{[\widetilde{\jmath};j]}(\boldsymbol{x})\right\|_2^2}
$$

$$
\leq \sup_{\boldsymbol{x}\in\mathbb{X}} \frac{\left\|\mathbf{b}^{[j]}\right\|_2^2+\sum_{\widetilde{\jmath}\in\mathbb{A}^{[j]}}\left\|\mathbf{W}^{[\widetilde{\jmath};j]}\right\|_2^2\left\|\boldsymbol{\phi}^{[\widetilde{\jmath};j]}(\boldsymbol{x})\right\|_2^2 C_{\Theta,\eta}^{[\widetilde{\jmath};j]2}}{\beta^{[j]2}+\sum_{\widetilde{\jmath}\in\mathbb{A}^{[j]}}\omega^{[\widetilde{\jmath};j]2}\left\|\boldsymbol{\phi}^{[\widetilde{\jmath};j]}(\boldsymbol{x})\right\|_2^2}
$$

$$
\leq \sup_{\boldsymbol{x}\in\mathbb{X}} \frac{\beta^{[j]2}+\sum_{\widetilde{\jmath}\in\mathbb{A}^{[j]}}\omega^{[\widetilde{\jmath};j]2}\left\|\boldsymbol{\phi}^{[\widetilde{\jmath};j]}(\boldsymbol{x})\right\|_2^2}{\beta^{[j]2}+\sum_{\widetilde{\jmath}\in\mathbb{A}^{[j]}}\omega^{[\widetilde{\jmath};j]2}\left\|\boldsymbol{\phi}^{[\widetilde{\jmath};j]}(\boldsymbol{x})\right\|_2^2}\max\left\{\gamma^{[j]},\max_{\widetilde{\jmath}\in\mathbb{A}^{[j]}}\left\{C_{\Theta,\eta}^{[\widetilde{\jmath};j]2}\right\}\right\}
$$

$$
= \max\left\{\gamma^{[j]},\max_{\widetilde{\jmath}\in\mathbb{A}^{[j]}}\left\{C_{\Theta,\eta}^{[\widetilde{\jmath};j]2}\right\}\right\}
$$

and:

$$\sup_{\Theta \in \mathbb{W}} \frac{\left\|\left\langle \boldsymbol{\Psi}^{[j]}(\Theta), \boldsymbol{\phi}^{[j]}(\boldsymbol{x})\right\rangle_{\mathbf{g}^{[j]}}\right\|_2^2}{\left\|\boldsymbol{\Psi}^{[j]}(\Theta)\right\|_F^2} \leq \sup_{\Theta \in \mathbb{W}} \frac{\left\|\mathbf{b}^{[j]}\right\|_2^2 + \sum_{\widetilde{j} \in \mathbb{A}^{[j]}} \left\|\mathbf{W}^{[\widetilde{j}:j]}\right\|_2^2 \left\|\boldsymbol{\Psi}^{[\widetilde{j}:j]}(\Theta)\right\|_F^2 C_{\boldsymbol{x},\eta}^{[\widetilde{j}:j]2}}{\left\|\mathbf{b}^{[j]}\right\|_2^2 + \sum_{\widetilde{j} \in \mathbb{A}^{[j]}} \left\|\mathbf{W}^{[\widetilde{j}:j]}\right\|_2^2 \left\|\boldsymbol{\phi}^{[\widetilde{j}:j]}(\boldsymbol{x})\right\|_2^2}$$

$$\leq \frac{\left\|\mathbf{b}^{[j]}\right\|_2^2 + \sum_{\widetilde{j} \in \mathbb{A}^{[j]}} \left\|\mathbf{W}^{[\widetilde{j}:j]}\right\|_2^2 \left\|\boldsymbol{\phi}^{[\widetilde{j}:j]}(\boldsymbol{x})\right\|_2^2}{\left\|\mathbf{b}^{[j]}\right\|_2^2 + \sum_{\widetilde{j} \in \mathbb{A}^{[j]}} \left\|\mathbf{W}^{[\widetilde{j}:j]}\right\|_2^2 \left\|\boldsymbol{\phi}^{[\widetilde{j}:j]}(\boldsymbol{x})\right\|_2^2} \max\left\{\gamma^{[j]}, \max_{\widetilde{j} \in \mathbb{A}^{[j]}}\left\{C_{\boldsymbol{x},\eta}^{[\widetilde{j}:j]2}\right\}\right\}$$

$$\leq \max\left\{\gamma^{[j]}, \max_{\widetilde{j} \in \mathbb{A}^{[j]}}\left\{C_{\boldsymbol{x},\eta}^{[\widetilde{j}:j]2}\right\}\right\}$$

For Kronecker-product nodes $\bigcirc^{[j]} = \bigotimes$ (so $\square^{[j]} = \bigotimes$, $\boxdot^{[j]} = \bigotimes$):

$$\sup_{\boldsymbol{x} \in \mathbb{X}} \frac{\left\|\left\langle \boldsymbol{\Psi}^{[j]}(\Theta), \boldsymbol{\phi}^{[j]}(\boldsymbol{x})\right\rangle_{\mathbf{g}^{[j]}}\right\|_2^2}{\left\|\boldsymbol{\phi}^{[j]}(\boldsymbol{x})\right\|_2^2} = \sup_{\boldsymbol{x} \in \mathbb{X}} \frac{\left\|\boldsymbol{x}^{[j]}\right\|_2^2}{\left\|\boldsymbol{\phi}^{[j]}(\boldsymbol{x})\right\|_2^2} = \sup_{\boldsymbol{x} \in \mathbb{X}} \frac{\left\|\mathbf{b}^{[j]} + \bigotimes_{\widetilde{j} \in \mathbb{A}^{[j]}} \mathbf{W}^{[\widetilde{j}:j]\mathbf{T}} \boldsymbol{x}^{[\widetilde{j}:j]}\right\|_2^2}{\left\|\boldsymbol{\phi}^{[j]}(\boldsymbol{x})\right\|_2^2}$$

$$= \sup_{\boldsymbol{x} \in \mathbb{X}} \frac{\left\|\mathbf{b}^{[j]} + \bigotimes_{\widetilde{j} \in \mathbb{A}^{[j]}} \mathbf{W}^{[\widetilde{j}:j]\mathbf{T}} \boldsymbol{x}^{[\widetilde{j}:j]}\right\|_2^2}{\beta^{[j]2} + \prod_{\widetilde{j} \in \mathbb{A}^{[j]}} \omega^{[\widetilde{j}:j]2} \left\|\boldsymbol{\phi}^{[\widetilde{j}:j]}(\boldsymbol{x})\right\|_2^2}$$

$$\leq \sup_{\boldsymbol{x} \in \mathbb{X}} \frac{\left\|\mathbf{b}^{[j]}\right\|_2^2 + \prod_{\widetilde{j} \in \mathbb{A}^{[j]}} \left\|\mathbf{W}^{[\widetilde{j}:j]\mathbf{T}} \boldsymbol{x}^{[\widetilde{j}:j]}\right\|_2^2}{\beta^{[j]2} + \prod_{\widetilde{j} \in \mathbb{A}^{[j]}} \omega^{[\widetilde{j}:j]2} \left\|\boldsymbol{\phi}^{[\widetilde{j}:j]}(\boldsymbol{x})\right\|_2^2}$$

$$= \sup_{\boldsymbol{x} \in \mathbb{X}} \frac{\left\|\mathbf{b}^{[j]}\right\|_2^2 + \prod_{\widetilde{j} \in \mathbb{A}^{[j]}} \left\|\mathbf{W}^{[\widetilde{j}:j]\mathbf{T}} \left\langle \boldsymbol{\Psi}^{[\widetilde{j}:j]}(\Theta), \boldsymbol{\phi}^{[\widetilde{j}:j]}(\boldsymbol{x})\right\rangle_{\mathbf{g}^{[\widetilde{j}:j]}}\right\|_2^2}{\beta^{[j]2} + \prod_{\widetilde{j} \in \mathbb{A}^{[j]}} \omega^{[\widetilde{j}:j]2} \left\|\boldsymbol{\phi}^{[\widetilde{j}:j]}(\boldsymbol{x})\right\|_2^2}$$

$$\leq \sup_{\boldsymbol{x} \in \mathbb{X}} \frac{\left\|\mathbf{b}^{[j]}\right\|_2^2 + \prod_{\widetilde{j} \in \mathbb{A}^{[j]}} \left\|\mathbf{W}^{[\widetilde{j}:j]}\right\|_2^2 \left\|\boldsymbol{\phi}^{[\widetilde{j}:j]}(\boldsymbol{x})\right\|_2^2 C_{\Theta,\eta}^{[\widetilde{j}:j]2}}{\beta^{[j]2} + \prod_{\widetilde{j} \in \mathbb{A}^{[j]}} \omega^{[\widetilde{j}:j]2} \left\|\boldsymbol{\phi}^{[\widetilde{j}:j]}(\boldsymbol{x})\right\|_2^2}$$

$$\leq \sup_{\boldsymbol{x} \in \mathbb{X}} \frac{\beta^{[j]2} + \prod_{\widetilde{j} \in \mathbb{A}^{[j]}} \omega^{[\widetilde{j}:j]2} \left\|\boldsymbol{\phi}^{[\widetilde{j}:j]}(\boldsymbol{x})\right\|_2^2}{\beta^{[j]2} + \prod_{\widetilde{j} \in \mathbb{A}^{[j]}} \omega^{[\widetilde{j}:j]2} \left\|\boldsymbol{\phi}^{[\widetilde{j}:j]}(\boldsymbol{x})\right\|_2^2} \max\left\{\gamma^{[j]}, \prod_{\widetilde{j} \in \mathbb{A}^{[j]}} C_{\Theta,\eta}^{[\widetilde{j}:j]2}\right\}$$

$$= \max\left\{\gamma^{[j]}, \prod_{\widetilde{j} \in \mathbb{A}^{[j]}} C_{\Theta,\eta}^{[\widetilde{j}:j]2}\right\}$$

and:

$$\sup_{\Theta \in \mathbb{W}} \frac{\left\|\left\langle \boldsymbol{\Psi}^{[j]}(\Theta), \boldsymbol{\phi}^{[j]}(\boldsymbol{x})\right\rangle_{\mathbf{g}^{[j]}}\right\|_2^2}{\left\|\boldsymbol{\Psi}^{[j]}(\Theta)\right\|_F^2} \leq \sup_{\Theta \in \mathbb{W}} \frac{\left\|\mathbf{b}^{[j]}\right\|_2^2 + \prod_{\widetilde{j} \in \mathbb{A}^{[j]}} \left\|\mathbf{W}^{[\widetilde{j}:j]}\right\|_2^2 \left\|\boldsymbol{\Psi}^{[\widetilde{j}:j]}(\Theta)\right\|_F^2 C_{\boldsymbol{x},\eta}^{[\widetilde{j}:j]2}}{\left\|\mathbf{b}^{[j]}\right\|_2^2 + \prod_{\widetilde{j} \in \mathbb{A}^{[j]}} \left\|\mathbf{W}^{[\widetilde{j}:j]}\right\|_2^2 \left\|\boldsymbol{\phi}^{[\widetilde{j}:j]}(\boldsymbol{x})\right\|_2^2}$$

$$\leq \frac{\left\|\mathbf{b}^{[j]}\right\|_2^2 + \prod_{\widetilde{j} \in \mathbb{A}^{[j]}} \left\|\mathbf{W}^{[\widetilde{j}:j]}\right\|_2^2 \left\|\boldsymbol{\phi}^{[\widetilde{j}:j]}(\boldsymbol{x})\right\|_2^2}{\left\|\mathbf{b}^{[j]}\right\|_2^2 + \prod_{\widetilde{j} \in \mathbb{A}^{[j]}} \left\|\mathbf{W}^{[\widetilde{j}:j]}\right\|_2^2 \left\|\boldsymbol{\phi}^{[\widetilde{j}:j]}(\boldsymbol{x})\right\|_2^2} \max\left\{\gamma^{[j]}, \prod_{\widetilde{j} \in \mathbb{A}^{[j]}} C_{\boldsymbol{x},\eta}^{[\widetilde{j}:j]2}\right\}$$

$$\leq \max\left\{\gamma^{[j]}, \prod_{\widetilde{j} \in \mathbb{A}^{[j]}} C_{\boldsymbol{x},\eta}^{[\widetilde{j}:j]2}\right\}$$

For Hadamard product nodes $\bigcirc^{[j]} = \bigodot$ (so $\square^{[j]} = \bigotimes^{\ddagger}$, $\boxdot^{[j]} = \bigotimes$), using that the norm of the

Hadamard product of unit vectors is $\leq 1$:

$$\sup_{\boldsymbol{x}\in\mathbb{X}} \frac{\left\|\left\langle\boldsymbol{\Psi}^{[j]}(\Theta),\boldsymbol{\phi}^{[j]}(\boldsymbol{x})\right]_{\mathbf{g}^{[j]}}\right\|_2^2}{\left\|\boldsymbol{\phi}^{[j]}(\boldsymbol{x})\right\|_2^2} = \sup_{\boldsymbol{x}\in\mathbb{X}} \frac{\left\|\boldsymbol{x}^{[j]}\right\|_2^2}{\left\|\boldsymbol{\phi}^{[j]}(\boldsymbol{x})\right\|_2^2} = \sup_{\boldsymbol{x}\in\mathbb{X}} \frac{\left\|\mathbf{b}^{[j]}+\bigodot_{\widetilde{\jmath}\in\mathbb{A}^{[j]}}\mathbf{W}^{[\widetilde{\jmath};j]\mathrm{T}}\boldsymbol{x}^{[\widetilde{\jmath};j]}\right\|_2^2}{\left\|\boldsymbol{\phi}^{[j]}(\boldsymbol{x})\right\|_2^2}$$

$$= \sup_{\boldsymbol{x}\in\mathbb{X}} \frac{\left\|\mathbf{b}^{[j]}+\bigodot_{\widetilde{\jmath}\in\mathbb{A}^{[j]}}\mathbf{W}^{[\widetilde{\jmath};j]\mathrm{T}}\boldsymbol{x}^{[\widetilde{\jmath};j]}\right\|_2^2}{\beta^{[j]2}+\prod_{\widetilde{\jmath}\in\mathbb{A}^{[j]}}\omega^{[\widetilde{\jmath};j]2}\left\|\boldsymbol{\phi}^{[\widetilde{\jmath};j]}(\boldsymbol{x})\right\|_2^2}$$

$$\leq \sup_{\boldsymbol{x}\in\mathbb{X}} \frac{\left\|\mathbf{b}^{[j]}\right\|_2^2+\prod_{\widetilde{\jmath}\in\mathbb{A}^{[j]}}\left\|\mathbf{W}^{[\widetilde{\jmath};j]\mathrm{T}}\boldsymbol{x}^{[\widetilde{\jmath};j]}\right\|_2^2}{\beta^{[j]2}+\prod_{\widetilde{\jmath}\in\mathbb{A}^{[j]}}\omega^{[\widetilde{\jmath};j]2}\left\|\boldsymbol{\phi}^{[\widetilde{\jmath};j]}(\boldsymbol{x})\right\|_2^2}$$

$$= \sup_{\boldsymbol{x}\in\mathbb{X}} \frac{\left\|\mathbf{b}^{[j]}\right\|_2^2+\prod_{\widetilde{\jmath}\in\mathbb{A}^{[j]}}\left\|\mathbf{W}^{[\widetilde{\jmath};j]\mathrm{T}}\left\langle\boldsymbol{\Psi}^{[\widetilde{\jmath};j]}(\Theta),\boldsymbol{\phi}^{[\widetilde{\jmath};j]}(\boldsymbol{x})\right]_{\mathbf{g}^{[\widetilde{\jmath};j]}}\right\|_2^2}{\beta^{[j]2}+\prod_{\widetilde{\jmath}\in\mathbb{A}^{[j]}}\omega^{[\widetilde{\jmath};j]2}\left\|\boldsymbol{\phi}^{[\widetilde{\jmath};j]}(\boldsymbol{x})\right\|_2^2}$$

$$\leq \sup_{\boldsymbol{x}\in\mathbb{X}} \frac{\left\|\mathbf{b}^{[j]}\right\|_2^2+\prod_{\widetilde{\jmath}\in\mathbb{A}^{[j]}}\left\|\mathbf{W}^{[\widetilde{\jmath};j]}\right\|_2^2\left\|\boldsymbol{\phi}^{[\widetilde{\jmath};j]}(\boldsymbol{x})\right\|_2^2 C_{\Theta,\eta}^{[\widetilde{\jmath};j]2}}{\beta^{[j]2}+\prod_{\widetilde{\jmath}\in\mathbb{A}^{[j]}}\omega^{[\widetilde{\jmath};j]2}\left\|\boldsymbol{\phi}^{[\widetilde{\jmath};j]}(\boldsymbol{x})\right\|_2^2}$$

$$\leq \sup_{\boldsymbol{x}\in\mathbb{X}} \frac{\beta^{[j]2}+\prod_{\widetilde{\jmath}\in\mathbb{A}^{[j]}}\omega^{[\widetilde{\jmath};j]2}\left\|\boldsymbol{\phi}^{[\widetilde{\jmath};j]}(\boldsymbol{x})\right\|_2^2}{\beta^{[j]2}+\prod_{\widetilde{\jmath}\in\mathbb{A}^{[j]}}\omega^{[\widetilde{\jmath};j]2}\left\|\boldsymbol{\phi}^{[\widetilde{\jmath};j]}(\boldsymbol{x})\right\|_2^2} \max\left\{\gamma^{[j]},\prod_{\widetilde{\jmath}\in\mathbb{A}^{[j]}}C_{\Theta,\eta}^{[\widetilde{\jmath};j]2}\right\}$$

$$= \max\left\{\gamma^{[j]},\prod_{\widetilde{\jmath}\in\mathbb{A}^{[j]}}C_{\Theta,\eta}^{[\widetilde{\jmath};j]2}\right\}$$

and:

$$\sup_{\Theta\in\mathbb{W}} \frac{\left\|\left\langle\boldsymbol{\Psi}^{[j]}(\Theta),\boldsymbol{\phi}^{[j]}(\boldsymbol{x})\right]_{\mathbf{g}^{[j]}}\right\|_2^2}{\left\|\boldsymbol{\Psi}^{[j]}(\Theta)\right\|_F^2} \leq \sup_{\Theta\in\mathbb{W}} \frac{\left\|\mathbf{b}^{[j]}\right\|_2^2+\prod_{\widetilde{\jmath}\in\mathbb{A}^{[j]}}\left\|\mathbf{W}^{[\widetilde{\jmath};j]}\right\|_2^2\left\|\boldsymbol{\Psi}^{[\widetilde{\jmath};j]}(\Theta)\right\|_F^2 C_{\boldsymbol{x},\eta}^{[\widetilde{\jmath};j]2}}{\left\|\mathbf{b}^{[j]}\right\|_2^2+\prod_{\widetilde{\jmath}\in\mathbb{A}^{[j]}}\left\|\mathbf{W}^{[\widetilde{\jmath};j]}\right\|_2^2\left\|\boldsymbol{\phi}^{[\widetilde{\jmath};j]}(\boldsymbol{x})\right\|_2^2}$$

$$\leq \frac{\left\|\mathbf{b}^{[j]}\right\|_2^2+\prod_{\widetilde{\jmath}\in\mathbb{A}^{[j]}}\left\|\mathbf{W}^{[\widetilde{\jmath};j]}\right\|_2^2\left\|\boldsymbol{\phi}^{[\widetilde{\jmath};j]}(\boldsymbol{x})\right\|_2^2}{\left\|\mathbf{b}^{[j]}\right\|_2^2+\prod_{\widetilde{\jmath}\in\mathbb{A}^{[j]}}\left\|\mathbf{W}^{[\widetilde{\jmath};j]}\right\|_2^2\left\|\boldsymbol{\phi}^{[\widetilde{\jmath};j]}(\boldsymbol{x})\right\|_2^2} \max\left\{\gamma^{[j]},\prod_{\widetilde{\jmath}\in\mathbb{A}^{[j]}}C_{\boldsymbol{x},\eta}^{[\widetilde{\jmath};j]2}\right\}$$

$$\leq \max\left\{\gamma^{[j]},\prod_{\widetilde{\jmath}\in\mathbb{A}^{[j]}}C_{\boldsymbol{x},\eta}^{[\widetilde{\jmath};j]2}\right\}$$

For multi-inner-product nodes $\bigcirc^{[j]}=\langle\!\langle\cdot\rangle\!\rangle$ (so $\square^{[j]}=\bigotimes^{\updownarrow}(\cdot)\,\mathbf{1}$, $\boxdot^{[j]}=\bigotimes$), using that the multi-inner-product of (2-norm) unit vectors is at most 1:

$$\sup_{\boldsymbol{x}\in\mathbb{X}} \frac{\left\|\left\langle\boldsymbol{\Psi}^{[j]}(\Theta),\boldsymbol{\phi}^{[j]}(\boldsymbol{x})\right]_{\mathbf{g}^{[j]}}\right\|_2^2}{\left\|\boldsymbol{\phi}^{[j]}(\boldsymbol{x})\right\|_2^2} = \sup_{\boldsymbol{x}\in\mathbb{X}} \frac{\left\|\boldsymbol{x}^{[j]}\right\|_2^2}{\left\|\boldsymbol{\phi}^{[j]}(\boldsymbol{x})\right\|_2^2} = \sup_{\boldsymbol{x}\in\mathbb{X}} \frac{\left\|\mathbf{b}^{[j]}+\langle\!\langle\mathbf{W}^{[\widetilde{\jmath};j]\mathrm{T}}\boldsymbol{x}^{[\widetilde{\jmath};j]}\rangle\!\rangle_{\widetilde{\jmath}\in\mathbb{A}^{[j]}}\right\|_2^2}{\left\|\boldsymbol{\phi}^{[j]}(\boldsymbol{x})\right\|_2^2}$$

$$= \sup_{\boldsymbol{x}\in\mathbb{X}} \frac{\left\|\mathbf{b}^{[j]}+\langle\!\langle\mathbf{W}^{[\widetilde{\jmath};j]\mathrm{T}}\boldsymbol{x}^{[\widetilde{\jmath};j]}\rangle\!\rangle_{\widetilde{\jmath}\in\mathbb{A}^{[j]}}\right\|_2^2}{\beta^{[j]2}+\prod_{\widetilde{\jmath}\in\mathbb{A}^{[j]}}\omega^{[\widetilde{\jmath};j]2}\left\|\boldsymbol{\phi}^{[\widetilde{\jmath};j]}(\boldsymbol{x})\right\|_2^2}$$

$$\leq \sup_{\boldsymbol{x}\in\mathbb{X}} \frac{\left\|\mathbf{b}^{[j]}\right\|_2^2+\prod_{\widetilde{\jmath}\in\mathbb{A}^{[j]}}\left\|\mathbf{W}^{[\widetilde{\jmath};j]\mathrm{T}}\boldsymbol{x}^{[\widetilde{\jmath};j]}\right\|_2^2}{\beta^{[j]2}+\prod_{\widetilde{\jmath}\in\mathbb{A}^{[j]}}\omega^{[\widetilde{\jmath};j]2}\left\|\boldsymbol{\phi}^{[\widetilde{\jmath};j]}(\boldsymbol{x})\right\|_2^2}$$

$$= \sup_{\boldsymbol{x}\in\mathbb{X}} \frac{\left\|\mathbf{b}^{[j]}\right\|_2^2+\prod_{\widetilde{\jmath}\in\mathbb{A}^{[j]}}\left\|\mathbf{W}^{[\widetilde{\jmath};j]\mathrm{T}}\left\langle\boldsymbol{\Psi}^{[\widetilde{\jmath};j]}(\Theta),\boldsymbol{\phi}^{[\widetilde{\jmath};j]}(\boldsymbol{x})\right]_{\mathbf{g}^{[\widetilde{\jmath};j]}}\right\|_2^2}{\beta^{[j]2}+\prod_{\widetilde{\jmath}\in\mathbb{A}^{[j]}}\omega^{[\widetilde{\jmath};j]2}\left\|\boldsymbol{\phi}^{[\widetilde{\jmath};j]}(\boldsymbol{x})\right\|_2^2}$$

$$\leq \sup_{\boldsymbol{x}\in\mathbb{X}} \frac{\left\|\mathbf{b}^{[j]}\right\|_2^2+\prod_{\widetilde{\jmath}\in\mathbb{A}^{[j]}}\left\|\mathbf{W}^{[\widetilde{\jmath};j]}\right\|_2^2\left\|\boldsymbol{\phi}^{[\widetilde{\jmath};j]}(\boldsymbol{x})\right\|_2^2 C_{\Theta,\eta}^{[\widetilde{\jmath};j]2}}{\beta^{[j]2}+\prod_{\widetilde{\jmath}\in\mathbb{A}^{[j]}}\omega^{[\widetilde{\jmath};j]2}\left\|\boldsymbol{\phi}^{[\widetilde{\jmath};j]}(\boldsymbol{x})\right\|_2^2}$$

$$\leq \sup_{\boldsymbol{x}\in\mathbb{X}} \frac{\beta^{[j]2}+\prod_{\widetilde{\jmath}\in\mathbb{A}^{[j]}}\omega^{[\widetilde{\jmath};j]2}\left\|\boldsymbol{\phi}^{[\widetilde{\jmath};j]}(\boldsymbol{x})\right\|_2^2}{\beta^{[j]2}+\prod_{\widetilde{\jmath}\in\mathbb{A}^{[j]}}\omega^{[\widetilde{\jmath};j]2}\left\|\boldsymbol{\phi}^{[\widetilde{\jmath};j]}(\boldsymbol{x})\right\|_2^2} \max\left\{\gamma^{[j]},\prod_{\widetilde{\jmath}\in\mathbb{A}^{[j]}}C_{\Theta,\eta}^{[\widetilde{\jmath};j]2}\right\}$$

$$= \max\left\{\gamma^{[j]},\prod_{\widetilde{\jmath}\in\mathbb{A}^{[j]}}C_{\Theta,\eta}^{[\widetilde{\jmath};j]2}\right\}$$

and:

$$\sup_{\Theta\in\mathbb{W}}\frac{\left\|\left\langle\boldsymbol{\Psi}^{[j]}(\Theta),\boldsymbol{\phi}^{[j]}(\boldsymbol{x})\right]_{\mathbf{g}^{[j]}}\right\|_2^2}{\left\|\boldsymbol{\Psi}^{[j]}(\Theta)\right\|_F^2}\leq\sup_{\Theta\in\mathbb{W}}\frac{\left\|\mathbf{b}^{[j]}\right\|_2^2+\prod_{\widetilde{\jmath}\in\mathbb{A}^{[j]}}\left\|\mathbf{W}^{[\widetilde{\jmath}:j]}\right\|_2^2\left\|\boldsymbol{\Psi}^{[\widetilde{\jmath}:j]}(\Theta)\right\|_F^2 C_{\boldsymbol{x},\eta}^{[\widetilde{\jmath}:j]2}}{\left\|\mathbf{b}^{[j]}\right\|_2^2+\prod_{\widetilde{\jmath}\in\mathbb{A}^{[j]}}\left\|\mathbf{W}^{[\widetilde{\jmath}:j]}\right\|_2^2\left\|\boldsymbol{\phi}^{[\widetilde{\jmath}:j]}(\boldsymbol{x})\right\|_2^2}$$

$$\leq\frac{\left\|\mathbf{b}^{[j]}\right\|_2^2+\prod_{\widetilde{\jmath}\in\mathbb{A}^{[j]}}\left\|\mathbf{W}^{[\widetilde{\jmath}:j]}\right\|_2^2\left\|\boldsymbol{\phi}^{[\widetilde{\jmath}:j]}(\boldsymbol{x})\right\|_2^2}{\left\|\mathbf{b}^{[j]}\right\|_2^2+\prod_{\widetilde{\jmath}\in\mathbb{A}^{[j]}}\left\|\mathbf{W}^{[\widetilde{\jmath}:j]}\right\|_2^2\left\|\boldsymbol{\phi}^{[\widetilde{\jmath}:j]}(\boldsymbol{x})\right\|_2^2}\max\left\{\gamma^{[j]},\prod_{\widetilde{\jmath}\in\mathbb{A}^{[j]}}C_{\boldsymbol{x},\eta}^{[\widetilde{\jmath}:j]2}\right\}$$

$$\leq\max\left\{\gamma^{[j]},\prod_{\widetilde{\jmath}\in\mathbb{A}^{[j]}}C_{\boldsymbol{x},\eta}^{[\widetilde{\jmath}:j]2}\right\}$$

Thus in general, for all nodes considered here:

$$\left.\begin{array}{ll}\sup_{\boldsymbol{x}\in\mathbb{X}}\dfrac{\left\|\left\langle\boldsymbol{\Psi}^{[j]}(\Theta),\boldsymbol{\phi}^{[j]}(\boldsymbol{x})\right]_{\mathbf{g}^{[j]}}\right\|_2^2}{\left\|\boldsymbol{\phi}^{[j]}(\boldsymbol{x})\right\|_2^2}\leq C_{\Theta,\eta}^{[j]2}\triangleq\max\left\{\gamma^{[j]},\bigdiamond^{[j]}_{\widetilde{\jmath}\in\mathbb{A}^{[j]}}C_{\Theta,\eta}^{[\widetilde{\jmath}:j]2}\right\}&\text{for given }\Theta\in\mathbb{W}\\[18pt]\sup_{\boldsymbol{x}\in\mathbb{X}}\dfrac{\left\|\left\langle\boldsymbol{\Psi}^{[j]}(\Theta),\boldsymbol{\phi}^{[j]}(\boldsymbol{x})\right]_{\mathbf{g}^{[j]}}\right\|_2^2}{\left\|\boldsymbol{\phi}^{[j]}(\boldsymbol{x})\right\|_2^2}\leq C_{\mathbb{W},\eta}^{[j]2}\triangleq\max\left\{\gamma^{[j]},\bigdiamond^{[j]}_{\widetilde{\jmath}\in\mathbb{A}^{[j]}}C_{\mathbb{W},\eta}^{[\widetilde{\jmath}:j]2}\right\}&\forall\Theta\in\mathbb{W}\\[18pt]\sup_{\Theta\in\mathbb{W}}\dfrac{\left\|\left\langle\boldsymbol{\Psi}^{[j]}(\Theta),\boldsymbol{\phi}^{[j]}(\boldsymbol{x})\right]_{\mathbf{g}^{[j]}}\right\|_2^2}{\left\|\boldsymbol{\Psi}^{[j]}(\Theta)\right\|_F^2}\leq C_{\boldsymbol{x},\eta}^{[j]2}\triangleq\max\left\{\gamma^{[j]},\bigdiamond^{[j]}_{\widetilde{\jmath}\in\mathbb{A}^{[j]}}C_{\boldsymbol{x},\eta}^{[\widetilde{\jmath}:j]2}\right\}&\text{for given }\boldsymbol{x}\in\mathbb{X}\\[18pt]\sup_{\Theta\in\mathbb{W}}\dfrac{\left\|\left\langle\boldsymbol{\Psi}^{[j]}(\Theta),\boldsymbol{\phi}^{[j]}(\boldsymbol{x})\right]_{\mathbf{g}^{[j]}}\right\|_2^2}{\left\|\boldsymbol{\Psi}^{[j]}(\Theta)\right\|_F^2}\leq C_{\mathbb{X},\eta}^{[j]2}\triangleq\max\left\{\gamma^{[j]},\bigdiamond^{[j]}_{\widetilde{\jmath}\in\mathbb{A}^{[j]}}C_{\mathbb{X},\eta}^{[\widetilde{\jmath}:j]2}\right\}&\forall\boldsymbol{x}\in\mathbb{X}\end{array}\right\}\begin{array}{l}C_{\Theta,\eta}^{[j]}\leq C_{\mathbb{W},\eta}^{[j]}\\[36pt]C_{\boldsymbol{x},\eta}^{[j]}\leq C_{\mathbb{X},\eta}^{[j]}\end{array}$$

where we have defined:

$$\bigdiamond^{[j]}=\left\{\begin{array}{ll}\max&\text{if }\bigcirc^{[j]}\in\{\bigoplus,\sum\}\\\prod&\text{if }\bigcirc^{[j]}\in\{\bigotimes,\langle\!\langle\cdot\rangle\!\rangle\}\end{array}\right.$$

Consequently, defining $C_{\Theta,\eta}=C_{\Theta,\eta}^{[E]}$, $C_{\mathbb{W},\eta}=C_{\mathbb{W},\eta}^{[E]}$, $C_{\boldsymbol{x},\eta}=C_{\boldsymbol{x},\eta}^{[E]}$, and $C_{\mathbb{X},\eta}=C_{\mathbb{X},\eta}^{[E]}$:

$$\left.\begin{array}{ll}\sup_{\boldsymbol{x}\in\mathbb{X}}\dfrac{\left\|\langle\boldsymbol{\Psi}(\Theta),\boldsymbol{\phi}(\boldsymbol{x})]_{\mathbf{g}}\right\|_2^2}{\left\|\boldsymbol{\phi}(\boldsymbol{x})\right\|_2^2}\leq C_{\Theta,\eta}^2&\text{for given }\Theta\in\mathbb{W}\\[14pt]\sup_{\boldsymbol{x}\in\mathbb{X}}\dfrac{\left\|\langle\boldsymbol{\Psi}(\Theta),\boldsymbol{\phi}(\boldsymbol{x})]_{\mathbf{g}}\right\|_2^2}{\left\|\boldsymbol{\phi}(\boldsymbol{x})\right\|_2^2}\leq C_{\mathbb{W},\eta}^2&\forall\Theta\in\mathbb{W}\end{array}\right\}\quad C_{\Theta,\eta}\leq C_{\mathbb{W},\eta}$$

$$\left.\begin{array}{ll}\sup_{\Theta\in\mathbb{W}}\dfrac{\left\|\langle\boldsymbol{\Psi}(\Theta),\boldsymbol{\phi}(\boldsymbol{x})]_{\mathbf{g}}\right\|_2^2}{\left\|\boldsymbol{\Psi}(\Theta)\right\|_F^2}\leq C_{\boldsymbol{x},\eta}^2&\text{for given }\boldsymbol{x}\in\mathbb{X}\\[14pt]\sup_{\Theta\in\mathbb{W}}\dfrac{\left\|\langle\boldsymbol{\Psi}(\Theta),\boldsymbol{\phi}(\boldsymbol{x})]_{\mathbf{g}}\right\|_2^2}{\left\|\boldsymbol{\Psi}(\Theta)\right\|_F^2}\leq C_{\mathbb{X},\eta}^2&\forall\boldsymbol{x}\in\mathbb{X}\end{array}\right\}\quad C_{\boldsymbol{x},\eta}\leq C_{\mathbb{X},\eta}$$

where $C_{\mathbb{W},\eta}$ is finite in general and $C_{\mathbb{X},\eta}$ is finite if all neural activations are Lipschitz.

The limit case $\eta\to0^+$ is of particular interest here. Defining $C_\Theta=\lim_{\eta\to0^+}C_{\Theta,\eta}$, $C_\Theta^{[j]}=\lim_{\eta\to0^+}C_{\Theta,\eta}^{[j]}$, $C_\mathbb{W}=\lim_{\eta\to0^+}C_{\mathbb{W},\eta}$, $C_\mathbb{W}^{[j]}=\lim_{\eta\to0^+}C_{\mathbb{W},\eta}^{[j]}$, we observe that, using the form of the base case and recursion:

$$\begin{array}{ll}C_\Theta=C_\Theta^{[j]}=C_\mathbb{W}=C_\mathbb{W}^{[j]}=1&\text{if all neural activations are Lipschitz or bounded}\\C_{\boldsymbol{x}}=C_{\boldsymbol{x}}^{[j]}=C_\mathbb{X}=C_\mathbb{X}^{[j]}=1&\text{if all neural activations are Lipschitz}\end{array}\quad\forall j\in\mathbb{Z}_E$$

This result, combined with Theorem 1, suffices to prove Corollaries 2 and 3.

## C.4  Bounds for Data Drawn from a Distribution

A common variation of our assumption $\boldsymbol{x}\in\mathbb{X}_{\rho,r}$ - that is, the assumption that $\boldsymbol{x}$ is hard-limited in terms of its 2-norm - is that $\boldsymbol{x}\sim\mathcal{X}$ is drawn from some data distribution $\mathcal{X}$. With regard to our analysis, for arbitrary data distributions it is not possible to extend our analysis; however if it can be proven that $\boldsymbol{x}\in\mathbb{X}_{\rho,r}$ with-high-probability $\geq1-\epsilon$ for suitable $\rho,r$ then our results will follow whp $\geq1-\epsilon$. To take a simple example, suppose we draw data from an $n$-dimensional normal distribution:

$$\boldsymbol{x}\sim\mathcal{X}=\mathcal{N}\left(\mathbf{0}_n,\sigma^2\mathbf{I}_n\right)$$

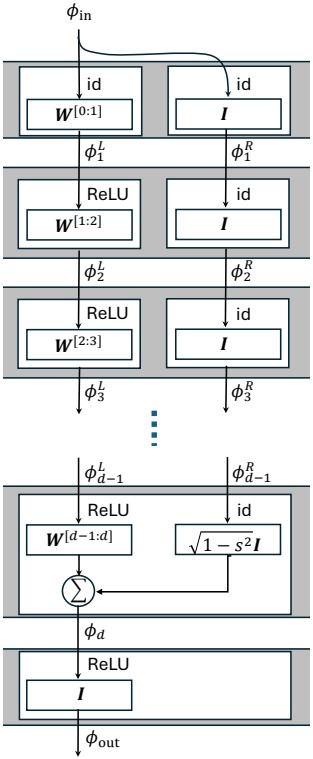

Figure 7: Calculation of $\phi_{\text{out}}$ in a residual network block.

Trivially, for $\boldsymbol{x} \sim \mathcal{X}$:

$$\Pr\left[\|\boldsymbol{x}\|_2 \leq \rho\right] \leq \frac{1}{2^{n/2}\sigma\Gamma\left(\frac{n}{2}+1\right)}\rho^n$$

$$\Pr\left[\|\boldsymbol{x}\|_2 \geq r\right] \leq 2e^{-\frac{r^2}{2n\sigma^2}}$$

Thus we have $\boldsymbol{x} \in \mathbb{X}_{\rho,r}$ with high probability $\geq 1 - \epsilon$, where:

$$r = \sqrt{2n\ln\left(\frac{2}{(1-\upsilon)\epsilon}\right)}\sigma$$

$$\rho = \sqrt{2}\left(\Gamma\left(\frac{n}{2}+1\right)\upsilon\epsilon\right)^{\frac{1}{n}}\sigma$$

for some $\upsilon \in [0, 1)$, In the purely Lipschitz case we can simplify this by setting $\upsilon = 0$ (so $\rho = 0$):

$$r = \sqrt{2n\ln\left(\frac{2}{\epsilon}\right)}\sigma \tag{33}$$

and more generally, if we allow non-Lipschitz neural activations, whp $\geq 1 - \epsilon$:

$$\frac{r}{\rho} = \frac{\sqrt{n\ln\left(\frac{2}{(1-\upsilon)\epsilon}\right)}}{\left(\Gamma\left(\frac{n}{2}+1\right)\upsilon\epsilon\right)^{\frac{1}{n}}} \tag{34}$$

# D   Non-Trivial Blocks

In this section we consider norm- and continuity- bounds for particular common neural network archictectural blocks. Note that in all cases the continuity bounds $C_\Theta, C_\mathbb{W}, C_{\boldsymbol{x}}, C_\mathbb{X}$ are well-behaved, so our task is to analyse the norm-bound $\phi$. In this regard we refer the reader to (21) in Figure 6.

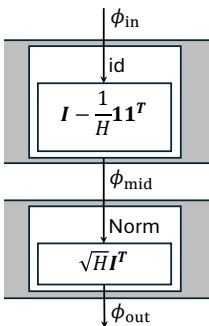

Figure 8: Calculation of $\phi_{\text{out}}$ in a residual network block.

## D.1 Residual Block Bounds

In this section we consider the calculation of $\phi$ for a residual block. Figure 7 shows the notation we use here. All neural activations in this block are 1-Lipschitz so trivially, using our bounds:

$$\psi_{d-1}^{\text{R}} = \psi_{d-2}^{\text{R}} = \psi_{d-3}^{\text{R}} = \ldots = \psi_2^{\text{R}} = \psi_1^{\text{R}} = \psi_{\text{in}}$$
$$\phi_{d-1\downarrow}^{\text{R}} = \phi_{d-2\downarrow}^{\text{R}} = \phi_{d-3\downarrow}^{\text{R}} = \ldots = \phi_{2\downarrow}^{\text{R}} = \phi_{1\downarrow}^{\text{R}} = \phi_{\text{in}\downarrow}$$
$$\phi_{d-1}^{\text{R}} = \phi_{d-2}^{\text{R}} = \phi_{d-3}^{\text{R}} = \ldots = \phi_2^{\text{R}} = \phi_1^{\text{R}} = \phi_{\text{in}}$$

and:

$$\begin{aligned}
\psi_{d-1}^{\text{L}} &= \omega^{[d-2:d-1]}\psi_{d-2}^{\text{L}}\\
&= \omega^{[d-2:d-1]}\omega^{[d-3:d-2]}\psi_{d-3}^{\text{L}} = \ldots\\
&= \omega^{[d-2:d-1]}\omega^{[d-3:d-2]}\ldots\omega^{[1:2]}\psi_1^{\text{L}}\\
&= \omega^{[d-2:d-1]}\omega^{[d-3:d-2]}\ldots\omega^{[1:2]}\omega^{[0:1]}\psi_{\text{in}}\\
\phi_{d-1\downarrow}^{\text{L}} &= \omega^{[d-2:d-1]}\phi_{d-2\downarrow}^{\text{L}}\\
&= \omega^{[d-2:d-1]}\omega^{[d-3:d-2]}\phi_{d-3\downarrow}^{\text{L}} = \ldots\\
&= \omega^{[d-2:d-1]}\omega^{[d-3:d-2]}\ldots\omega^{[1:2]}\phi_{1\downarrow}^{\text{L}}\\
&= \omega^{[d-2:d-1]}\omega^{[d-3:d-2]}\ldots\omega^{[1:2]}\omega^{[0:1]}\phi_{\text{in}\downarrow}\\
\phi_{d-1}^{\text{L}} &= \omega^{[d-2:d-1]}\phi_{d-2}^{\text{L}}\\
&= \omega^{[d-2:d-1]}\omega^{[d-3:d-2]}\phi_{d-3}^{\text{L}} = \ldots\\
&= \omega^{[d-2:d-1]}\omega^{[d-3:d-2]}\ldots\omega^{[1:2]}\phi_1^{\text{L}}\\
&= \omega^{[d-2:d-1]}\omega^{[d-3:d-2]}\ldots\omega^{[1:2]}\omega^{[0:1]}\phi_{\text{in}}
\end{aligned}$$

and subsequently:

$$\begin{aligned}
\psi_{\text{out}}^2 &= \psi_d^2\\
&= \psi_{d-1}^{\text{L}2} + \left(1 - s^2\right)\psi_{d-1}^{\text{R}2}\\
&= \left(\omega^{[d-1:d]2}\ldots\omega^{[1:2]2}\omega^{[0:1]2} + 1 - s^2\right)\psi_{\text{in}}^2\\
\phi_{\text{out}\downarrow}^2 &= \phi_{d\downarrow}^2\\
&= \phi_{d-1\downarrow}^{\text{L}2} + \left(1 - s^2\right)\phi_{d-1\downarrow}^{\text{R}2}\\
&= \left(\omega^{[d-1:d]2}\ldots\omega^{[1:2]2}\omega^{[0:1]2} + 1 - s^2\right)\phi_{\text{in}\downarrow}^2\\
\phi_{\text{out}}^2 &= \phi_d^2\\
&= \phi_{d-1}^{\text{L}2} + \left(1 - s^2\right)\phi_{d-1}^{\text{R}2}\\
&= \left(\omega^{[d-1:d]2}\ldots\omega^{[1:2]2}\omega^{[0:1]2} + 1 - s^2\right)\phi_{\text{in}}^2
\end{aligned} \tag{35}$$

where we note that, for $\rho > 0$:

$$\frac{\phi_{\text{out}}}{\phi_{\text{out}\downarrow}} = \frac{\phi_{\text{in}}}{\phi_{\text{in}\downarrow}}$$

## D.2 LayerNorm Block Bounds

As shown in Figure 8, the LayerNorm block is distinct insofar as it is non-Lipschitz. First we note that that $\|\sqrt{H}\mathbf{I}\|_2 = \sqrt{H}$, $\|\mathbf{I} - \frac{1}{H}\mathbf{1}\mathbf{1}^{\text{T}}\|_2 = 1$, so we may set $\omega^{[\text{in:mid}]} = 1$, $\omega^{[\text{mid:out}]} = \sqrt{H}$. Noting

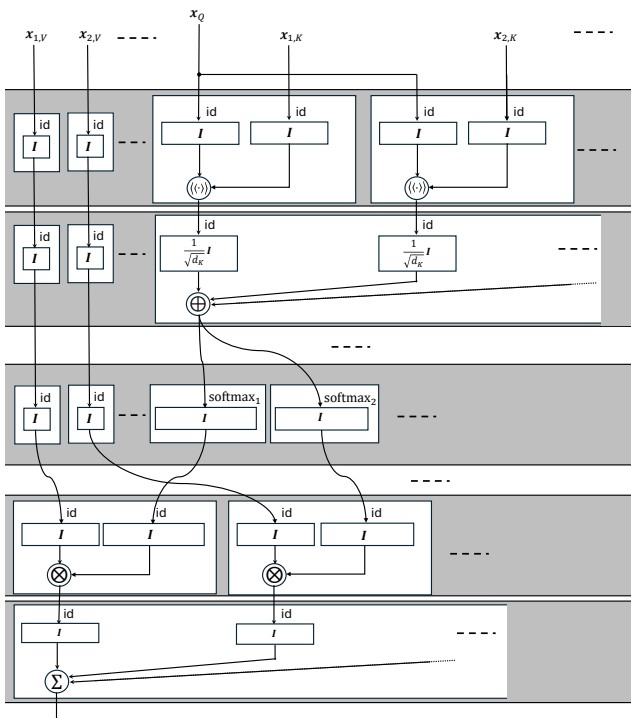

Figure 9: Single-query attention block.

that the Norm activation is non-Lipschitz and bounded by $B^{[\text{Norm}]} = 1$, we see that:

$$\psi_{\text{out}} = \omega^{[\text{mid:out}]} \frac{\psi_{\text{mid}}}{\phi_{\downarrow\text{mid}}} = \sqrt{H} \frac{\psi_{\text{mid}}}{\phi_{\text{mid}\downarrow}}$$
$$\phi_{\text{out}\downarrow} = \omega^{[\text{mid:out}]} = \sqrt{H}$$
$$\phi_{\text{out}} = \omega^{[\text{mid:out}]} \frac{\phi_{\text{mid}}}{\phi_{\downarrow\text{mid}}} = \sqrt{H} \frac{\phi_{\text{mid}}}{\phi_{\text{mid}\downarrow}}$$

and trivially $\psi_{\text{mid}} = \omega^{[\text{in:mid}]} \psi_{\text{in}} = \psi_{\text{in}}$, $\phi_{\text{mid}\downarrow} = \omega^{[\text{in:mid}]} \phi_{\text{in}\downarrow} = \phi_{\text{in}\downarrow}$ and $\phi_{\text{mid}} = \omega^{[\text{in:mid}]} \phi_{\text{in}} = \phi_{\text{in}}$. Hence, overall:

$$\psi_{\text{out}} = \sqrt{H} \frac{\psi_{\text{in}}}{\phi_{\text{in}\downarrow}}$$
$$\phi_{\text{out}\downarrow} = \sqrt{H} \tag{36}$$
$$\phi_{\text{out}} = \sqrt{H} \frac{\phi_{\text{in}}}{\phi_{\text{in}\downarrow}}$$

where we note that, for $\rho > 0$:

$$\frac{\phi_{\text{out}}}{\phi_{\text{out}\downarrow}} = \frac{\phi_{\text{in}}}{\phi_{\text{in}\downarrow}} \tag{37}$$

### D.3 Single-query Attention Block Bounds

The standard bounds as presented in (21) are needlessly pessimistic for softmax nodes in attention blocks (Figure 2) as they are derived without taking into account the operation of the softmax in layer 3, which is a full softmax that has been split into components here - so while we can bound the set of *all* QK outputs, the standard bounds only bound the individual components without taking into account the interaction between then. The following more nuanced analysis gives a tighter bound.

In the following analysis we make the simplifying assumption $\psi_{\eta,\tilde{\imath},V} = \psi_{\eta,V}$, $\phi_{\eta,\tilde{\imath},V\downarrow} = \phi_{\eta,V\downarrow}$, $\phi_{\eta,\tilde{\imath},V} = \phi_{\eta,V}$; $\psi_{\eta,\tilde{\imath},K} = \psi_{\eta,K}$, $\phi_{\eta,\tilde{\imath},K\downarrow} = \phi_{\eta,K\downarrow}$, $\phi_{\eta,\tilde{\imath},K} = \phi_{\eta,K}$. Given this:

Layer 1: following the standard approach:

$$\psi_{\eta,\tilde{\imath},V}^{[1]} = \psi_{\eta,V}$$
$$\phi_{\eta,\tilde{\imath},V\downarrow}^{[1]} = \phi_{\eta,V\downarrow}$$
$$\phi_{\eta,\tilde{\imath},V}^{[1]} = \phi_{\eta,V}$$

$$\psi^{[1]}_{\eta,\widetilde{i},QK} = \psi_{\eta,Q}\psi_{\eta,K}$$
$$\phi^{[1]}_{\eta,\widetilde{i},QK\downarrow} = \phi_{\eta,Q\downarrow}\phi_{\eta,K\downarrow}$$
$$\phi^{[1]}_{\eta,\widetilde{i},QK} = \phi_{\eta,Q}\phi_{\eta,K}$$

Layer 2: following the standard approach:

$$\psi^{[2]}_{\eta,\widetilde{i},V} = \psi^{[1]}_{\eta,\widetilde{i},V} = \psi_{\eta,V}$$
$$\phi^{[2]}_{\eta,\widetilde{i},V\downarrow} = \phi^{[1]}_{\eta,\widetilde{i},V\downarrow} = \phi_{\eta,V\downarrow}$$
$$\phi^{[2]}_{\eta,\widetilde{i},V} = \phi^{[1]}_{\eta,\widetilde{i},V} = \phi_{\eta,V}$$

$$\psi^{[2]2}_{\eta,QK} = \frac{1}{d_K}\sum_{\widetilde{i}}\psi^{[1]2}_{\eta,\widetilde{i},QK} = \psi^2_{\eta,Q}\psi^2_{\eta,K}$$
$$\phi^{[2]2}_{\eta,QK\downarrow} = \frac{1}{d_K}\sum_{\widetilde{i}}\phi^{[1]2}_{\eta,\widetilde{i},QK\downarrow} = \phi^2_{\eta,Q\downarrow}\phi^2_{\eta,K\downarrow}$$
$$\phi^{[2]2}_{\eta,QK} = \frac{1}{d_K}\sum_{\widetilde{i}}\phi^{[1]2}_{\eta,\widetilde{i},QK} = \phi^2_{\eta,Q}\phi^2_{\eta,K}$$

Layer 3: we need to take some care with this layer. In particular, noting that the output of the layer is effectively the softmax split componentwise, we can constrain the sum of $\phi^{[3]}_{\eta,\widetilde{i},QK}$ as:

$$\psi^{[3]}_{\eta,\widetilde{i},V} = \psi^{[2]}_{\eta,\widetilde{i},V} = \psi_{\eta,V}$$
$$\phi^{[3]}_{\eta,\widetilde{i},V\downarrow} = \phi^{[2]}_{\eta,\widetilde{i},V\downarrow} = \phi_{\eta,V\downarrow}$$
$$\phi^{[3]}_{\eta,\widetilde{i},V} = \phi^{[2]}_{\eta,\widetilde{i},V} = \phi_{\eta,V}$$

$$\psi^{[3]2}_{\eta,\widetilde{i},QK} = \lambda^2\frac{s^{[2:3]}_\eta\left(\psi^{[2]2}_{\eta,QK}\right)}{s^{[2:3]}_\eta(1)} = \lambda^2\frac{s^{[2:3]}_\eta\left(\psi^2_{\eta,Q}\psi^2_{\eta,K}\right)}{s^{[2:3]}_\eta(1)}$$
$$\phi^{[3]2}_{\eta,\widetilde{i},QK\downarrow} = c^2_{\widetilde{i}}\lambda^2\frac{s^{[2:3]}_\eta\left(\phi^{[2]2}_{\eta,QK\downarrow}\right)}{s^{[2:3]}_\eta(1)} = c^2_{\widetilde{i}}\lambda^2\frac{s^{[2:3]}_\eta\left(\phi^2_{\eta,Q\downarrow}\phi^2_{\eta,K\downarrow}\right)}{s^{[2:3]}_\eta(1)}$$
$$\phi^{[3]2}_{\eta,\widetilde{i},QK} = c^2_{\widetilde{i}}\lambda^2\frac{s^{[2:3]}_\eta\left(\phi^{[2]2}_{\eta,QK}\right)}{s^{[2:3]}_\eta(1)} = c^2_{\widetilde{i}}\lambda^2\frac{s^{[2:3]}_\eta\left(\phi^2_{\eta,Q}\phi^2_{\eta,K}\right)}{s^{[2:3]}_\eta(1)}$$

for some $c_1, c_2, \ldots \geq 0 : \sum_{\widetilde{i}}c^2_{\widetilde{i}} = 1$ (in the standard analysis we would let $c_1 = c_2 = \ldots = 1$).

Layer 4: following the standard approach:

$$\psi^{[4]2}_{\eta,\widetilde{i}} = \psi^{[3]2}_{\eta,\widetilde{i},V}\psi^{[3]2}_{\eta,\widetilde{i},QK} = \lambda^2\psi^2_{\eta,V}\frac{s^{[2:3]}_\eta\left(\psi^2_{\eta,Q}\psi^2_{\eta,K}\right)}{s^{[2:3]}_\eta(1)}$$
$$\phi^{[4]2}_{\eta,\widetilde{i}\downarrow} = \phi^{[3]2}_{\eta,\widetilde{i},V\downarrow}\phi^{[3]2}_{\eta,\widetilde{i},QK\downarrow} = \lambda^2 c^2_{\widetilde{i}}\phi^2_{\eta,V\downarrow}\frac{s^{[2:3]}_\eta\left(\phi^2_{\eta,Q\downarrow}\phi^2_{\eta,K\downarrow}\right)}{s^{[2:3]}_\eta(1)}$$
$$\phi^{[4]2}_{\eta,\widetilde{i}} = \phi^{[3]2}_{\eta,\widetilde{i},V}\phi^{[3]2}_{\eta,\widetilde{i},QK} = \lambda^2 c^2_{\widetilde{i}}\phi^2_{\eta,V}\frac{s^{[2:3]}_\eta\left(\phi^2_{\eta,Q}\phi^2_{\eta,K}\right)}{s^{[2:3]}_\eta(1)}$$

Layer 5: recalling that $c_1, c_2, \ldots \geq 0$ satisfy $\sum_{\widetilde{i}}c^2_{\widetilde{i}} = 1$:

$$\psi^{[5]2}_\eta = \sum_{\widetilde{i}}\psi^{[4]2}_{\eta,\widetilde{i}} = d_K\lambda^2\psi^2_{\eta,V}\frac{s^{[2:3]}_\eta\left(\psi^2_{\eta,Q}\psi^2_{\eta,K}\right)}{s^{[2:3]}_\eta(1)}$$
$$\phi^{[5]2}_{\eta,\downarrow} = \sum_{\widetilde{i}}\phi^{[4]2}_{\eta,\widetilde{i}\downarrow} = \lambda^2\phi^2_{\eta,V\downarrow}\frac{s^{[2:3]}_\eta\left(\phi^2_{\eta,Q\downarrow}\phi^2_{\eta,K\downarrow}\right)}{s^{[2:3]}_\eta(1)}$$
$$\phi^{[5]2}_\eta = \sum_{\widetilde{i}}\phi^{[4]2}_{\eta,\widetilde{i}} = \lambda^2\phi^2_{\eta,V}\frac{s^{[2:3]}_\eta\left(\phi^2_{\eta,Q}\phi^2_{\eta,K}\right)}{s^{[2:3]}_\eta(1)}$$

Taking the limit $\eta \to 0^+$ we summarise the overall operation of this block as:

$$\psi_{\text{out}} = \sqrt{d_K}\lambda\psi_{\text{in},V}\psi_{\text{in},Q}\psi_{\text{in},K}$$
$$\phi_{\text{out}\downarrow} = \lambda\phi_{\text{in},V\downarrow}\phi_{\text{in},Q\downarrow}\phi_{\text{in},K\downarrow}$$
$$\phi_{\text{out}} = \lambda\phi_{\text{in},V}\phi_{\text{in},Q}\phi_{\text{in},K}$$

where we note that, for $\rho > 0$:

$$\frac{\phi_{\text{out}}}{\phi_{\text{out}\downarrow}} = \frac{\phi_{\text{in},V}}{\phi_{\text{in},V\downarrow}}\frac{\phi_{\text{in},Q}}{\phi_{\text{in},Q\downarrow}}\frac{\phi_{\text{in},K}}{\phi_{\text{in},K\downarrow}}$$

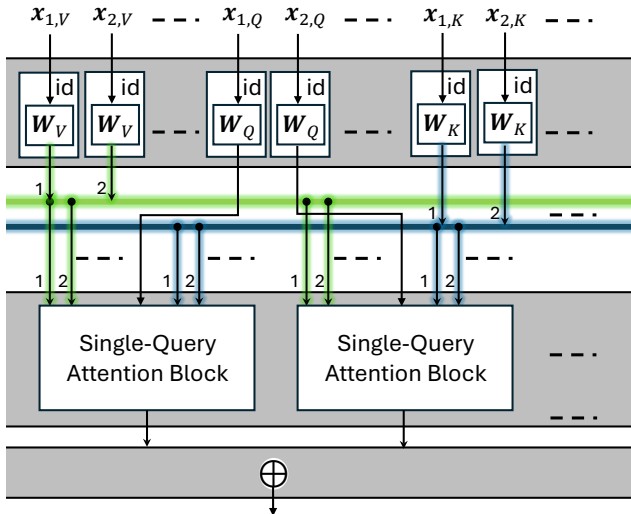

Figure 10: Single-Head attention block.

### D.4 Single-Head and Multi-Head Attention Block Bounds

The standard single-head attention block is constructed from from single-query attention blocks as shown in figure 10. Multi-head attention is similar, with an additional $h$ concatenations. Making the additional assumption, over section D.3, that $\psi_{\eta,\widetilde{\imath},V} = \psi_{\eta,V}$, $\phi_{\eta,\widetilde{\imath},V\downarrow} = \phi_{\eta,V\downarrow}$, $\phi_{\eta,\widetilde{\imath},V} = \phi_{\eta,V}$, it is not difficult to see that:

$$\psi_{\text{out}} = \sqrt{hd_K d_Q}\lambda\psi_{\text{in},V}\psi_{\text{in},Q}\psi_{\text{in},K}$$
$$\phi_{\text{out}\downarrow} = \sqrt{hd_Q}\lambda\phi_{\text{in},V\downarrow}\phi_{\text{in},Q\downarrow}\phi_{\text{in},K\downarrow}$$
$$\phi_{\text{out}} = \sqrt{hd_Q}\lambda\phi_{\text{in},V}\phi_{\text{in},Q}\phi_{\text{in},K}$$

where $d_Q$ is the number of queries and $h$ is the number of heads. We note that, for $\rho > 0$:

$$\frac{\phi_{\text{out}}}{\phi_{\text{out}\downarrow}} = \frac{\phi_{\text{in},V}}{\phi_{\text{in},V\downarrow}}\frac{\phi_{\text{in},Q}}{\phi_{\text{in},Q\downarrow}}\frac{\phi_{\text{in},K}}{\phi_{\text{in},K\downarrow}} \tag{38}$$

## E    Bounds for Standard Network Toplogies

In this section we apply our results, and in particular our norm-bound $\|\phi(\boldsymbol{x})\|_2 \leq \phi \ \forall \boldsymbol{x} \in \mathbb{X}_{\rho,r}$ which is central in our Rademacher complexity bound, to standard network topologies.

### E.1    Simple Unbiased Lipschitz Layerwise Network and ResNet

Consider a simple network with 1 unbiased node with $L$-Lipschitz activations per layer, so $D = E$, $j = \jmath \in \mathbb{Z}_D$, and $\mathbb{A}^{[\jmath]} = \{\jmath - 1\}$. In this case, using (21), $\forall \jmath \in \mathbb{Z}_D$:

$$\phi^{[\jmath]} = L\omega^{[\jmath-1:\jmath]}\phi^{[\jmath-1]}$$

and hence, using that $\phi^{[0]} = r$:

$$\phi = rL^D \prod_{\jmath \in \mathbb{Z}_D} \omega^{[\jmath-1:\jmath]}$$

and we find that the norm-bound $\phi$ (and hence our Rademacher complexity bound) is proportional to the product of the weight-matrix norms, the maximum input norm $r$, and the exponentiated Lipschitz constant. In the distributional case, assuming $\boldsymbol{x} \sim \mathcal{N}(\boldsymbol{0}_n, \sigma^2 \mathbf{I}_n)$ and using (33), whp $\geq 1 - \epsilon$:

$$\phi = \sigma L^D \sqrt{2n \ln\left(\frac{2}{\epsilon}\right)} \prod_{\jmath \in \mathbb{Z}_D} \omega^{[\jmath-1:\jmath]}$$

We can also bound the residual network (ResNet) norm with this by including residual blocks as

nodes in the network. For example, if node $\jmath$ is a residual block then the effective weight-norm bound $\omega^{[\jmath-1:\jmath]}$ becomes, for that non-trivial block, using (35):

$$\omega^{[\jmath-1:\jmath]} = \left(\omega^{[\jmath-1:\jmath]_d 2} \cdots \omega^{[\jmath-1:\jmath]_2 2}\omega^{[\jmath-1:\jmath]_1 2} + 1 - s^2\right)$$

where $\omega^{[\jmath-1:\jmath]_k}$ is the norm-bound for the $k^{\text{th}}$ weight matrix $\mathbf{W}^{[\jmath-1:\jmath]_k}$ in the residual block $\jmath$.

## E.2   Simple Unbiased non-Lipschitz Layerwise Network and LayerNorm

In this section we consider the same network as in the previous section E.1, excepting that we assume at least 1 non-Lipschitz, bounded neural activation. In this case, using (21), $\forall \jmath \in \mathbb{Z}_D$:

$$
\begin{aligned}
\phi_\downarrow^{[\jmath]} &= \omega^{[\jmath-1:\jmath]} \begin{cases} L^{[\jmath-1:\jmath]}_{\phi^{[\jmath-1]}} \phi_\downarrow^{[\jmath-1]} & \text{if } \tau^{[\jmath-1:\jmath]} \text{ is Lipschitz} \\ B^{[\jmath-1:\jmath]}_{\phi^{[\jmath]}} & \text{otherwise} \end{cases} \\
\phi^{[\jmath]} &= \omega^{[\jmath-1:\jmath]} \begin{cases} L^{[\jmath-1:\jmath]}_{\phi^{[\jmath-1]}} \phi^{[\jmath-1]} & \text{if } \tau^{[\jmath-1:\jmath]} \text{ is Lipschitz} \\ B^{[\jmath-1:\jmath]}_{\phi^{[\jmath-1]}} \frac{\phi^{[\jmath-1]}}{\phi_\downarrow^{[\jmath-1]}} & \text{otherwise} \end{cases}
\end{aligned}
\tag{39}
$$

where $\phi_\downarrow^{[0]} = \rho$ and $\phi^{[0]} = r$. We immediately observe that:

$$\frac{\phi^{[D]}}{\phi_\downarrow^{[D]}} = \frac{\phi^{[D-1]}}{\phi_\downarrow^{[D-1]}} = \ldots = \frac{\phi^{[0]}}{\phi_\downarrow^{[0]}} = \frac{r}{\rho} \qquad \forall \jmath \in \mathbb{Z}_D$$

and hence (39) simplifies to:

$$
\begin{aligned}
\phi_\downarrow^{[\jmath]} &= \omega^{[\jmath-1:\jmath]} \begin{cases} L^{[\jmath-1:\jmath]}_{\phi^{[\jmath-1]}} \phi_\downarrow^{[\jmath-1]} & \text{if } \tau^{[\jmath-1:\jmath]} \text{ is Lipschitz} \\ B^{[\jmath-1:\jmath]} & \text{otherwise} \end{cases} \\
\phi^{[\jmath]} &= \omega^{[\jmath-1:\jmath]} \begin{cases} L^{[\jmath-1:\jmath]}_{\phi^{[\jmath-1]}} \phi^{[\jmath-1]} & \text{if } \tau^{[\jmath-1:\jmath]} \text{ is Lipschitz} \\ B^{[\jmath-1:\jmath]} \frac{r}{\rho} & \text{otherwise} \end{cases}
\end{aligned}
\tag{40}
$$

If we further assume that node $\jmath = D_\downarrow$ is the non-Lipschitz node closest to the output node, bounded by $B^{[D_\downarrow-1:D_\downarrow]}$, the norm bound becomes:

$$\phi = \frac{r}{\rho} B^{[D_\downarrow-1:D_\downarrow]} L^{D-D_\downarrow} \prod_{\jmath=D_\downarrow}^{D} \omega^{[\jmath-1:\jmath]}$$

The first thing to note with this bound is that it is no longer depth exponential, but rather depth-to-non-Lipschitz $(D - D_\downarrow)$ exponential. This may appear surprising at first, but it is perhaps not so surprising when we note that the Lipschitz norm-bound scales with the max weight-matrix norm-bound, while a bounded neural-activation displays attributes that, in a crude sense, *flatten out* the magnitude of their input from previous layers. The obvious extreme case is a network combining ReLU and LayerNorm nodes, in which case we can scale weight matrices preceeding the LayerNorm arbitrarily without affecting the operation of the network in any way. This is directly reflected in the above expression, where the norm-bound $\phi$ is *independent* of the magnitude (matrix norm) of the weight-matrices in layers $1, 2, \ldots, D_\downarrow - 1$ before the LayerNorm.

The ratio $\frac{r}{\rho}$ in the bound is perhaps less intuitive. In particular, while we would expect that the norm bound of $\|\phi(\boldsymbol{x})\|_2$ should scale (increase) as $\|\boldsymbol{x}\|_2 \leq r$ increases (which the norm-bound does), it is less obvious that the bound should *increase* as the lower bound $\|\boldsymbol{x}\|_2 \geq \rho$ *decreases*. To understand this behaviour, recall that we only characterise neural activation $\tau^{[D_\downarrow-1:D_\downarrow]}$ by its upper bound $B^{[D_\downarrow-1:D_\downarrow]}$ (1 for simplicity), so we must make a worst-case assumption that $\|\boldsymbol{x}^{[D_\downarrow]}\|_2 = 1$ for all $\boldsymbol{x} \in \mathbb{X}_{\rho,r}$. If $\|\boldsymbol{x}\|_2 = \rho$ then, in our worst-case analysis, the node must, in effect, *amplify* the input so that $\|\boldsymbol{x}^{[D_\downarrow]}\|_2 = 1$; the smaller $\rho$, the larger the amplification required.[9] This is why we take care not to over-claim in the case $\rho = r = 1$ in the main body of the paper.

Another apparent difficulty with this norm-bound is that one may argue that the lower bound $\|\boldsymbol{x}\|_2 \geq \rho$ is artificial, and that real data may not satisfy this bound. To cover this, we may use the distributional

---

[9]In the limit $\rho \to 0^+$ the amplication must approach $\infty$, which is why we insist $\rho > 0$ in this case.

case. For example, if node $D_\downarrow$ is a LayerNorm node and assuming $\boldsymbol{x} \sim \mathcal{N}(\mathbf{0}_n, \sigma^2 \mathbf{I})$ then, using (36) and (34), with high probability $\geq 1 - \epsilon$:

$$\phi = \frac{\sqrt{nH \ln\left(\frac{4}{\epsilon}\right)}}{\left(\Gamma\left(\frac{n}{2}+1\right)\frac{\epsilon}{2}\right)^{\frac{1}{n}}} L^{D-D_\downarrow} \prod_{j=D_\downarrow}^{D} \omega^{[j-1:j]}$$

We observe that this bound is scale-independent, both in terms of the "size" $\sigma$ of the data distribution and weight-norm bounds for layers prior to the final non-Lipschitz node $D_\downarrow$. The proportionality to $\sqrt{H}$ arises from the choice of LayerNorm, and the exact form of the new scaling arises from our choice of distribution.[10]

### E.3 Transformers

Finally we may consider the Transformer. For concreteness we will assume the structure described in (Vaswani et al., 2017, Figure 1); and for tractability we will ignore the input/output embedding and positional encoding, and instead assume inputs and outputs (post-embedding/encoding) $\boldsymbol{x}_I, \boldsymbol{x}_O \in \mathbb{X}_{\rho,r} = \{\boldsymbol{x} \in \mathbb{R}^{d_K} : \rho \leq \|\boldsymbol{x}\|_2 \leq r\}$.

**Encoder:** The first layer in the encoder stack consists of a multi-head attention block inside a residual block, followed by a LayerNorm block. Using (38), the output norm-bound of the multihead attention block will satisfy:

$$\frac{\phi_{\mathrm{mha}}}{\phi_{\mathrm{mha}\downarrow}} = \left(\frac{\rho}{r}\right)^3$$

Subsequently, the output norm-bound of the residual block will satisfy:

$$\frac{\phi_{\mathrm{res}}}{\phi_{\mathrm{res}\downarrow}} = \frac{\phi_{\mathrm{in}}+\phi_{\mathrm{mha}}}{\phi_{\mathrm{in}\downarrow}+\phi_{\mathrm{mha}\downarrow}} = \frac{\phi_{\mathrm{mha}}}{\phi_{\mathrm{mha}\downarrow}} \frac{\frac{\phi_{\mathrm{in}}}{\phi_{\mathrm{mha}}}+1}{\frac{\phi_{\mathrm{in}\downarrow}}{\phi_{\mathrm{mha}\downarrow}}+1} = \left(\frac{\rho}{r}\right)^3$$

and we see from (37) that the output of the LayerNorm will satisfy:

$$\frac{\phi_{\mathrm{m}}}{\phi_{\mathrm{m}\downarrow}} = \left(\frac{\rho}{r}\right)^3$$

This is followed by a feed-forward network inside a residual block, again followed by a LayerNorm block. The analysis of this is similar to the above, excepting that, because the block inside the residual block is additive, there is no need to cube the ratio. The output of the LayerNorm in this layer will therefore satisfy:

$$\frac{\phi_1}{\phi_{1\downarrow}} = \left(\frac{\rho}{r}\right)^3$$

At total of[11] $M = 6$ of these layers occur sequentially, where for each the ratio is cubed due to the presence of the multi-head attention block. Subsequently, for the output of the encoder, we find:

$$\frac{\phi_{\mathrm{enc}}}{\phi_{\mathrm{enc}\downarrow}} = \left(\frac{\rho}{r}\right)^{3^M}$$

**Decoder:** The decoder is similar, with some important caveats. Perhaps most importantly, in the first layer the output of the second attention block (and therefore the output of the first layer in the decoder) will satisfy:

$$\frac{\phi_{\mathrm{MHA}}}{\phi_{\mathrm{MHA}\downarrow}} = \left(\frac{\rho}{r}\right)^{3+2.3^M} \leq \left(\frac{\rho}{r}\right)^{3^{M+1}}$$

This is followed by $(M - 1) = 5$ additional layers, and so it may be seen that the output of the decoder, prior to the final linear and softmax, will satisfy:

$$\frac{\phi_{\mathrm{dec}}}{\phi_{\mathrm{dec}\downarrow}} \leq \left(\frac{\rho}{r}\right)^{3^{M+1}3^{2M-2}} = \left(\frac{\rho}{r}\right)^{3^{3M-1}}$$

and, using (36):

$$\phi_{\mathrm{dec}} \leq \left(\frac{\rho}{r}\right)^{3^{3M-1}} \sqrt{d_{\mathrm{model}}}$$

Subsequently, assuming the weights in the linear output layer of the Transformer satisfy $\|\mathbf{W}\|_2 \leq \omega$ and assuming $\lambda = 1$ in the final softmax we find that the overall norm-bound for the Transformer is:

$$\phi \leq \sqrt{d_{\mathrm{model}}} \omega \left(\frac{\rho}{r}\right)^{3^{3M-1}}$$

---

[10]It may be informative to investigate the impact of the distribution $\boldsymbol{x} \sim \mathcal{X}$ on this bound in future work.

[11]We use $M$ here rather than $N$ due to the notational clash between (Vaswani et al., 2017) and our use of $N$.

# F Proof of Theorem 4 - Rademacher Complexity

We are concerned with calculating the Rademacher complexity of:

$$\mathcal{R}_N(\mathcal{F}) = \mathbb{E}_\nu \mathbb{E}_\epsilon \left[ \sup_{\Theta \in \mathbb{W}} \frac{1}{N} \sum_{i,k} \epsilon_k h\left(\mathbf{f}\left(\boldsymbol{x}_k\right)\right) \right]$$

where $h$ is $L$-Lipschitz. We have from (Maurer, 2016) that:

$$\mathcal{R}_N(\mathcal{F}) \leq \sum_i \sqrt{2} L \mathbb{E}_\nu \mathbb{E}_\epsilon \left[ \sup_{\Theta \in \mathbb{W}} \frac{1}{N} \sum_k \epsilon_k f_i\left(\boldsymbol{x}_k\right) \right]$$

Thus we reduce the dimensionality of the problem to 1-dimension. Proceeding with the standard argument:

$$\mathbb{E}_\nu \mathbb{E}_\epsilon \left[ \sup_{\Theta \in \mathbb{W}} \frac{1}{N} \sum_k \epsilon_k f_i\left(\boldsymbol{x}_k\right) \right] = \frac{1}{N} \mathbb{E}_\nu \mathbb{E}_\epsilon \left[ \sqrt{\sup_{\Theta \in \mathbb{W}} \left(\sum_k \epsilon_k f_i\left(\boldsymbol{x}_k\right)\right)^2} \right]$$

$$\leq^{\text{Jensen's-inequality}} \frac{1}{N} \mathbb{E}_\nu \left[ \sqrt{\mathbb{E}_\epsilon \sup_{\Theta \in \mathbb{W}} \left(\sum_k \epsilon_k f_i\left(\boldsymbol{x}_k\right)\right)^2} \right]$$

$$= \frac{1}{N} \mathbb{E}_\nu \left[ \sqrt{\mathbb{E}_\epsilon \sup_{\Theta \in \mathbb{W}} \left(\sum_{k,l} \epsilon_k \epsilon_l f_i\left(\boldsymbol{x}_k\right) f_i\left(\boldsymbol{x}_l\right)\right)} \right]$$

$$=^{\mathbb{E}_\epsilon \epsilon_k \epsilon_l = \delta_{k,l}} \frac{1}{N} \mathbb{E}_\nu \left[ \sqrt{\sum_k \sup_{\Theta \in \mathbb{W}} f_i^2\left(\boldsymbol{x}_k\right)} \right]$$

$$= \frac{1}{N} \mathbb{E}_\nu \left[ \sqrt{\sum_k \sup_{\Theta \in \mathbb{W}} \langle \boldsymbol{\Psi}_{:i}\left(\Theta\right), \boldsymbol{\phi}\left(\boldsymbol{x}_k\right)]_{\mathbf{g}}^2} \right]$$

$$= \frac{1}{N} \mathbb{E}_\nu \left[ \sqrt{\sum_k \sup_{\Theta \in \mathbb{W}} \left(\frac{\langle \boldsymbol{\Psi}_{:i}(\Theta), \boldsymbol{\phi}(\boldsymbol{x}_k)]_{\mathbf{g}}}{\|\boldsymbol{\phi}(\boldsymbol{x}_k)\|_2}\right)^2 \|\boldsymbol{\phi}\left(\boldsymbol{x}_k\right)\|_2^2} \right]$$

and so:

$$\mathcal{R}_N(\mathcal{F}) \leq \frac{\sqrt{2}L}{N} \mathbb{E}_\nu \left[ \sum_i \sqrt{\sum_k \sup_{\Theta \in \mathbb{W}} \left(\frac{\langle \boldsymbol{\Psi}_{:i}(\Theta), \boldsymbol{\phi}(\boldsymbol{x}_k)]_{\mathbf{g}}}{\|\boldsymbol{\phi}(\boldsymbol{x}_k)\|_2}\right)^2 \|\boldsymbol{\phi}\left(\boldsymbol{x}_k\right)\|_2^2} \right]$$

$$\leq^{\text{1-norm-2-norm-inequality}} \frac{\sqrt{2m}L}{N} \mathbb{E}_\nu \left[ \sqrt{\sum_i \sum_k \sup_{\Theta \in \mathbb{W}} \left(\frac{\langle \boldsymbol{\Psi}_{:i}(\Theta), \boldsymbol{\phi}(\boldsymbol{x}_k)]_{\mathbf{g}}}{\|\boldsymbol{\phi}(\boldsymbol{x}_k)\|_2}\right)^2 \|\boldsymbol{\phi}\left(\boldsymbol{x}_k\right)\|_2^2} \right]$$

$$= \frac{\sqrt{2m}L}{N} \mathbb{E}_\nu \left[ \sqrt{\sum_k \sup_{\Theta \in \mathbb{W}} \frac{\left\|\langle \boldsymbol{\Psi}_{:i}(\Theta), \boldsymbol{\phi}(\boldsymbol{x}_k)]_{\mathbf{g}}\right\|_2^2}{\|\boldsymbol{\phi}(\boldsymbol{x}_k)\|_2^2} \|\boldsymbol{\phi}\left(\boldsymbol{x}_k\right)\|_2^2} \right]$$

$$=^{\text{bilinear-continuity}} \frac{\sqrt{2m}L}{N} \mathbb{E}_\nu \left[ \sqrt{\sum_k \sup_{\Theta \in \mathbb{W}} C_{\Theta,\eta}^2 \|\boldsymbol{\phi}\left(\boldsymbol{x}_k\right)\|_2^2} \right]$$

$$=^{\text{norm-bounding}} \frac{\sqrt{2m}L}{N} \mathbb{E}_\nu \left[ \sqrt{\sum_k C_{\mathbb{W},\eta}^2 \phi_\eta^2} \right] =^{\text{cleanup}} \frac{\sqrt{2m}L}{\sqrt{N}} C_{\mathbb{W},\eta} \phi_\eta$$

and the final result follows in the limit $\eta \to 0^+$, recalling $\lim_{\eta \to 0^+} C_{\mathbb{W},\eta} = 1$, $\lim_{\eta \to 0^+} \phi_\eta = \phi$:

$$\mathcal{R}_N(\mathcal{F}) \leq \frac{\sqrt{2m}L}{\sqrt{N}} \phi$$

**NB**: in the special case $m = 1$, $h = \text{id}$, we can skip the first step which contributed the factor $\sqrt{2}L$ and the 1-norm-2-norm-inequality.

