# OpenReview forum: "Reproducing Kernel Banach Space Models for Neural Networks with Application to Rademacher Complexity Analysis"
_NeurIPS.cc/2025/Conference — NeurIPS 2025 poster_

### Official Review · Reviewer_c4ND · 2025-06-19

**Clarity:** 1
**Significance:** 3
**Originality:** 3
**Rating:** 4
**Confidence:** 3

**Summary:**

This paper introduces a theoretical framework which characterizes general neural network architectures using a reproducing kernel Banach space (RKBS) construction. It uses this construction to find bounds for the Rademacher complexity of various architectures which are sharper for deeper networks.

**Questions:**

See weaknesses. I am not confident in the correctness of the arguments for bounded activation functions and would like to see some clarification on this point.

**Ethical Concerns:**

["NO or VERY MINOR ethics concerns only"]

**Final Justification:**

All of my comments were addressed. The point about the transformer Rademacher complexity was my biggest concern, and the authors have resolved it providing some intuition for why it should work and what the edge cases look like. The main outstanding weakness is the readability of the paper.

**Limitations:**

yes

**Quality:**

3

**Strengths And Weaknesses:**

### Strengths:
- The techniques in the paper allow one to improve existing Rademacher complexity bounds by a constant factor at each layer, which leads to an exponential improvement as a function of the number of layers.
- The RKBS formalism is interesting and different from existing methods used to bound the expressivity of neural networks. It seems general enough to be useful to prove other theoretical results.
- The paper considers a family of network architectures which is quite general, including fully connected networks, resnets, transformers, etc. I appreciated that there was some discussion on the special case of fully connected networks to make the results more concrete.
- There is some intuition behind proofs of the theorems in the main text, which makes the ideas clearer.

### Weaknesses:
- The paper has a lot of cumbersome, non-standard notation which is hard to parse. This makes it difficult to understand the details of the formulas and check that they are correct.
- For some architectures, the constants $\phi_{\eta}$ and $\psi_{\eta}$ determining the Rademacher complexity are complicated and hard to interpret.
- For the depth-independence of the transformer, the argument seems to be that one gets depth-independence from having layer normalization late in the network layers, and using that layer-normalization is a bounded function? Why would this argument not hold for any architecture satisfying assumptions 1-4 with any bounded function at the end? This argument seems like it is proving too much; by this reasoning, one could have an architecture with high Rademacher complexity with layer normalization in the end to reduce it to a constant.

Minor points:
- The links to sections and references are broken.
- On line 258, $\partial \mathbb{X} \in$ should be $\partial \mathbb{X} =$.

---

> ### Author Rebuttal · Authors · 2025-07-30
>
> **Response to reviewer c4ND:**
>
> **Q1:** The paper has a lot of cumbersome, non-standard notation which is hard to parse. This makes it difficult to understand the details of the formulas and check that they are correct.
>
> **A:** In this paper we have tried to cover the widest range of NN topologies possible to make our results as widely applicable as possible. For example:
>
> - we consider non-layerwise topologies, so a simply "layer j" superscript does not suffice when specifying which weight, activation etc we are refering to. This motivates the $[j]$ (with respect to node $j$) and $[\tilde{j}:j]$ (with respect the directed edge from node $\tilde{j}$ to node $j$) indices for clarity.
>
> - we consider multiple "types" of node - including attention, layernorm etc. We could consider these separately, but this would result in a lot of redundant (repetetive) results with very low informational content after the first, which we would like to avoid for readability. This motivates the operator notation described in figure 4.
>
> - the use of $H$ for width is not original here, and the superscript versions $H^{[j]}$, $H^{[\tilde{j}:j]}$ is necessary for the general case as we are not primarily interested in the infinitely-wide case (in which case "the width" is typically just taken as a constant for all layers).
>
> - the Hermite-related and multi-index notations (as well as other mathematical notations) are all fairly standard, though perhaps not so widely used in the NN community.
>
> To help the reader we have summarised relevant notations in section 1.1. Note that we did consider dropping equation (5) from the body of the paper to the appendix, but decided that on balance it was important to retain it so that the interested reader could inspect the feature map underlying the bilinear (RKBS) representation if they so desired, even though it is not strictly necessary for the following Rademacher complexity bounds. Again, this is a question of balance.
>
>
> **Q2:** For some architectures, the constants and determining the Rademacher complexity are complicated and hard to interpret.
>
> **A:** We will endeavour to improve the writing and description around these. If possible could you please elaborate on which constants you have difficulty interpretting so that we can improve these cases in particular?
>
>
> **Q3:** For the depth-independence of the transformer, the argument seems to be that one gets depth independence from having layer normalization late in the network layers, and using that layer normalization is a bounded function? Why would this argument not hold for any architecture satisfying assumptions 1-4 with any bounded function at the end? This argument seems like it is proving too much; by this reasoning, one could have an architecture with high Rademacher complexity with layer normalization in the end to reduce it to a constant.
>
> **A:** The reason we have not reached this conclusion is that, for this final result to hold, it is necessary that the training data lies *on the boundary of the unit sphere* - that is, $|| {\bf x} ||_2 = 1$, as stated in our caveat on line 257. The reason we require this is that, in this special case we can calculate
>
> $|| {\bf{\phi}} ({\bf x}) ||_2$
>
> exactly (see appendix D.3), which is subsequently required to prove the continuity bound in Appendix D.4, in particular the equation immediately after line 545, where the inequality relies on $|| {\bf{\phi}} ({\bf x}) ||_2$ being strictly non-zero. While we can imagine scenarios where the data does lie on the boundary of the unit sphere, in general this seems unlikely. Thus we include this result as an interesting special case, but take care to note the special circumstances involved, and subsequently not claim too much from it.
>
> As an aside we note that we can "expand" the definition here very slightly. If we require instead that $0 < \epsilon \leq || {\bf x} ||_2 \leq 1$ then we find that:
>
> $|| {\bf{\phi}} ({\bf x}) ||_2 > 0$
>
> which is not exact, but is non-zero. Because this is non-zero we obtain an inequality on the continuity constant similar to (though not as tight as) that on line 545 in Appendix D.4. However the choice of $\epsilon$ here is somewhat arbitrarily, and moreover the additional scaling factor on the Rademacher complexity bound grows (loosely speaking) as $1/\epsilon^D$, which appears to counteract the apparent "too muchness" of the result. Thus, to reiterate what we have written above, we consider this final corollary an interesting special case with rather limited applicability that is nevertheless worth noting.

---

> > ### Comment · Reviewer_c4ND · 2025-08-02
> >
> > Thank you for your detailed response.
> >
> > > To help the reader we have summarised relevant notations in section 1.1. Note that we did consider dropping equation (5) from the body of the paper to the appendix, but decided that on balance it was important to retain it so that the interested reader could inspect the feature map underlying the bilinear (RKBS) representation if they so desired, even though it is not strictly necessary for the following Rademacher complexity bounds. Again, this is a question of balance.
> >
> > Thanks for explaining your reasoning behind these choices. I acknowledge that there is a tradeoff between generality and simplicity of notation, and the generality of architectures is a strength of the paper.
> >
> > > We will endeavour to improve the writing and description around these. If possible could you please elaborate on which constants you have difficulty interpretting so that we can improve these cases in particular?
> >
> > I was thinking, for instance, about the expressions for $\phi_{\eta}$ in Appendix D.1. I think in general, it would help with clarity to have a sketch or short explanation of how one translates from these expressions to the simpler Rademacher complexity results stated in the main text.
> >
> > > The reason we have not reached this conclusion is that, for this final result to hold, it is necessary that the training data lies _on the boundary of the unit sphere_
> >
> > > As an aside we note that we can "expand" the definition here very slightly. If we require instead that $0 < \epsilon \leq \|\mathbf{x}\|_2 \leq 1$ then we find that: $\|\phi(\mathbf{x})\| > 0$ which is not exact, but is non-zero.
> >
> > This intuition is helpful, although I still find it surprising; I would not expect the points being restricted to a unit sphere to impact the Rademacher complexity by that much.
> >
> > The concerns I had in my original review have been adequately addressed, so I will increase my score from a 3 to a 4.

---

### Official Review · Reviewer_t2Jm · 2025-07-02

**Clarity:** 2
**Significance:** 3
**Originality:** 2
**Rating:** 4
**Confidence:** 3

**Summary:**

This paper formulates extensive classes of neural networks as elements of a Reproducing Kernel Banach Space (RKBS), using Hermite transforms. The main result is that neural networks can be written in the corresponding bilinear form, and coefficients are provided explicitly (coefficients can be computed by induction on nodes of the network).
This formulation is then used to bound the Rademacher complexity of neural network function classes, in particular, the authors provide conditions under which this Rademacher complexity does not grow exponentially with depth.

**Questions:**

Main questions above. Minor questions/comments below.

p1 l26-27. There seem to be missing words.

p1 l29. What do you mean by "spectral-norm bound weight matrices"? Do you mean "spectral-norm bounded weight matrices"?

p2 l63. At this point, neither $C_\Theta$ nor $C_W$ have been defined, so perhaps deleting $C_W$ at this point of the presentation would be clearer.

p2 Add definition of $\oplus$ in this context.

p6 caption of Fig 4. "neither Lipschitz or bounded" -> "neither Lipschitz nor bounded"

p7 The statement of Theorem 1 is quite obscure with respect to the parameter $\eta$. This parameter has not been appropriately discussed at this point. At this point in the paper, it is impossible for the reader to appreciate what the limit statements as $\eta\to 0^+$ mean/refer to. Some of the quantities inside these statements do not have explicit dependency in $\eta$ either (e.g. $\|\phi^{[j]}(x)\|_2$ and $\|\phi(x)\|_2$).

p7 l165 "quarantined" do you mean "guaranteed"?

p8 Eq(8). Where is $\phi$ defined in Figure 4? Is it supposed to be $\phi^{[E]}$?


p8 l223. "exponentially" do you mean "exponential"?

p8 l220-222. From what I understand, writing these requires that the functions are 1-Lipschitz. If so, this should be explcitly stated before writing the equation. Otherwise, where did the factors $L^{[j-1:j]}$ go? Also, what are the assumptions required for Golowich
221 et al., 2018, Theorem 7? In your result, if the Lipschitz constants become small, these would contradict that result.

p8 Eq (10). Should the second quantifier $\forall \tilde j\in A^{[j]}$ be deleted?


p9 l257-258 some grammatical issues.

p17 l447 "tivially" -> "trivially"

**Ethical Concerns:**

["NO or VERY MINOR ethics concerns only"]

**Final Justification:**

I maintain my score. The paper provides some good contributions to Rademacher analysis for neural networks, but the presentation makes it hard to digest

**Limitations:**

Limitations could be discussed in more depth -- see comments above. N/A for societal impact.

**Quality:**

3

**Strengths And Weaknesses:**

Strengths:

The proposed RKBS representation is extremely general and covers most (basically all) neural networks architectures of interest. The representation is also exact (so long as the coefficients in the Hermite decomposition can be nicely computed).

This representation also naturally leads to bounds on the Rademacher complexity of the considered function classes---given by the neural network architecture, and appropriate coefficient/spectral bounds. In some sense, these are cleaner than previous works as in they do not explicitly contain exponential factors.

Weaknesses:

The paper would gain by giving more details on the motivation for the current approach. Viewing neural networks (NNs) through the lens of RKBS (or RKHS) is quite standard in the literature, but I am less familiar with analyzing the Rademacher complexity of neural networks for understanding their performance in practice. I think the paper would gain by giving more details as to why using a uniform convergence approach is useful/reasonable for NNs (especially for non-experts), since this is currently the main motivation given in the introduction. Usually, training is performed in the over-parametrized regime and achieves 0 training error -- so in practice I imagine that we never have uniform convergence, since we do not even have convergence for the trained model.



Using the language from the paper, why should one expect the bounds to be non-vacuous ($\phi<<\sqrt N$) in standard regimes of training? Along the same lines, the paper provides some conditions under which dependency of the Rademacher in depth is not exponential, but these are very technical and hard to interpret. As a non-expert these seem quite restrictive---are these actually realistic? Further, the paper claims to remove explicit exponential in depth bounds from previous papers (Golowich et al. 2018), but the main term $\phi$ grows exponentially with depth in an implicit manner and is not related in an obvious way (at least to me) with the terms from their bound (the $w$'s) whenever the neural network has width>1. Is there any way to compare the bounds formally?



Last, the presentation is very heavy in notations and technicalities, which makes it quite hard to read. Most of these technicalities, however, seem necessary to properly define the RKBS representation and other results. Still, there seems to be a few imprecisions/typos within the text, I listed those that I saw below but surely missed others.
A careful pass to fix these minor issues is necessary.

---

> ### Author Rebuttal · Authors · 2025-07-30
>
> **Response to reviewer t2Jm:**
>
> **Q1:** The paper would gain by giving more details on the motivation for the current approach. Viewing neural networks (NNs) through the lens of RKBS (or RKHS) is quite standard in the literature, but I am less familiar with analyzing the Rademacher complexity of neural networks for understanding their performance in practice. I think the paper would gain by giving more details as to why using a uniform convergence approach is useful/reasonable for NNs (especially for non-experts), since this is currently the main motivation given in the introduction. Usually, training is performed in the over-parametrized regime and achieves 0 training error -- so in practice I imagine that we never have uniform convergence, since we do not even have convergence for the trained model.
>
> **A:** The gap between training error and testing (actual) error remains is an area of active research. The classical result states that, without taking the training algorithm into account and only considering the set $\mathcal{F}$ of possible neural networks under certain constraints (in our case the assumption of norm-bounded weights and biases), the gap between training error and (expected) testing error is bounded, roughly speaking and with-high-probability $\geq 1-\delta$, as:
>
> $\sup_{f \in \mathcal{F}} | \frac{1}{N} \sum_i l ( f(x_i), y_i ) - \mathbb{E}_{x,y \sim \mathcal{D}} [ l ( f(x), y ) ] | \sim \mathcal{O} ( \sqrt{\frac{{\rm complexity} + \log (\frac{1}{\delta})}{N}} )$
>
> where many different measures of complexity have been studied (out of which we would argue that Rademacher complexity $\mathcal{R}_N (\mathcal{F})$ is one of the more rigorous and well-motivated). This is the *architectural bias* (note also that the term uniform convergence indicates that the gap converges with increasing training set size $N$ (assuming the complexity is well-behaved), which is distinct from convergence in the sense of training). In addition to this, more recent work has studied the effect of the training algorithm on this gap, which is known *training bias* - for example benign-overfitting can be studied by considering the gradient-flow model of training, in particular by analysing the spectral structure of the corresponding NTK.
>
> Considering your example where training terminates prematurely with 0 error, we would expect this to have a strong effect on the training bias, as any eg. benign overfitting effects will be halted before they can have a significant impact on the result. By contrast, the architectural bias is *independent* of the training algorithm (unless training results in a significant increase in weights, which is unlikely if a training error of 0 is achieved quickly), and so should not be affected. Hence in your example the bound becomes (ie. for 0 training loss, $\frac{1}{N} \sum_i l ( f(x_i), y_i ) = 0$):
>
> $\sup_{f \in \mathcal{F}} | \mathbb{E}_{x,y \sim \mathcal{D}} [ l ( f(x), y ) ] | \sim \mathcal{O} ( \sqrt{\frac{\mathcal{R}_N (\mathcal{F}) + \log (\frac{1}{\delta})}{N}} )$
>
> with high probability. Thus while the *training bias* would be adversally effected in your (early terminated) example, the *architectural bias* bound from Rademacher complexity will remain unchanged.
>
> I hope this helps to answers your question, but I would be happy to discuss this further if desired.
>
>
> **Q2:** Using the language from the paper, why should one expect the bounds to be non-vacuous ($\phi \ll \sqrt{N}$) in standard regimes of training? Along the same lines, the paper provides some conditions under which dependency of the Rademacher in depth is not exponential, but these are very technical and hard to interpret. As a non-expert these seem quite restrictive---are these actually realistic? Further, the paper claims to remove explicit exponential in depth bounds from previous papers (Golowich et al. 2018), but the main term grows exponentially with depth in an implicit manner and is not related in an obvious way (at least to me) with the terms from their bound (the $\omega$s) whenever the neural network has width>1. Is there any way to compare the bounds formally?
>
> **A:** Whether the bounds are non-vacuous is a difficult question in the most general case, as this requires $\phi \sim \mathcal{O} (\sqrt{N})$, noting that this same problem will be present in all weight-norm based analyses of uniform convergence (eg Golowich et al, Neyshabur et al etc). We discuss this issue in section 4, in particular lines 210-225. To summarise, if we consider the case of LeCun or He initialization and take seriously the over-parameterised limit analysis then it is reasonable to conclude that the weight matrices will remain close to their initialization, and hence that their norm-bound will remain close to $|| {\bf W}^{[\tilde{j}:j]} ||_2 \leq \omega^{[\tilde{j}:j]} \approx 1$. When (or if) these assumptions hold more generally remains an area of active debate.
>
> With regard to the depth-exponentiality of our bound, as noted on lines 222-225 our bound does exhibit exponential depth dependence. The point we are making here is that this is only a problem if the norm-bounds $\omega^{[\tilde{j}:j]}$ are greater than 1. Put simply, if we assume for simplicity that $\omega^{[\tilde{j}:j]} = \omega$ (the bound is the same for all layers) then $\mathcal{R}_N \sim \mathcal{O} (\omega^D)$, so if $\omega = 1$ then this collapses to $\mathcal{R}_N \sim \mathcal{O} (1)$ - that is, no explicit depth independence. Much of section 4 is dedicated to generalizing this intuition to a variety of common network topologies and giving conditions under which it is true.
>
> Finally, with regard to width, while we cannot entirely rule out the possibility of it playing a role in our bounds through the weight norm-bound, we would observe that LeCun and He initialization both result in weight matrices whose norms are decoupled from network width, and it is difficult to see how such a dependency could occur during usual training.
>
>
> **Q3:** [too much notation]
>
> **A:** We have spent a lot of time trying to minimize the notational load in this paper. As you note, we do need some level of notation to unambiguously present our theory. We have also done our best to choose notations we feel will help to reader to navigate the mathematics. For example the superscript $[j]$ to indicate "relating to node $j$, or $[\tilde{j}:j]$ indicating "relating to the link between nodes $\tilde{j}$ and $j$." And the operator notation (as described in figure 4) has been used to avoid the needless repetition that would have otherwise been needed to describe each type of node independently - see figure 4 for the "key". To address specific concerns:
>
> - p2 Add definition of $\oplus$ in this context: we define this as a matrix concatenation operator on page 2, line 76.
>
> - p7, parameter $\eta$ and associated notation: I will add a short, preemptive discussion of $\eta$ after the theorem. Essentially this parameter is a "free parameter" in this theorem. It makes no difference to the theorem (the result holds true for all $\eta > 0$), but it becomes necessary later when bounding the Rademacher complexity as the bounds take a much simpler form in the limit $\eta \to 0_+$ (the limits on lines 158-159 are effectively a precursor to the later usage). With regard to us writing $\phi ({\bf x})$ etc instead of $\phi_\eta ({\bf x})$ here, we considered including the subscript but decided that the loss of readability from the additional visual clutter was too much (we can reinstate these subscripts if you feel differently).
>
>
> **Q4:** p8, lines 220-222: From what I understand, writing these requires that the functions are 1-Lipschitz. If so, this should be explcitly stated before writing the equation. Otherwise, where did the factors go? Also, what are the assumptions required for Golowich 221 et al., 2018, Theorem 7? In your result, if the Lipschitz constants become small, these would contradict that result.
>
> **A:** in this context we are discussing the $1$-Lipschitz case, with a simple layerwise network as per Golowich, so I will note this in the paper. With regard the Lipschitz constants it is worth noting that we can always absorb these constants. Precisely, consider (dropping superscripts for simplicity):
>
> ${\bf W}^T \tau ({\bf x})$
>
> where $|| {\bf W} || \leq \omega$ and $\tau$ is $L$-Lipschitz. We can always rewrite this:
>
> $\hat{\bf W}^T \hat{\tau} ({\bf x})$
>
> where $\hat{\bf W} = L {\bf W}$, $|| \hat{\bf W} || \leq \hat{\omega} = L\omega$ and $\hat{\tau} ({\bf x}) = \frac{1}{L} \tau ({\bf x})$ is $1$-Lipschitz. Thus the $1$-Lipschitz assumption is non-restrictive.
>
>
> **Minor points:**
>
> - p2, Line 63: we will delete $C_{W}$ as suggested.
>
> - p7, line 165: we do actually mean quarantined here. It is possible to incorporate the Hermite numbers into the feature maps and not the metric (this is largely a stylistic decision), but this complicates the analysis, so we "quarantine" (wall off) these in the metric to make our lives a little easier.
>
> - p8, eq (8): $\phi$ is defined in figure 4, at the end of the last line of equation 5.
>
> - p8, eq (10): we will remove the second qualifier.

---

> > ### Comment · Reviewer_t2Jm · 2025-08-02
> >
> > Thank you very much for the responses, these are very useful.
> >
> > As a clarification for Q1. I am familiar with Rademacher bounds, their applicability is not really the concern.
> > The question I was asking is the following (and this would be in common with other works studying Rademacher bounds for deep learning so perhaps there is a classical answer):
> > In practical scenarios, one can imagine that the optimal learning error is non-zero (e.g. there is a constant amount of noise) say $\alpha>0$. In practice in neural networks, we achieve 0 error in training which means that there is some function that achieves 0 empirical error. So the uniform deviation is at least $\alpha>0$ -- which may not be much better than the trivial bound 1. In such a scenario, Rademacher bounds would not tell us much more than the trivial bound simply because the function class has not uniformly-converged (for the given nb of samples) given that neural networks are so expressive. From that perspective, why is studying uniform convergence practically relevant for deep learning?

---

> ### Author Response · Authors · 2025-08-04
>
> Thanks for your response, I think I now have a better understanding of what you are saying here.
>
> In your comment you say "...the function class has not uniformly-converged". However, the function class $\mathcal{F}$ does not change at all. It is just the set of all possible functions the network can represent provided weights satisfy the norm-bounds. To emphasise, as this is key: the function $f$ performed by the network corresponding to the particular weights and biases evolves - weights change during training - but the *function class* $\mathcal{F}$ does not.
>
> Now:
>
> - the Rademacher complexity function depends only on the function class $\mathcal{F}$ (for simplicity assuming the scalar-output case $m=1$, $h = id$), so:
>
> $R_N (\mathcal{F}) = E \left[ {\rm sup}_{f \in \mathcal{F}} \frac{1}{N} \sum_i \epsilon_i f \left( {\bf x}_i \right) \right]$
>
> - our bound on the Rademacher complexity depends only on the function class $\mathcal{F})$, ie Theorem 4 in the paper (again for $m=1$, $h = id$):
>
> $R_N (\mathcal{F}) \leq \frac{\phi}{\sqrt{N}}$
>
> - the bound associated with this also depends only on the function class $\mathcal{F}$. In your scenario where the training risk is $0$ and the actual risk is $\alpha$ a representative bound is, with high probability $\geq 1-\delta$ (this variant is from [1, Theorem 8], substituting training and actual risk to match, however there are numerous variations in other sources following the same essential template):
>
> $\alpha \leq R_N (\mathcal{F}) + \sqrt{\frac{8{\rm ln}(\frac{2}{\delta})}{N}} + \left(\mbox{training risk}=0\right)$
>
> - subsequently, dependent only on the function class $\mathcal{F}$, we find the bound (in your scenario, $m=1$, $h = id$), with high probability $\geq 1-\delta$:
>
> $\alpha \leq \frac{\phi}{\sqrt{N}} + \sqrt{\frac{8{\rm ln}(\frac{2}{\delta})}{N}} + \left(\mbox{training risk}=0\right)$
>
> The key point to none of these bounds are in any way connected to the training of the network. Indeed, to pick an extreme case, we could randomly sample weights until we found a network with $0$ training error, and the bound would still hold (with $\phi \approx 1$ if we are considering a simple layerwise network with LeCun or He draws). Uniform convergence in the above example is about how the bound (on $\alpha$) changes (converges) as the training set size $N \to \infty$.
>
> Whether the training terminates early (prematurely) is immaterial, potentially excepting that in this case the weights are more likely to stay close to their initialization, so for example for a simple layerwise network, as discussed in section 4, we may be able to assert even tighter norm-bounds on the weight matrices, subsequently reducing $\phi$ and making the bound on $\alpha$ tighter than would be the case if training had proceeded to find a ``better'' solution.
>
> More generally, despite recent results giving potentially vacuous bounds, uniform convergence using Rademacher complexity remains relevant for deep networks as it offers theoretical insight into function class complexity. For example, if we consider the infinite-width case with lazy training, we can still apply (local) Rademacher analysis of the neighbourhood of the initialization to bound the generalization error; and even outside of the wide-network case where we don't stay in the neighbourhood of weight initialization (so, in a sense, the neighbourhood ``evolves'') the set of functions that are reachable in this analysis is a subset of the larger neighbourhood $\mathcal{F}$ we consider - so by the properties of Rademacher complexity our results place an upper bound on the more refined analysis.
>
> [1] Bartlett, P. L. and Mendelson, S. Rademacher and gaussian complexities: Risk bounds and structural results. Journal of Machine Learning Research, 3:463–482, 2002.

---

> > ### Comment · Reviewer_t2Jm · 2025-08-04
> >
> > Thank you for your comments. I think this discussion is deviating a little bit from the paper (it is not really a criticism of your work, but mostly a general question which does not aim to downplay your specific contributions at all).
> >
> > >"the function class $\mathcal{F}$ does not change at all"
> >
> > Yes, I am aware of this, and agree with your discussion about Rademacher analysis. I am not quite talking about training. By "has not uniformly converged" I simply mean $\sup_{f\in\mathcal F} |\hat{\mathcal L}_N(f) - \mathcal L(f)| = \Theta(1) $ (in which case Rademacher bounds would also be vacuous, e.g. $\phi \gg \sqrt N$). I somewhat referred to training as a thought argument to try lower bound this quantity.
> >
> > The question I had was: it seems plausible to me that in practice this holds (if there is a function which perfectly fits the data, one has $\sup_{f\in\mathcal F} |\hat{\mathcal L}_N(f) - \mathcal L(f)| >\alpha$ where $\alpha$ is the minimum noise. Next, if $\alpha=\Theta(1)$ then we get the above). First, do you think this is the case in practice? If so, why study uniform convergence for deep learning? (This basically the classical concern that VC and Rademacher theory do not really explain why NN actually work well in overparametrized regimes)

---

> > > ### Author Response · Authors · 2025-08-04
> > >
> > > I would expect that the question of whether $\phi \sim \Theta (\sqrt{N})$ would very much depend on the network structure and loss function. Certainly in the layer-wise, infinite-width regime of, for example, NTK theory, my understanding is that weights stay very close to their initialization, so assuming LeCun or He initialization for this limit (using [2], Lemma 1, and defining $\epsilon$ to be the change in weight-norm due to training):
> > >
> > > $\omega^{[j-1:j]} = 1 + \mathcal{O} (\frac{\ln H}{H}) + \epsilon \to_{H \to \infty} 1 + \epsilon$
> > >
> > > So I would expect that $\phi \sim \mathcal{O} ((1+\epsilon)^D)$ (assuming 1-Lipschitz activations for simplicity), where $0 < \epsilon \ll 1$ (weights stay close to initialization). On the flipside, for a very narrow network you could well find that $\phi \sim \Theta (\sqrt{N})$. More generally I would expect something between these extremes.
> > >
> > > One benefit of our approach is that it allows us to give conditions under which we can have non-vaccuous bounds. Noting that weight initialization like LeCan and He give weight matrices $||{\bf W}^{[j-1:j]}|| = 1 + \mathcal{O} (\frac{\ln H}{H})$, the question becomes whether $||{\bf W}^{[j-1:j]}||$ grows as $\Theta (\sqrt{N})$ or not - one possibility gives strong bounds, the other vaccuous. Alternatively, we could view this as giving a strong motivation for regularization (directly penalizing the matrix norm, or early stopping) to achieve provably better generalization.
> > >
> > > So to answer your question: no, I do not think that $\phi \sim \Theta (\sqrt{N})$ in general. I would expect that this may happen with some networks, but I would be very surprised if it was true universally or even in the majority of cases, particularly for large networks. And I would expect that early termination - whether intentional (early stopping) or due to stalling when the training error becomes $0$ - would *limit* the potential (worst case) growth of network weights, which should (at least to a first-order approximation) make any growth in $\omega$ (and subsequently $\phi$) proportional to training time rather than training set size. And I think that it is important to continue to study these questions, particular for more complex network types which have been less studied, which seem (at least to me) to lie at the core of our understanding of neural networks.
> > >
> > > [2] Laurent, B. and Massart, P. Adaptive estimation of a quadratic functional by model selection. The Annals of Statistics, 28(5):1302 – 1338, 2000

---

### Official Review · Reviewer_szLf · 2025-07-03

**Clarity:** 3
**Significance:** 4
**Originality:** 3
**Rating:** 5
**Confidence:** 2

**Summary:**

The authors apply RKBS to represent neural network architectures, including ResNet and Transformers networks. They use Hermite polynomials to expand activation functions and construct feature maps. They show that the neural networks are representaed using dual representation that separates the parameter feature and the input feature. Using this representation, they derive a generalization bound whose dependency on the network width is milder than existing typical bounds.

**Questions:**

Are there any ideas of deriving a generalization bound whose dependency on the depth of the network is milder than the exponential dependency?

**Ethical Concerns:**

["NO or VERY MINOR ethics concerns only"]

**Final Justification:**

I think this paper provides an important theoretical contribution.

**Limitations:**

In assumption 1, they assume the data is normalized to a unit ball. However, this assumption may not true for test data. Is it difficult to generalize the results to unbounded data space?

Minor comment:
The notations in Corollary 5 are confusing. Could you check it and clarify the statement?

**Paper Formatting Concerns:**

N / A

**Quality:**

3

**Strengths And Weaknesses:**

The application of RKBS to derive generalization bounds of deep neural networks is very interesting topic. They construct a feature maps for representing wide range of neural network architectures, and use the reproducing property to derive the generalization bound. The bound depends on the operator norm of weights, whose dependency on the width is milder than typical bounds with the Frobenius norm.

Although this paper provides a promissing direction of theoretical analysis of deep neural networks, understanding the overview of this work may be difficult for readers from various backgrounds. Providing a figure that shows the workflow of deriving the results could be helpful for readers (e.g. visualize the connection among expanding activation function, constructing feature maps, and deriving a generalization bound).

---

> ### Author Rebuttal · Authors · 2025-07-30
>
> **Response to reviewer szLf:**
>
> **Q1:** Are there any ideas of deriving a generalization bound whose dependency on the depth of the network is milder than the exponential dependency?
>
> **A:** Unfortunately no. Moreover as we discuss in section 4 (in particular lines 220-221), this may not be possible in many cases using Rademacher complexity, as (Golowich et al, 2018, Theorem 7) gives an effective lower bound on Rademacher complexity that is depth-exponential. However as we note in the paper this is only problematic if the base exceeds 1 - otherwise the result is (in effect) depth-independent.  Nor does it mean that such a result is ruled out for particular cases, only that it may not be possible for simple feed-forward networks with norm-bounded weights and inputs in the unit ball. Indeed, as a counter-example we give a depth-independent bound for the transformer network on lines 252-258; however to do so we are forced to restrict ourselves to the case where data lies *on the boundary of the unit ball* ${\bf x} \in \partial \mathbb{X} = $ { ${\bf x} : || {\bf x} ||_2 = 1$ }, which is a strong assumption (stronger than made for all other results in our paper) that may not be met in practical scenarios.
>
>
> **Q2:** In assumption 1, they assume the data is normalized to a unit ball. However, this assumption may not true for test data. Is it difficult to generalize the results to unbounded data space?
>
> **A:** The assumption that data lies in a unit ball is fairly standard in the literature. It's not difficult to extend this to the case $|| {\bf x} ||_2 \leq r$. If we consider the derivation of the norm-bound $\phi$ in appendix D, the only change is on line 492, specifically: in the base case $j=0$, we have (note that we have now fixed the minor typo here in the paper):
>
> $|| {\bf \phi}^{[0]} ({\bf x}) ||_2^2 \leq \phi^{[0]2} = r^2$
>
> rather than $\phi^{[0]2} = 1$. The net result of this is that $\phi \to r \phi$ in (5), figure 4 (that is, a linear rescaling of the norm-bound, and subsequently a linear rescaling of the Rademacher complexity bound).
>
> To treat the unbounded case would require additional assumptions on the data distribution. If you could assert (or demonstrate/prove) that $|| {\bf x} ||_2 \leq r$ with high probability then we may reframe our results as high-probability results. For example if we assume a Gaussian (or sub-Gaussian) distribution on ${\bf x}$ then standard concentration inequalities (eg the Chernoff bound) tell us that there exists $r \sim \mathcal{O} (1 / \log \epsilon)$ such that $|| {\bf x} || \leq r$ with probability $\geq \epsilon$, and our result follow with probability $\geq \epsilon$. Once again we note that the only impact on the Rademacher complexity bound is a linear rescaling and an additional (with-high-probability) caveat in the theorem.
>
> We will add a formalised version of this answer to the supplementary material.
>
>
> **Q3:** The notations in Corollary 5 are confusing. Could you check it and clarify the statement?
>
> **A:** Sorry about that - we have now removed the confusing second $\forall \tilde{j} \in \mathbb{A}^{[j]}$ condition which we had mistakenly included in the original.

---

> > ### Comment · Reviewer_szLf · 2025-08-05
> >
> > Thank you for the response. I think this paper provides an important theoretical contribution. I would like to keep my score unchanged.

---

### Official Review · Reviewer_oQnH · 2025-07-03

**Clarity:** 3
**Significance:** 2
**Originality:** 3
**Rating:** 4
**Confidence:** 4

**Summary:**

For the problem that the theoretical results of Rademacher complexity in capacity-based generalization analysis may be vacuous,
a novel theoretical framework is proposed, which uses the reproducing kernel Banach space (RKBS) based on the Hermite transform to accurately model feedforward neural networks, including complex structures such as ResNet and Transformer. This model does not require approximate assumptions, as long as the weights are bounded and the activation functions are finite energy. Based on this model, the authors derive two novel upper bounds on Rademacher complexity: one is an upper bound that is independent of width but exponential in depth, and the other is an upper bound that is independent of width and depth when the weights are constrained below a certain threshold. These novel theory bounds bring new insights into understanding the generalization of deep learning.

**Questions:**

Please refer to the Weaknesses for details.

**Ethical Concerns:**

["NO or VERY MINOR ethics concerns only"]

**Final Justification:**

The authors' response partially addressed my concerns. I recommend acceptance on the condition that the authors include a summary of the key points of their theoretical proof and an explanation of the essence of the theoretical improvements in the final version to improve readability.

**Limitations:**

This work does not seem to have any potential negative societal impact.

**Paper Formatting Concerns:**

NO.

**Quality:**

3

**Strengths And Weaknesses:**

Strengths:

1. This paper extends the RKBS framework to neural networks of arbitrary topology (including Transformer) without the need for "wide network" or "lazy training" assumptions.

2. The derived upper bound of Rademacher complexity has no nuisance factors, which avoids the $2^D$ term induced by the traditional "peeling" argument. This is the most important theoretical contribution of this paper.

3. The proposed model is compatible with complex structures such as residual connections and attention mechanisms, and provides a modular recursive definition of the corresponding bilinear representation.

Weaknesses:

1. In my opinion, the greatest contribution of this paper is that the upper bound of Rademacher complexity can be analyzed under the RHBS framework to avoid nuisance factors. However, the readability of this paper is relatively poor. From the perspective of theoretical techniques, where does this improvement essentially come from? The authors need to add relevant clarifications and explanations to improve readability.

2. In addition to avoiding nuisance factors, what special advantages do the theoretical results obtained in this paper under the RHBS framework have? Compared with the NTK model, in addition to not requiring assumptions about network width and being an accurate model without approximation, what other essential advantages does the RKBS model have? In order to enhance the readability of the paper, some relevant discussions and comparative analyses are necessary.

3. In order to obtain a complexity bound that is independent of depth, the weights and biases need to be constrained below certain thresholds. Do these threshold constraints affect the actual generalization performance? What do they correspond to in practice? Due to the lack of experimental verification, it is somewhat difficult to understand, although this is a theoretical work. Some explanation and discussion of these threshold constraints is necessary.

4. Failure to summarize and present the key steps of the theoretical proof in the main paper will undermine the theoretical contribution of this paper. It is recommended to add key proof ideas and steps in the main paper to increase readability and theoretical novelty.

---

> ### Author Rebuttal · Authors · 2025-07-30
>
> **Response to reviewer oQnH:**
>
> **Q1:** In my opinion, the greatest contribution of this paper is that the upper bound of Rademacher complexity can be analyzed under the RHBS framework to avoid nuisance factors... where does this improvement essentially come from?
>
> **A:** Roughly, the nuisance factors seem to come from the fact that Golowich et al (and other peeling-based bounds) work directly with the Rademacher complexity when recursing through the network. To take a simple example, starting from the output layer (or any intermediate latyer), the Rademacher complexity of the neural network terminated at that layer can be re-written in terms of the Rademacher complexity of the neural network terminated at the previous layer using the Rademacher contraction principle, which states that, for an L-Lipschitz neural activation $\tau$:
>
> $\mathcal{R}_N (\tau(\mathbb{A})) = 2L \mathcal{R}_N (\mathbb{A})$
>
> (weights are also factored in, but this does not introduce additional nuisance factors in general). Applying this recursively for a $D$-layer network gives a multiplicatively built-up nuisance factor $2^D$. The important point to note here is that *the nuisance factor is a side-effect of the Rademacher contraction principle* - if we can avoid using this principle, we may avoid the introduction of the nuisance factors.
>
> The key difference in our approach is that, rather than diving directly into the question of Rademacher complexity, we first re-write the network as a bilinear product. This allows us to bound the Rademacher complexity in terms of the continuity constant $C_{\mathbb{W}}$ (which is a property of the bilinear form) and the norm of the data feature map $\phi$ as our first step, using an approach analogous to that used in the analysis of e.g. SVM Rademacher complexity. Only after this do we bound $C_{\mathbb{W}}$ and $\phi$ using a procedure analogous to peeling using the method sketched in the body of the paper and detailed in appendix D. The key observation is that, because we are no longer dealing directly with the Rademacher complexity but rather matrix/vector norms, we can use inequality bounds that do not have nuisance factors (in contrast to the Rademacher contraction principle). *Hence our peeling procedure does not introduce nuisance factors into the bound*.
>
> We have expanded our discussion in the paper to make this point more clearly (unfortunately we cannot upload the new version as yet due to the pdf restrictions in the review process).
>
>
> **Q2:** In addition to avoiding nuisance factors, what special advantages do the theoretical results obtained in this paper under the RHBS framework have? Compared with the NTK model, in addition to not requiring assumptions about network width and being an accurate model without approximation, what other essential advantages does the RKBS model have?
>
> **A:** There are two immediate benefits to our RKBS framework:
>
> - it enables us to analyse e.g. the Rademacher complexity of neural networks in the abstract, rather than directly, allowing us to remove nuisance factors from the bound.
>
> - it opens up the possibility of an exact analogue of the NTK to better understand the evolution of the NN during training. In this paper we have formulated our theory using the origin (zero weights and biases) as our "zero point", which rules out a straight-forward NTK-type (ie gradient flow) analysis that takes the current weights (at some point during training) as the "zero point" and formulates the change in the network relative to this. However it is possible to rewrite much of the analysis using an arbitrary "zero point", enabling an analysis more analogous to (corrected) gradient flow than the simpler version presented in the paper. However the resulting formulation is significantly more complicated and very notation-heavy, so we thought it best to publish this simpler precursor (and the insights that arise from it) first before proceeding to the more difficult case.
>
> More generally, history shows that re-casting complex models into a simpler (equivalent) dual forms that cleanly disentangle the role of the factors involved - in this case network structure, network parameters (weight/biases), and input data - can lead, directly or indirectly, to better understanding and practical (algorithmic) advances.
>
>
> **Q3:** Do the [norm-bounds on the weights and biases] affect the actual generalization performance? What do they correspond to in practice?
>
> **A:** The effect of these thresholds is directly observable in the Rademacher complexity bounds. It is difficult to say too much about these in practice without extensive experimental work that is beyond our means at present, but in the unbiased, wide network or lazy training case we may make "first-order" assumptions - LeCun or He initialization, weights close to their initialization - and then, as discussed in section 4 of the paper, we would expect that any increase over $\phi = 1$ would cause a depth-exponential increase in the Rademacher complexity.
>
>
> **Q4:** ...present the key steps of the theoretical proof in the main paper...
>
> **A:** We have endeavoured to do this (see lines 160-167 and 206-209), but on reflection in our efforts to reduce to mathematical load we may have made these summaries too brief, so we will expand them somewhat (particularly in line with our response to Q1) to improve readability and better highlight the key points of difference in our method.

---

> > ### Comment · Reviewer_oQnH · 2025-08-06
> >
> > The authors' responses partially addressed my concerns, and I maintain my score.

---

### Decision · Program_Chairs · 2025-09-17

**Decision:**

Accept (poster)

**Comment:**

This manuscript develops a reproducing kernel Banach space (RKBS) model for neural networks; the model is then used to produce upper bounds on Rademacher complexity of a broad class of models.

The main strength of the manuscript is its approach to the theoretical analysis and, more significantly, that the RKBS approach yields bounds free of typical "nuisance" parameters. Given the latent importance of such bounds in understanding generalization capabilities there is a clear contribution here.

The main weakness of the manuscript is its presentation: it can be challenging to read and technically dense. As such, the submitted manuscript did raise several questions that were clarified in the rebuttal. Nevertheless, further refinement of the presentation is important.